# SCALABLE EXPLORATION VIA ENSEMBLE++

## ABSTRACT

Scalable exploration in high-dimensional, complex environments is a significant challenge in sequential decision making, especially when utilizing neural networks. Ensemble sampling, a practical approximation of Thompson sampling, is widely adopted but often suffers performance degradation due to *ensemble coupling* in shared layer architectures, leading to reduced diversity and ineffective exploration. In this paper, we introduce Ensemble++, a novel method that addresses these challenges through architectural and algorithmic innovations. To prevent ensemble coupling, Ensemble++ decouples mean and uncertainty estimation by separating the base network and ensemble components, employs a symmetrized loss function and the stop-gradient operator. To further enhance exploration, it generates richer hypothesis spaces through random linear combinations of ensemble components using continuous index sampling. Theoretically, we prove that Ensemble++ matches the regret bounds of exact Thompson sampling in linear contextual bandits while maintaining a scalable per-step computational complexity of $\tilde{O}(\log T)$. This provides the first rigorous analysis demonstrating that ensemble sampling can be an scalable and effective approximation to Thompson Sampling, closing a key theoretical gap in exploration efficiency. Empirically, we demonstrate Ensemble++'s effectiveness in both regret minimization and computational efficiency across a range of nonlinear bandit environments, including a language-based contextual bandits where the agents employ GPT backbones. Our results highlight the capability of Ensemble++ for real-time adaptation in complex environments where computational and data collection budgets are constrained.
⌂ https://anonymous.4open.science/r/EnsemblePlus2-1E54

## 1 INTRODUCTION

Sequential decision-making under uncertainty is a fundamental challenge in machine learning, with critical applications in reinforcement learning, contextual bandits, recommendation systems, autonomous systems, and healthcare (Russo et al., 2018). In these domains, agents must balance the dual objectives of *exploration*—gathering new information about the environment—and *exploitation*—maximizing rewards based on current knowledge. This balance is particularly challenging in high-dimensional, complex environments where computational resources are limited and data collection is costly.

A key requirement for effective decision-making in such settings is the accurate estimation and utilization of *epistemic uncertainty*, which arises from incomplete knowledge about the environment. Traditional methods like *Thompson Sampling* (TS) (Thompson, 1933) provide a principled approach to balancing exploration and exploitation by sampling from the posterior distribution over model parameters. While TS achieves optimal regret bounds in simple settings with conjugate priors, it becomes computationally infeasible in high-dimensional environments involving complex models like neural networks, where exact posterior updates and sampling are intractable (Russo et al., 2018).

Approximate methods have been proposed to overcome these computational challenges. *Randomized Least Squares* (RLS) (Osband et al., 2019) leverages neural function approximation but requires retraining or resampling historical data at each decision step, leading to unbounded computational complexity over time. Ensemble-based methods, such as *Bootstrapped Thompson Sampling* (Osband & Van Roy, 2015; Osband et al., 2016) and *Ensemble+* (Osband et al., 2018; 2019), maintain multiple models to approximate the posterior distribution and support incremental updates, avoiding the need for full retraining. However, these methods face significant limitations:

- **Computational Overhead**: Maintaining multiple models increases computational and memory costs, especially in high-dimensional settings.
- **Ensemble Coupling in Shared-Layer Architectures**: To reduce computational load, ensemble members often share layers (Osband et al., 2016; 2018; 2019; Lee et al., 2021). This introduces gradient coupling across ensembles during training, leading to ineffective exploration.

These challenges highlight a critical gap in the literature: the need for scalable exploration methods that can handle high-dimensional neural function approximation, maintain bounded per-step computational complexity, and achieve effective exploration by accurately estimating epistemic uncertainty. In this paper, we propose *Ensemble++*, a novel method designed to address the challenges of scalable exploration in high-dimensional environments through architectural and algorithmic innovations that enhance both computational efficiency and exploration diversity.

**Our main contributions are:**

1. **Scalable Exploration Algorithm Design**: We introduce Ensemble++, which mitigates ensemble coupling by:
   - **Decoupled Optimization**: Separating mean and variance estimation into distinct modules with a symmetrized optimization objective, and employing a stop-gradient operator to enable independent learning across ensemble components.
   - **Ensemble++ Sampling**: Introducing richer exploration dynamics through linear combinations of ensemble components with carefully designed random weights.

2. **Theoretical Analysis and Guarantees**: We provide rigorous theoretical analysis demonstrating that Ensemble++ matches the regret bounds of exact Thompson Sampling under linear function approximation while achieving a scalable per-step computational complexity of $O(d^3 \log T)$. This exponentially improves over prior approximate Thompson Sampling methods (Xu et al., 2022; Qin et al., 2022), offering the first rigorous analysis showing ensemble sampling as a scalable and effective approximation to Thompson Sampling.

3. **Empirical Validation**: In Figure 1, Section 6, and Appendix I.2, we empirically show that Ensemble++ outperforms existing methods in both regret performance and computational efficiency across a range of nonlinear bandit environments, including language-input contextual bandits using a GPT backbone. We also perform extensive simulations in linear bandit environments (Appendix I.1) to validate our theoretical guarantee.

By bridging the gap between computational efficiency and exploration quality, Ensemble++ offers a robust framework for real-time adaptation in complex environments. Our method is particularly suitable for applications where both computational and data collection budgets are constrained.

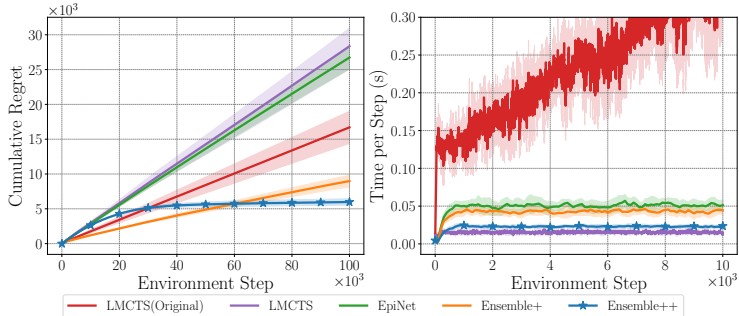

Figure 1: Comparison of cumulative regret and per-step computational time among Ensemble+ (Osband et al., 2018; 2019), EpiNet (Osband et al., 2023), and LMC-TS (Xu et al., 2022) in a nonlinear bandit environment. Ensemble++ achieves the sublinear regret (effective exploration) with bounded per-step computational costs while others suffer linear regret (ineffective exploration). LMCTS(Original) follows official implementation with neural function approximation, incurring increasing per-step computational costs. All other agents employs comparable layer-sharing neural architectures and optimization configurations. (Details in Appendix I.3).

**Organization of the Paper** The remainder of this paper is organized as follows: In Sections 2 and 3, we introduce the background knowledge and analyze the problem of ensemble coupling and

sequential dependency in shared-layer ensembles. Section 4 introduces the Ensemble++ method, detailing its architecture, training, and sampling strategies. In Section 5, we provide theoretical insights and regret analysis of Ensemble++ in the context of linear contextual bandits. Empirical results demonstrating the effectiveness of Ensemble++ are presented in Section 6. We conclude with discussions and potential future work in Section 7. Section A reviews related work and discusses their limitations. Detailed comparisons and additional discussions are provided in the appendices.

## 2 BACKGROUND

In this section, we formalize the sequential decision-making problem and introduce ensemble methods using standard notation.

**Sequential Decision Making** We consider a sequential decision-making problem over a discrete time horizon $T$. At each time step $t$, the agent observes an action set $\mathcal{A}_t \subseteq \mathcal{A}$, selects an action $A_t \in \mathcal{A}_t$ based on the history $\mathcal{H}_t = \{\mathcal{A}_1, A_1, Y_1, \ldots, \mathcal{A}_{t-1}, A_{t-1}, Y_{t-1}, \mathcal{A}_t\}$, and receives a reward $Y_t = f^*(A_t) + \epsilon_t$, where $f^*$ is the unknown reward function and $\epsilon_t$ is zero-mean noise. The objective is to minimize the *cumulative regret*: $R(T) = \sum_{t=1}^{T} \left( \max_{a \in \mathcal{A}_t} f^*(a) - f^*(A_t) \right)$.

**Ensemble Sampling** Ensemble sampling methods (Lu & Van Roy, 2017) , including *Bootstrapped DQN* (Osband et al., 2016) and Ensemble+ (Osband et al., 2018; 2019), approximate the posterior distribution of $f^*$ by maintaining multiple models, each representing a hypothesis about $f^*$ based on historical data. An ensemble consists of $M$ models with parameters $\theta_t = \{\theta_{t,m}\}_{m=1}^{M}$. At each time step $t$, the agent samples an ensemble member $m_t$ uniformly from $\{1, 2, \ldots, M\}$ and selects an action: $A_t = \arg\max_{a \in \mathcal{A}_t} f_{\theta_{t,m_t}}(a)$, where $f_{\theta_{t,m_t}}(a)$ is the prediction of ensemble member $m_t$ for action $a$. After observing the reward $Y_t$, each ensemble member $m$ updates its parameters $\theta_{t+1,m}$ by performing stochastic gradient descent on the loss starting from previous iterate $\theta_{t,m}$:

$$L(\theta_{t+1,m}; D) = \sum_{s=1}^{t} \left( Y_s + Z_{s,m} - f_{\theta_{t+1,m}}(A_s) \right)^2 + \Psi(\theta_{t+1,m}) \tag{1}$$

Here, $D = \mathcal{H}_t$ and $Z_{s,m}$ are independent random perturbations added to encourage diversity among ensemble members, and $\Psi(\theta_{t+1,m})$ is a regularization term. This perturbed training procedure ensures that each ensemble member captures different aspects of the uncertainty in $f^*$, representing different plausible hypotheses consistent with the history. The random perturbations $Z_{s,m}$ are independent across time index $s$ and model index $m$. Once realized, $Z_{s,m}$ are fixed throughout the rest of the training, enabling incremental updates for real-time adaptation. This is a key computational feature compared to methods like Randomized Least Squares (RLS) (Osband et al., 2019) or Perturbed History Exploration (PHE) (Kveton et al., 2020a). In RLS and PHE, fresh independent perturbations for all historical data are introduced at each time $t$, and the model requires full retraining from scratch to ensure diverse exploration of different plausible hypotheses.

## 3 ENSEMBLE COUPLING AND SEQUENTIAL DEPENDENCY

Ensemble methods are powerful tools for approximating the posterior distribution of an unknown function $f^*$ in sequential decision-making tasks. By maintaining multiple models, each providing independent estimates, ensembles help mitigate *sequential dependency*. However, when models are updated incrementally and ensemble members share layers, an *ensemble coupling issue* arises, leading to reduced ensemble diversity, reintroduction of sequential dependency, and, ultimately, linear regret. In this section, we analyze this phenomenon and explain why it occurs.

**Sequential Dependency in Incremental Updates** In sequential decision-making, agents aim to make optimal decisions under uncertainty by continuously updating their models based on new observations and selecting actions accordingly. Incremental updates, such as those in recursive methods, cause the parameters at time $t$ to depend on those at time $t-1$, creating a chain of dependencies. This introduces *sequential dependency*, wherein the model parameters and action selection become intertwined over time. In the context of *Randomized Least Squares* (RLS) (Osband et al., 2018; 2019), fresh independent perturbations at each time step are crucial to ensure that the sampled

parameters match the posterior distribution. Naive recursive implementations introduce sequential dependency because the action at time $t$ depends on past perturbations through the updated parameters, violating the independence required for accurate posterior approximation, i.e., *epistemic uncertainty estimation*. A detailed analysis of this issue is provided in Appendix B.2.

**Ensemble Methods and the Problem of Coupling**   To address sequential dependency while allowing incremental updates, ensemble methods like *Ensemble Sampling* (ES) have been proposed (Osband & Van Roy, 2015; Osband et al., 2016; Lu & Van Roy, 2017). ES maintains multiple independent models, each with its own perturbations, and selects actions by randomly choosing a model at each time step. This approach reduces sequential dependency over time by decoupling model updates from action selection. Connections between RLS, Recursive RLS (RRLS), and ensemble methods are discussed in Appendix B.3.

However, when ensemble members share layers—a common practice in neural network implementations due to computational efficiency—an *ensemble coupling issue* emerges. In shared-layer ensembles, ensemble members share the hidden representations but have separate output layers. The shared parameters are updated based on the aggregated gradients from all ensemble members, which leads to several problems. First, *gradient interference* occurs because the random perturbations introduced to encourage diversity cause each ensemble member to have different loss landscapes. Their gradients with respect to the shared parameters may point in conflicting directions, resulting in destructive interference during updates. This interference hampers the effective learning of shared features that support diverse hypotheses about $f^*$. Second, there is a *homogenization of ensemble members*. Since all ensemble members share the same feature extractor, the hidden representations become similar across members. The shared feature extractor tends to capture common patterns favored by the majority, reducing the uniqueness of individual ensemble members' internal representations. Consequently, the ensemble members produce *correlated predictions*, leading to reduced ensemble diversity. The variance among ensemble outputs decreases, undermining the ensemble's ability to represent uncertainty effectively. This homogenization reintroduces sequential dependency because the shared parameters $w$ depend on the entire history of actions and observations, creating a feedback loop between the agent's actions and the model's parameters. Detailed mathematical analysis of this phenomenon is provided in Appendix B.4.

**Ineffective Exploration and Linear Regret**   The reduced diversity and reintroduction of sequential dependency have significant impacts on the agent's performance. The ensemble fails to represent a wide range of plausible hypotheses about $f^*$, leading to *ineffective exploration* of the action space. As ensemble members suggest similar actions, the agent tends to exploit known actions and fails to discover better ones. This results in *persistent suboptimal decisions*. Moreover, the cumulative regret $R(T)$ grows linearly with time $T$ because the agent consistently misses opportunities to find better actions through exploration. This linear regret indicates that the agent's performance does not improve adequately over time, which is undesirable in sequential decision-making tasks that require learning from experience. Experiments demonstrate that shared-layer ensembles exhibit

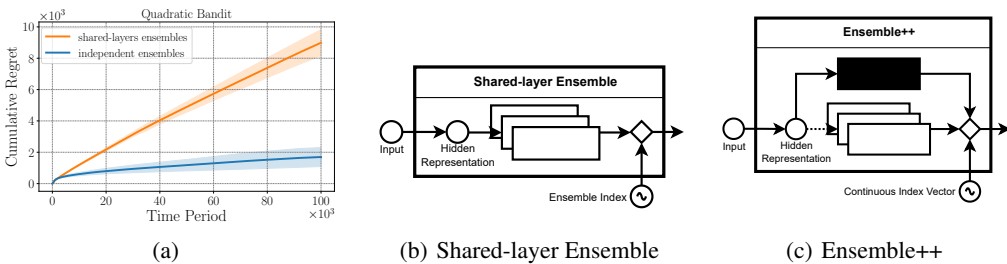

(a)                    (b) Shared-layer Ensemble              (c) Ensemble++

Figure 2: (a) Comparison between independent ensembles and shared-layer ensembles on a quadratic bandit problem. (b) Illustration of the shared-layer ensembles where the hidden representation $\tilde{x} = h(x; w)$ are shared and gradients from ensemble components are coupled, leading to interference and homogenization. (c) *Ensemble++*: The base network (black part) allows gradient flow for updating shared layers. Ensemble components introduce uncertainty through random linear combinations, with gradients blocked from affecting $\tilde{x}$ due to the stop-gradient operator $\mathrm{sg}(\cdot)$.

linear regret in tasks requiring effective exploration. For instance, in a quadratic bandit problem, independent ensembles achieve sublinear regret, effectively balancing exploration and exploitation. In contrast, shared-layer ensembles suffer from linear regret due to their inability to explore the action space sufficiently. This comparison is illustrated in Figure 2. Additional empirical results, including ablation studies that vary hyperparameters such as prior scale and the number of updates per step, further demonstrate the ineffectiveness of shared-layer ensembles in achieving sublinear regret. These results are detailed in Appendix B.5.

## 4 METHODS

In this section, we introduce **Ensemble++**, a novel method designed to address the challenges of ensemble coupling and sequential dependency in shared-layer ensembles.

**Key Design Principles** Ensemble++ is built upon the following key design principles: **1.** Decoupling mean and uncertainty estimation by separating the base network (mean estimator) and ensemble components (uncertainty estimators). **2.** Introducing symmetric auxiliary variables allows the training objective to decouple into separate loss terms for the base network and ensemble components, facilitating independent learning. **3.** Applying the stop-gradient operator to the shared representation ensures that gradients from the ensemble components do not affect the shared layers, further preventing coupling. **4.** Using continuous index vectors for random linear combinations of ensemble components enables richer exploration dynamics without increasing computational costs.

**Architecture** As illustrated in Figure 2 (c), the architecture of Ensemble++ consists of a shared feature extractor, a base network for mean estimation, and ensemble components for capturing uncertainty. Let $x \in \mathcal{X}$ denote the input, and $h(x; w)$ be the shared feature extractor parameterized by $w$. The extracted features are denoted by $\tilde{x} = h(x; w)$. The base network $\psi(\tilde{x}; b)$, parameterized by $b$, estimates the mean prediction based on the shared features. The ensemble components $\{\psi(\mathrm{sg}(\tilde{x}); \theta_m)\}_{m=1}^M$, parameterized by $\theta_m$, is designed to capture the uncertainty in the prediction. The stop-gradient operator $\mathrm{sg}(\cdot)$ prevents gradients from flowing through $\tilde{x}$ when computing gradients with respect to $\theta_m$, effectively decoupling the ensemble components from the shared layers. Put all together, $\theta = \{w, b, \theta_1, \ldots, \theta_M\}$ are the model parameters[1].

**Training Objective** To decouple the training of the base network and ensemble components, we introduce symmetric auxiliary variables $\beta \in \{1, -1\}$ into the loss function. For a dataset $D = \{(A_s, Y_s)\}_{s=1}^N$, the training objective is:

$$L(\theta; D) = \frac{1}{2M} \sum_{m=1}^M \sum_{s=1}^N \sum_{\beta \in \{1,-1\}} (Y_s + \beta \mathbf{z}_{s,m} - \psi(\tilde{x}_s; b) - \beta \psi(\mathrm{sg}(\tilde{x}_s); \theta_m))^2 + \Phi(\theta) \quad (2)$$

$$= \frac{1}{M} \sum_{m=1}^M \sum_{s=1}^N \left[ \frac{1}{2} (Y_s - \psi(\tilde{x}_s; b))^2 + \frac{1}{2} (\mathbf{z}_{s,m} - \psi(\mathrm{sg}(\tilde{x}_s); \theta_m))^2 \right] + \Phi(\theta), \quad (3)$$

where $\mathbf{z}_s = (\mathbf{z}_{s,1}, \ldots, \mathbf{z}_{s,M}) \in \mathbb{R}^M$ are independent perturbation vectors sampled from a zero-mean distribution $P_{\mathbf{z}}$, and $\Phi(\theta)$ is a regularization term (e.g., weight decay). The equivalence between equation 2 and equation 3 arises because the cross-terms cancel out when summing over $\beta$, effectively decoupling the loss into separate terms for the base network and ensemble components.

**Gradient Decoupling** By using the stop-gradient operator and this symmetrized loss function, the gradients with respect to the shared parameters $w$ and $b$ depend only on the base network loss, while the gradients with respect to the ensemble parameters $\theta_m$ depend only on the ensemble component loss. This decoupling prevents ensemble coupling and maintains ensemble diversity, ensuring that the shared representation $\tilde{x}$ is optimized for mean estimation, while ensemble components capture uncertainty independently. Detailed gradient derivations are provided in Appendix C. After analyzing the content for redundancy and clarity, we have revised the introduction of random linear combinations to avoid unnecessary mention of the time step $t$ and removed redundant explanations.

---

[1]For the sake of clarity, we omit the presentation of the fixed prior ensembles, detailed in Appendix C.

---

**Algorithm 1** Ensemble++ Sampling

---

1: **for** $t = 1$ to $T$ **do**
2:      Sample $\zeta_t \sim P_\zeta$
3:      Select action $A_t = \arg\max_{a \in \mathcal{A}_t} f_{\theta_t}^{++}(a, \zeta_t)$
4:      Observe reward $Y_t$ and update dataset $D \leftarrow D \cup \{(A_t, Y_t)\}$
5:      Sample a perturbation vector $\mathbf{z}_t \sim P_\mathbf{z}$
6:      Update $\theta_{t+1}$ via stochastic gradient descent on $L(\theta; D)$

---

**Random Linear Combinations** Ensemble++ enhances exploration by using continuous multi-dimensional index vectors $\zeta$ sampled from a distribution $P_\zeta$. Instead of selecting a single ensemble member, we generate new hypotheses through random linear combinations of the ensemble components. Define the ensemble parameters matrix $\mathbf{A} = [\theta_1, \ldots, \theta_M] \in \mathbb{R}^{d \times M}$. Sampling $\zeta \in \mathbb{R}^M$ from $P_\zeta$, we compute the model prediction as:

$$f_\theta^{++}(x, \zeta) = \psi(\tilde{x}; b) + \psi(\text{sg}(\tilde{x}); \sum_{m=1}^{M} \zeta_m \theta_m) = \psi(\tilde{x}; b) + \psi\left(\text{sg}(\tilde{x}); \mathbf{A}\zeta\right).$$

This approach effectively creates an infinite set of ensemble hypotheses from a finite $M$, enabling richer exploration without additional computational cost. Detailed discussion on index distribution design is provided in Appendix C.

The complete algorithm is summarized in Algorithm 1.

**Ensemble++ offers several advantages over traditional shared-layer ensembles**: By decoupling ensemble components from shared layers, Ensemble++ maintains diversity and captures uncertainty more accurately. By preventing ensemble coupling and sequential dependency, Ensemble++ achieves sublinear regret, outperforming shared-layer ensembles. Continuous index vectors and random linear combinations enable richer exploration dynamics without extra computational cost.

The architectural and algorithmic designs of Ensemble++ align with theoretical principles that ensure effective exploration and sublinear regret. In Section 5, we provide a rigorous regret analysis demonstrating that Ensemble++ matches the regret bounds of exact Thompson Sampling under linear function approximation while maintaining scalable computational complexity. The key design elements—*decoupling mean and uncertainty estimation*, *symmetrized loss function* and *random linear combinations of ensembles via continuous index vectors*—are critical in avoiding sequential dependency and enabling accurate uncertainty estimation, leading to efficient exploration and improved performance. Since there is no hidden feature learning in the linear setting, the stop-gradient design does not affect the theoretical analysis but is crucial in practice when using neural networks.

## 5    THEORETICAL INSIGHT AND REGRET ANALYSIS

In this section, we provide a comprehensive theoretical examination of Linear Ensemble++ within the context of linear contextual bandit problems. We demonstrate how Linear Ensemble++ effectively addresses the *sequential dependency problem* inherent in incremental model updates, ensuring accurate uncertainty estimation and achieving favorable regret bounds. The analysis leverages innovative techniques, including *variance-aware discretization* and *sequential random projection*, to overcome challenges posed by adaptive action selection and incremental updates in high-dimensional parameter spaces.

**Problem Setup and Assumptions** We consider a linear contextual bandit setting where, at each time step $t$, the agent selects an action $A_t \in \mathcal{A}_t \subseteq \mathbb{R}^d$ based on a known feature mapping $\phi : \mathcal{A} \to \mathbb{R}^d$ that maps each action $a \in \mathcal{A}$ to a feature vector $\phi(a)$ satisfying $\|\phi(a)\|_2 \leq 1$. The reward function $f^*$ is linearly realizable, i.e., there exists a parameter vector $\theta^* \in \mathbb{R}^d$ such that $f^*(a) = \langle \phi(a), \theta^* \rangle$. After selecting action $A_t$, the agent observes a noisy reward $Y_t = f^*(A_t) + \varepsilon_t$, where $\varepsilon_t$ is zero-mean sub-Gaussian noise.

**Assumption 1.** *The reward function $f^*$ is linearly realizable with respect to the feature mapping $\phi$, and the noise $\varepsilon_t$ satisfies $\mathbb{E}\left[\exp\{s\varepsilon_t\} \mid \mathcal{H}_t, A_t\right] \leq \exp\{s^2/2\}, \quad \forall s \in \mathbb{R}$, where $\mathcal{H}_t$ represents the history up to time $t$.*

**Linear Ensemble++**    In the linear case, we adapt Ensemble++ to the linear contextual bandit setting. Since the feature mapping $\phi$ is known and fixed, we consider the linear version of Ensemble++:

$$f_\theta^{\text{lin}}(a, \zeta_t) = \langle \phi(a), \mu_t \rangle + \langle \phi(a), \mathbf{A}_t \zeta_t \rangle,$$

where $\mu_t \in \mathbb{R}^d$ is the base parameter estimate at time $t$, $\mathbf{A}_t \in \mathbb{R}^{d \times M}$ is the matrix formed by ensemble components and $\zeta_t \in \mathbb{R}^M$ is sampled from an index distribution $P_\zeta$. The parameters $\mu_t$ and $\mathbf{A}_t$ are updated incrementally using observed data.

**Proposition 1** (Closed-Form Incremental Updates). *Linear Ensemble++ with the training objective analogous to equation 3, $\ell_2$-regularization, and full data buffer $D = \mathcal{H}_t$ permits closed-form incremental updates:*

$$\mu_t = \boldsymbol{\Sigma}_t \left( \boldsymbol{\Sigma}_{t-1}^{-1} \mu_{t-1} + \phi(A_t) Y_t \right), \quad \mathbf{A}_t = \boldsymbol{\Sigma}_t \left( \boldsymbol{\Sigma}_{t-1}^{-1} \mathbf{A}_{t-1} + \phi(A_t) \mathbf{z}_t^\top \right), \tag{4}$$

*with $\boldsymbol{\Sigma}_t^{-1} = \boldsymbol{\Sigma}_{t-1}^{-1} + \phi(A_t)\phi(A_t)^\top$, $\mu_0 = 0$, and $\mathbf{A}_0 = \boldsymbol{\Sigma}_0^{1/2} \mathbf{Z}_0$, where $\boldsymbol{\Sigma}_0^{-1} = \lambda I$ and $\mathbf{Z}_0 = (\mathbf{z}_{0,1}, \dots, \mathbf{z}_{0,M})$ with $\mathbf{z}_{0,i} \sim P_Z$. $\boldsymbol{\Sigma}_t$ is updated incrementally via the Sherman-Morrison formula.*

**Explanation**    These updates mirror the recursive least squares (RLS) updates but include the ensemble components. The base estimate $\mu_t$ is updated using the observed reward $Y_t$, while the ensemble components $\mathbf{A}_t$ are updated using the perturbation vectors $\mathbf{z}_t$. Due to matrix multiplication, the computational complexity of each incremental update is $O(d^2 M)$, essentially depending on the number of the ensemble members $M$.

## 5.1 Key Lemma: Incremental Uncertainty Estimation

Linear Ensemble++ maintains parameter estimates $\mu_t \in \mathbb{R}^d$ and $\mathbf{A}_t \in \mathbb{R}^{d \times M}$, updated incrementally as per Proposition 1. A central innovation of Linear Ensemble++ is encapsulated in the following lemma, which ensures that the ensemble-based uncertainty estimates accurately reflect the true posterior variance despite sequential dependencies introduced by incremental updates.

Denote $s_{\min}^2 = \inf_{\|a\|_2=1} a^\top \boldsymbol{\Sigma}_0^{-1} a$ and $s_{\max}^2 = \sup_{\|a\|_2=1} a^\top \boldsymbol{\Sigma}_0^{-1} a$.

**Lemma 1** (Incremental Uncertainty Estimation). *Under the linear update rule in equation 4, if the perturbation distribution $P_{\mathbf{z}}$ is $\sqrt{\frac{1}{M}}$-sub-Gaussian and unit-norm, then for*

$$M \geq 320 \left( d \log \left( \frac{2 + \frac{96}{s_{\min}} \sqrt{s_{\max}^2 + T}}{\delta} \right) + \log \left( 1 + \frac{T}{s_{\min}^2} \right) \right) \simeq d \log T, \tag{5}$$

*the joint event $\mathcal{G} = \bigcap_{t=0}^T \mathcal{G}_t$ holds with probability at least $1 - \delta$, where*

$$\mathcal{G}_t = \left\{ \frac{1}{2} \phi(a)^\top \boldsymbol{\Sigma}_t \phi(a) \leq \phi(a)^\top \mathbf{A}_t \mathbf{A}_t^\top \phi(a) \leq \frac{3}{2} \phi(a)^\top \boldsymbol{\Sigma}_t \phi(a), \quad \forall a \in \mathcal{A}_t \right\}.$$

**Significance**    This lemma guarantees that the ensemble-based uncertainty estimates $\mathbf{A}_t \mathbf{A}_t^\top$ remain closely aligned with the true posterior covariance $\boldsymbol{\Sigma}_t$. Specifically, it bounds the estimated variance within a $1/2$-factor approximation of the true variance for all actions and time steps, ensuring reliable uncertainty estimation crucial for effective exploration and exploitation.

**Technical Innovation**    The primary technical challenge stems from the sequential dependence between the random perturbation vectors and the high-dimensional random variables in the decision-making process. To overcome this, we introduce two key innovations: (1) a variance-aware discretization argument that avoids the exponential increase in ensemble size $M$, and (2) a reduction to sequential random projection techniques (Li, 2024a). Unlike classical discretization methods requiring $M = \Omega(dT^2 \log T)$, our approach ensures scalability and computational efficiency by maintaining a logarithmic growth over time $T$. Detailed proofs are provided in Appendix E.

## 5.2 Regret Bound for Linear Ensemble++

Using Lemma 1 and existing theoretical frameworks for linear bandits, we establish the regret bound for Linear Ensemble++.

**Theorem 1** (Distribution-dependent Regret). *Under Assumption 1, with the conditions for Proposition 1 and Lemma 1, and ensemble size $M$ satisfying equation 5, Linear Ensemble++ achieves:*

$$R(T) \leq \frac{\rho(P_\zeta)}{p(P_\zeta)} \beta \sqrt{Td \log\left(1 + \frac{T}{\lambda d}\right)},$$

*with probability at least $1 - \delta$, where $\beta = \sqrt{\lambda}\|\theta^*\|_2 + \sqrt{2\log(1/\delta)}$, and $\rho(P_\zeta)$ and $p(P_\zeta)$ are constants depending on the index distribution $P_\zeta$.*

**Index Reference Distributions** Table 1 summarizes the values of $\rho(P_\zeta)$ and $p(P_\zeta)$ for different distributions, influencing the regret via their ratio. For continuous-support distributions like Gaussian and Spherical, this ratio leads to tighter bounds compared to discrete distributions such as Cube or Coordinate.

| $P_\zeta$ | Gaussian $\mathcal{N}(0, I_M)$ | Spherical $\sqrt{M} \cdot \mathcal{U}(\mathbb{S}^{M-1})$ | Cube $\mathcal{U}(\{1, -1\}^M)$ | Coordinate $\mathcal{U}(\{\pm e_i\}_{i\in[M]})$ | Sparse |
|---|---|---|---|---|---|
| $\rho(P_\zeta)$ | $\rho_1 \wedge \rho_3$ | $\rho_2 \wedge \rho_3$ | $\rho_2 \wedge \rho_3$ | $\rho_2$ | $\rho_2$ |
| $p(P_\zeta)$ | $\frac{1}{4\sqrt{e\pi}}$ | $\frac{1}{2} - \frac{e^{1/12}}{\sqrt{2\pi}}$ | $\frac{7}{32}$ | $\frac{1}{2M}$ | N/A |

Table 1: Values of $\rho(P_\zeta)$ and $p(P_\zeta)$ for index reference distributions $P_\zeta$. These influence the regret bound via the ratio $\frac{\rho(P_\zeta)}{p(P_\zeta)}$. Here $\rho_1 = O(\sqrt{M \log(\frac{M}{\delta})})$, $\rho_2 = O(\sqrt{M})$, and $\rho_3 = O(\sqrt{\log(\frac{|\mathcal{A}|}{\delta})})$

**Implications.** Continuous-support distributions like Gaussian and Spherical provide a more favorable $\rho(P_\zeta)/p(P_\zeta)$ ratio compared to discrete distributions such as Cube and Coordinate. This results in tighter regret bounds and more efficient exploration-exploitation trade-offs. Specifically, for $M = \Theta(d \log T)$, Linear Ensemble++ with continuous-support $P_\zeta$ achieves a regret bound that does not scale adversely with $M$, unlike coordinate-based index reference distribution. In settings with finite decision sets, where the ratio becomes $\tilde{O}(\log |\mathcal{A}|)$, this ensures scalability and performance without incurring additional regret costs as $M$ increases.

The proof of Theorem 1 can be found in Appendix F. We provide two important remarks highlighting the theoretical achievements of Ensemble++.

**Remark 1** (Efficiency). *As shown in Table 2, Ensemble++ is the first approximate TS method that achieves both provable scalability with $O(d^3 \log T)$ per-step computation and near-optimal regret matching exact TS (Agrawal & Goyal, 2013; Abeille & Lazaric, 2017) across all decision set setups. Notably, Ensemble++ achieves an exponential improvement in the $T$-dependency of per-step computational complexity compared to prior methods (Qin et al., 2022; Xu et al., 2022), and an $O(d(\log T)^2)$ improvement in the regret bound compared to concurrent ensemble sampling work (Janz et al., 2024). This closes a long-standing gap in scalable exploration since the introduction of ensemble methods in exploration (Osband & Van Roy, 2015; Lu & Van Roy, 2017).*

**Remark 2** (Flexibility). *The theoretical guarantees of Linear Ensemble++ hold for both finite and compact, as well as time-variant and time-invariant decision sets, providing broad applicability. Existing frequentist regret bounds for TS (Abeille & Lazaric, 2017) and approximate TS methods,*

Table 2: Regret upper bounds under various decision set setups in linear contextual bandits. Per-step computational complexity is $\Omega(d^2 + d|\mathcal{A}|T)$ for ES (Qin et al., 2022), $\Theta(d^3 \log T)$ for ES (Janz et al., 2024), and $\Theta(d^3 \log T)$ for our **Ensemble++**.

| Algorithm | Invariant & Compact | Variant & Compact | Invariant & Finite | Variant & Finite |
|---|---|---|---|---|
| TS | $O\left(d^{\frac{3}{2}}\sqrt{T}\log T\right)$ | $O\left(d^{\frac{3}{2}}\sqrt{T}\log T\right)$ | $O\left(d\sqrt{T\log|\mathcal{A}|}\log T\right)$ | $O\left(d\sqrt{T\log|\mathcal{A}|}\log T\right)$ |
| ES (Qin et al., 2022) | N/A | N/A | $O\left(\sqrt{dT\log|\mathcal{A}|\log\left(\frac{|\mathcal{A}|T}{d}\right)}\right)$ | N/A |
| ES (Janz et al., 2024) | $O\left((d\log T)^{\frac{5}{2}}\sqrt{T}\right)$ | $O\left((d\log T)^{\frac{5}{2}}\sqrt{T}\right)$ | N/A | N/A |
| **Ensemble++** | $O\left(d^{\frac{3}{2}}\sqrt{T}(\log T)^{\frac{3}{2}}\right)$ | $O\left(d^{\frac{3}{2}}\sqrt{T}(\log T)^{\frac{3}{2}}\right)$ | $O\left(d\sqrt{T\log|\mathcal{A}|}\log T\right)$ | $O\left(d\sqrt{T\log|\mathcal{A}|}\log T\right)$ |

*including LMC-TS (Xu et al., 2022) and ensemble sampling (Janz et al., 2024), are specialized to compact decision sets. The Bayesian analysis of ensemble sampling (Qin et al., 2022) applies only to time-invariant finite decision sets.*

# 6 EXPERIMENTS

Contextual bandits are a variant of the bandit problem where additional information (the *context*) influences the reward associated with an action. Formally, at each time step $t$, the agent observes a context $X_t \in \mathcal{X}$ and selects an action $A_t \in \mathcal{A}_t(X_t)$, where the action set $\mathcal{A}_t(X_t) \subseteq \mathcal{A}$ may depend on the observed context, i.e., $\mathcal{A}_t = \mathcal{A}(X_t)$. It has wide applications ranging from recommendation systems to online content moderation as discussed in Appendix H. We conduct comprehensive experiments on nonlinear contextual bandits to validate our main theoretical insights. Additionally, we provide detailed findings on linear contextual bandits in Appendix I.1.

We evaluate Ensemble++ in various nonlinear contextual bandits and conduct ablation studies to highlight the contributions of key design elements: (1) the stop-gradient operator, (2) the symmetrized optimization objective and (3) linear combinations of ensembles.

**Settings:** We consider three different nonlinear tasks: 1) **Quadratic Bandit**: Adapted from Zhou et al. (2020), the reward function is expressed as $f(a) = 10^{-2}(a^\top \Theta \Theta^\top a)$. Here, $a \in \mathbb{R}^d$ represents the action, while $\Theta \in \mathbb{R}^{d \times d}$ is a matrix filled with random variables from $\mathcal{N}(0, 1)$. 2) **UCI Mushroom**: Following prior works (Riquelme et al., 2018; Kveton et al., 2020b), we conduct contextual bandits with $N$-class classification using the UCI Mushroom dataset (Asuncion et al., 2007). 3) **Hate Speech Detection**: We leverage a language dataset[2] to build this bandit task. The agent must decide whether to publish or block content. Blocking any content yields a reward of 0.5. Publishing "free" content earns a reward of 1, while publishing "hate" content incurs a penalty of -0.5. This bandit task can further be extended to real applications such as online content moderation. A detailed description of these nonlinear bandits is provided in Appendix I.2. We apply 2-layer MLP networks with 64 units as the network backbone for the first two tasks and GPT-2[3] for the third one.

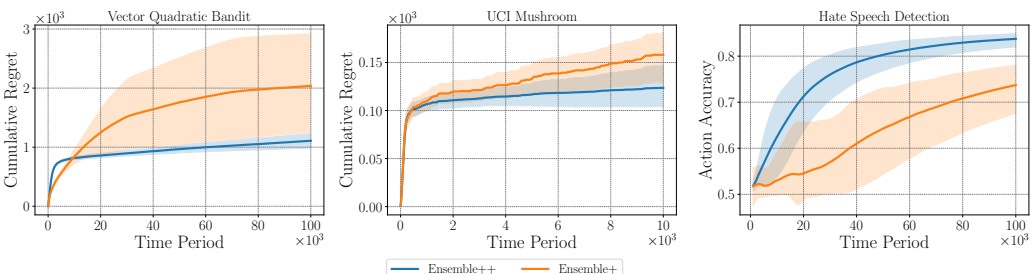

Figure 3: Comparison results across various nonlinear contextual bandits.

**Results Analysis:** We first consider Ensemble+ (Osband et al., 2018) with shared-layer ensembles as a strong baseline. The comparison results presented in Figure 3 demonstrate that our Ensemble++ consistently achieves sublinear regret and higher accuracy. Notably, in the Hate Speech Detection task, Ensemble++ outperforms Ensemble+ by 13.5%, underscoring its scalability in dealing with more complex networks, such as Transformers. Furthermore, we compare Ensemble++ with other baselines, such as EpiNet (Osband et al., 2023) and LMCTS (Xu et al., 2022) on extensive nonlinear bandit environments, as discussed in Figure 16 of Appendix I.2. In these comparisons, Ensemble++ consistently achieves sublinear regret with bounded per-step computation and outperforms the other methods.

**Ablation Studies:** We use the Quadratic Bandit as the testbed for ablation studies. Figure 4(a) demonstrates that the stop-gradient operator is crucial for achieving sublinear regret. The linear regret observed when applying the Onehot distribution for both reference and update in Figure 4(b) indicates that the symmetrized optimization objective achieved by either Sphere or Coord index is

---

[2]https://huggingface.co/datasets/ucberkeley-dlab/measuring-hate-speech
[3]https://huggingface.co/openai-community/gpt2

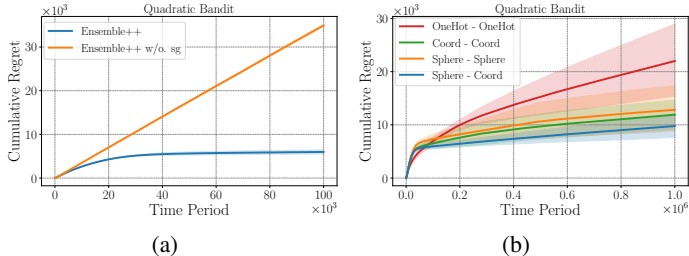

(a)                                           (b)

Figure 4: Experimental results on Quadratic Bandit. (a) Ablation study on the stop-gradient operator $\text{sg}(\cdot)$ in Equation (2). (b) Comparison results on different combination between update and reference distribution. The label $A - B$ indicates that Ensemble++ uses A as the reference distribution and B as the update distribution.

also essential. Additionally, the sublinear regret achieved by continuous-support (Sphere) reference index shows that linear combinations of ensembles bring about more efficient exploration.

We also conduct comparisons involving different perturbation distributions and other ablation studies on #ensembles $M$ and buffer size in Appendix I.2. Our findings indicate that the perturbation distribution does not significantly influence performance when neural network is involved. Moreover, Ensemble++ can achieve satisfactory performance even with a limited buffer size.

## 7    CONCLUSION AND FUTURE WORK

In this work, we presented *Ensemble++*, a novel method designed to address the challenges of scalable exploration in high-dimensional sequential decision-making tasks. By introducing architectural and algorithmic innovations—including the decoupling of mean and uncertainty estimation via a symmetrized loss function, the use of the stop-gradient operator to prevent gradient coupling, and enhanced exploration through continuous index sampling—Ensemble++ effectively mitigates the ensemble coupling issue inherent in shared-layer ensembles.

Our theoretical analysis demonstrates that Ensemble++ achieves regret bounds comparable to exact Thompson Sampling under linear function approximation, while maintaining a scalable per-step computational complexity of $O(d^3 \log T)$. This establishes Ensemble++ as the first approximate Thompson Sampling method to offer both provable scalability and near-optimal regret across various decision set configurations. These theoretical results validate the effectiveness of our algorithmic design schemes, including the separation of heads, the symmetrized loss function, and the continuous index sampling.

Empirical evaluations on a diverse set of nonlinear bandit tasks, including language-input contextual bandits using a GPT backbone, confirm the practical efficacy of Ensemble++. Ensemble++ consistently outperforms state-of-the-art methods such as Ensemble+, EpiNet, and LMC-TS, achieving superior regret performance and computational efficiency.

**Future Directions**    While Ensemble++ shows significant promise, several avenues for future research remain. Extending Ensemble++ to more complex environments, such as reinforcement learning with delayed rewards, multi-agent systems, and continuous action spaces, could further demonstrate its versatility. Investigating alternative index distributions and sampling strategies may yield additional improvements in exploration efficiency and theoretical guarantees. Moreover, integrating Ensemble++ with modern foundation models like large language models opens up possibilities for scalable exploration, multi-step reasoning, and planning in even more complex domains.

In summary, Ensemble++ represents a significant advancement in ensemble-based uncertainty estimation for sequential decision-making, providing scalable exploration suitable for real-time applications.

**Ethics Statement:**   This research was conducted in compliance with all applicable ethical guide-lines and institutional regulations. Since the study did not involve human participants, animals, or sensitive data, no specific ethical approvals were required. All data used in this research were obtained from publicly available sources, ensuring full transparency and reproducibility of the results.

**Reproducibility Statement:**   Detailed settings for the experiments can be found in Section 6 and Appendix I. We conduct the experiments on linear bandits using only CPUs, and the experiments on nonlinear bandits using P40 GPUs, except for those involving GPT-2, which were conducted on V100 GPUs.

For the baselines compared in the experiments, we reimplemented the following methods: *Ensemble+* following the repository https://github.com/google-deepmind/bsuite, *EpiNet* following the repository https://github.com/google-deepmind/neural_testbed.

Additionally, we used the source code from the repository https://github.com/devzhk/LMCTS for LMCTS to obtain the credited results.

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

# APPENDIX

## CONTENTS

## A  ADDITIONAL RELATED WORK

*Thompson Sampling* (TS) (Thompson, 1933) is a classical method renowned for achieving near-optimal regret bounds in sequential decision-making tasks by leveraging Bayesian principles (Russo et al., 2018). TS samples from the posterior distribution over model parameters to balance exploration and exploitation. However, in high-dimensional settings involving complex models like neural networks, maintaining and sampling from exact posterior distributions becomes computationally infeasible due to the intractability of posterior updates (Osband et al., 2019).

To address these computational challenges, approximate Bayesian methods have been developed. *Randomized Least Squares* (RLS) (Osband et al., 2019) approximates the posterior by adding random perturbations to least squares estimates. While effective in simple settings, RLS requires retraining or resampling historical data at each decision step, leading to unbounded computational complexity over time. *Perturbed History Exploration* (PHE) (Kveton et al., 2020a) introduces perturbations to historical data, scaling linearly with decision time horizons, but it is limited to linear bandit settings and does not generalize well to neural network-based environments. *Langevin Monte Carlo Thompson Sampling* (LMC-TS) (Xu et al., 2022) incrementally updates posterior approximations using stochastic gradient Langevin dynamics but incurs growing computational costs due to its gradient update schedule, making it less practical in long-horizon tasks.

Ensemble-based methods approximate posterior distributions by maintaining multiple models, each representing a plausible hypothesis about the environment. *Bootstrapped Thompson Sampling* (Osband et al., 2016) and *Ensemble+* (Osband et al., 2018) support incremental updates and avoid the computational overhead of full retraining required by RLS. Ensemble sampling methods have been applied across various domains, including recommendation systems (Lu et al., 2018; Zhu & Van Roy, 2023), robotics (Lee et al., 2021), and large language models (Dwaracherla et al., 2024). However, they face significant challenges. Maintaining multiple independent models increases computational and memory costs, especially in high-dimensional settings. To reduce computational load, ensemble members often share layers, but this introduces ensemble coupling during training, leading to homogenization of ensemble members, reduced diversity, and ineffective exploration. *EpiNet* (Osband et al., 2023) introduces an MLP that takes a hidden representation and random noise as input to estimate epistemic uncertainty. While innovative, EpiNet lacks theoretical grounding and suffers from large model parameter sizes due to its architecture, discussed in Appendix J. Empirical studies have shown that it demonstrates suboptimal exploration performance in practice (Li et al., 2024). Despite these advances, key challenges remain. Existing methods struggle to maintain bounded computational complexity while performing effective exploration in high-dimensional neural function approximation.

Ensemble++ addresses these challenges by introducing architectural and algorithmic innovations and achieves scalable exploration with theoretical guarantees, closing a long-standing gap in scalable exploration methods.

## B  FURTHER DETAILS ON SEQUENTIAL DEPENDENCY AND ENSEMBLE METHODS

In this appendix, we provide a detailed analysis of how sequential dependency arises in linear function approximation and how ensemble methods mitigate it. We discuss the limitations of naive recursive updates and explain the connections between Randomized Least Squares (RLS), Recursive RLS (RRLS), and ensemble methods, including extreme cases that highlight their differences and similarities. Furthermore, we delve into the phenomenon of ensemble coupling in shared-layer neural network ensembles, examining how it reintroduces sequential dependency, reduces ensemble diversity, and leads to ineffective exploration.

### B.1  RANDOMIZED LEAST SQUARES (RLS)

In sequential decision-making tasks, agents aim to make optimal decisions under uncertainty. *Thompson Sampling* involves sampling from the posterior distribution of unknown parameters to guide action selection. When conjugacy properties are absent, exact posterior updates become intractable due to the lack of closed-form solutions. This limitation motivates approximate methods

for generating posterior samples. The *Randomized Least Squares* (RLS) method, introduced by Osband et al. (2018; 2019), generates approximate posterior samples by solving an optimization problem using perturbed data.

For a function $g_\theta(x)$ parameterized by $\theta$, the RLS method at each time step $t$ involves:

1. **Sampling Independent Perturbations**:

$$\theta_{0,t} \sim \mathcal{N}(\mu_0, \Sigma_0), \tag{6}$$
$$Z_{s,t} \sim \mathcal{N}(0, \sigma^2), \quad s = 1, \ldots, t.$$

2. **Solving the Perturbed Optimization Problem**:

$$\theta_t = \arg\min_\theta \left\{ \sum_{s=1}^{t} (g_\theta(X_s) - Y_s - Z_{s,t})^2 + (\theta - \theta_{0,t})^\top \Sigma_0^{-1} (\theta - \theta_{0,t}) \right\}. \tag{7}$$

**Linear-Gaussian Model Case Study**  Consider a Linear-Gaussian environment model where $\theta^* \sim \mathcal{N}(\mu_0, \Sigma_0)$ and observations are $Y_s = X_s^\top \theta^* + \omega_s^*$, with $\omega_s^* \sim \mathcal{N}(0, \sigma^2)$. The agent uses linear function approximation $g_\theta(x) = x^\top \theta$. Solving equation 7 yields:

$$\theta_t^{\mathrm{RLS}} = \left( \Sigma_0^{-1} + \frac{1}{\sigma^2} \sum_{s=1}^{t} X_s X_s^\top \right)^{-1} \left( \Sigma_0^{-1} \theta_{0,t} + \frac{1}{\sigma^2} \sum_{s=1}^{t} X_s (Y_s + Z_{s,t}) \right) \tag{8}$$

$$= \Sigma_t \left( \Sigma_0^{-1} \theta_{0,t} + \frac{1}{\sigma^2} \sum_{s=1}^{t} X_s (Y_s + Z_{s,t}) \right), \tag{9}$$

where $\Sigma_t = \left( \Sigma_0^{-1} + \frac{1}{\sigma^2} \sum_{s=1}^{t} X_s X_s^\top \right)^{-1}$.

**Posterior Mean and Covariance Matching**  The independent perturbations $\theta_{0,t}$ and $Z_{s,t}$ ensure that both the conditional expectation and covariance of $\theta_t^{\mathrm{RLS}}$ match those of the posterior distribution $\theta^* \mid D_t$. Specifically, the conditional expectation:

$$\mathbb{E}[\theta_t^{\mathrm{RLS}} \mid D_t] = \Sigma_t \left( \Sigma_0^{-1} \mathbb{E}[\theta_{0,t}] + \frac{1}{\sigma^2} \sum_{s=1}^{t} X_s \left( Y_s + \mathbb{E}[Z_{s,t}] \right) \right) \tag{10}$$

$$= \Sigma_t \left( \Sigma_0^{-1} \mu_0 + \frac{1}{\sigma^2} \sum_{s=1}^{t} X_s Y_s \right) = \mathbb{E}[\theta^* \mid D_t].$$

Similarly, the conditional covariance is:

$$\mathrm{Cov}[\theta_t^{\mathrm{RLS}} \mid D_t] = \Sigma_t, \tag{11}$$

matching the posterior covariance of $\theta^* \mid D_t$. The independent perturbations $Z_{s,t}$ from fresh sources of randomness at each time step are crucial for this matching. This result aligns with the findings of Papandreou & Yuille (2010), indicating that the RLS method yields exact posterior samples in the Linear-Gaussian setting.

**Remark 3** (**Lack of Incremental Updates**). *Despite its theoretical appeal, the RLS method requires solving the optimization problem from scratch at each time step, incorporating all past data and fresh perturbations. This is computationally intensive and unsuitable for real-time applications.*

## B.2 LIMITATIONS OF NAIVE RECURSIVE UPDATES

An alternative is to update the model incrementally, as in recursive least squares:

$$\theta_t = \Sigma_t \left( \Sigma_{t-1}^{-1} \theta_{t-1} + \frac{1}{\sigma^2} X_t (Y_t + Z_t) \right), \quad Z_t \sim \mathcal{N}(0, \sigma^2). \tag{12}$$

We refer to the update in equation 12 as *Recursive RLS (RRLS)*.

### B.2.1 SEQUENTIAL DEPENDENCY IN RECURSIVE RLS

However, RRLS suffers from **sequential dependency**: the action $X_t$ depends on $\theta_{t-1}$, which itself depends on previous perturbations $Z_s$ for all $s < t$. This dependency violates the independence condition required for accurate posterior matching in equation 10 and equation 11, leading to biased estimates.

This sequential dependency is illustrated in Figure 5, where the parameter estimate $\theta_t$ depends on all past perturbations and actions.

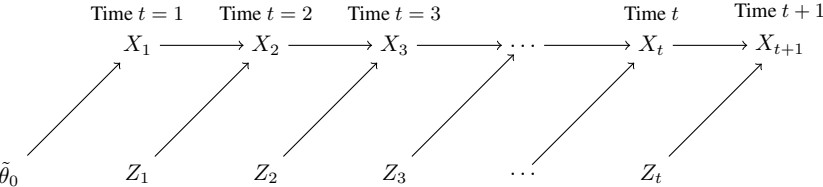

Figure 5: Sequential dependence due to incremental updates and sequential decision-making.

**Impact on Posterior Approximation** Due to sequential dependency, the conditional expectation and covariance of $\theta_t$ no longer match those of the posterior distribution $\theta^* \mid D_t$. This results in biased estimates and ineffective exploration, as the agent's action selection becomes intertwined with the model's parameter updates.

## B.3 ENSEMBLE METHODS TO MITIGATE SEQUENTIAL DEPENDENCY

To address sequential dependency while allowing incremental updates, Osband & Van Roy (2015); Osband et al. (2016); Lu & Van Roy (2017) introduce *Ensemble Sampling* (ES). It initializes $M$ independent models $\theta_{0,m}$ for $m = 1, \ldots, M$, and at each time $t$:

1. **Random Model Selection**: Randomly select a model $m_t$ uniformly from $\{1, \ldots, M\}$ to guide action selection with the selected parameter $\theta_{t-1,m_t}$.

2. **Model Update**: Update all models incrementally:

$$\theta_{t,m} = \Sigma_t \left( \Sigma_{t-1}^{-1} \theta_{t-1,m} + \frac{1}{\sigma^2} X_t (Y_t + Z_{t,m}) \right), \quad Z_{t,m} \sim \mathcal{N}(0, \sigma^2). \tag{13}$$

This approach maintains $M$ independent copies of perturbations $Z_{t,m}$ and introduces additional randomness by uniform sampling at each time step, reducing sequential dependency over time.

### B.3.1 EXTREME CASES CONNECTING RLS, RRLS, AND ENSEMBLE METHODS

Consider the following extreme cases:

- **RLS (Equivalent to Ensemble with $M = T$)**: When $M = T$ and each model is selected exactly once at time $t$ (i.e., $m_t = t$), the ensemble method becomes equivalent to RLS at each time step. This eliminates sequential dependency entirely but requires maintaining $T$ models, which is computationally expensive.

- **RRLS (Equivalent to Ensemble with $M = 1$):** When $M = 1$, Ensemble Sampling reduces to RRLS, which suffers from sequential dependency due to the reuse of the same model for action selection and updates.

These extreme cases highlight the trade-off between computational resources and sequential dependency. By choosing a manageable number of ensemble members $M \ll T$, Ensemble Sampling balances computational efficiency and statistical accuracy.

Table 3: Comparison of methods for addressing sequential dependency.

| Method | Computation per Step | Memory Usage | Sequential Dependency |
|---|---|---|---|
| RLS ($M = T$) | $O(T)$ | High | None |
| ES ($M \ll T$) | $O(M)$ | Moderate | Reduced |
| RRLS ($M = 1$) | $O(1)$ | Low | High |

**Empirical Results** Experiments demonstrate that Ensemble Sampling achieves performance comparable to Thompson Sampling while supporting incremental updates with constant storage and bounded per-step computation. As shown in Figure 6, ES with a manageable number of models $M \ll T$ provides a practical solution for real-time decision-making by balancing computational cost, memory usage, and statistical accuracy.

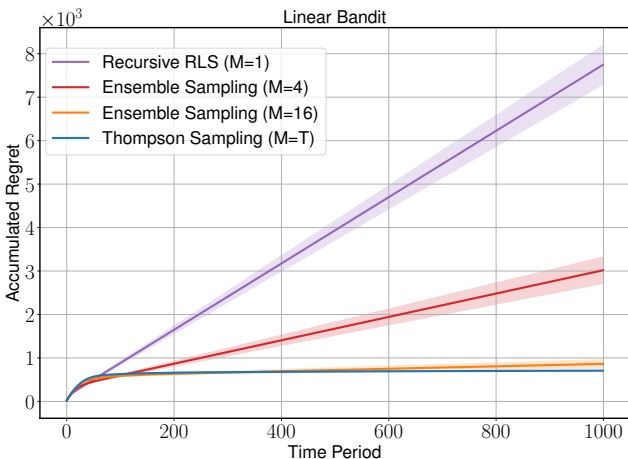

Figure 6: Performance comparison of Ensemble Sampling (ES) with other methods. The bandit setup includes an action size of $|\mathcal{A}| = 10,000$ and an action dimension of $d = 50$.

## B.4 ENSEMBLE COUPLING IN SHARED-LAYER ENSEMBLES AND ITS IMPACT

In neural network implementations of ensemble methods, it is common to share the lower layers (feature extractors) among ensemble members while maintaining separate output layers (heads). While this approach is computationally efficient, it can lead to **ensemble coupling**, where dependencies among ensemble members reduce their ability to explore independently. This subsection examines how ensemble coupling reintroduces sequential dependency, diminishes ensemble diversity, and ultimately leads to ineffective exploration.

### B.4.1 GRADIENT COMPUTATION AND COUPLING

Consider an ensemble of $M$ neural networks, where each network $m$ comprises shared parameters $w$ (the feature extractor) and individual parameters $\theta_m$ (the output head). The prediction of network $m$ on input $x$ is given by:

$$f_{\theta_m}(x) = \langle \theta_m, \tilde{x} \rangle, \quad \text{where} \quad \tilde{x} = \phi(x; w), \tag{14}$$

with $\phi(x; w)$ representing the shared feature mapping.

The ensemble is trained using perturbed observations to encourage diversity. The loss function over data $\{(A_s, Y_s)\}_{s=1}^t$ with perturbations $Z_{s,m}$ is:

$$L(w, \{\theta_m\}) = \sum_{s=1}^{t} \sum_{m=1}^{M} (f_{\theta_m}(A_s) - (Y_s + Z_{s,m}))^2 + \sum_{m=1}^{M} \Psi(\theta_m), \tag{15}$$

where $\Psi(\theta_m)$ is a regularization term for the individual parameters.

The gradient of the loss with respect to the shared parameters $w$ is:

$$\nabla_w L(w) = \sum_{s=1}^{t} \sum_{m=1}^{M} 2 \left( f_{\theta_m}(A_s) - Y_s - Z_{s,m} \right) \nabla_w f_{\theta_m}(A_s) \tag{16}$$

$$= \sum_{s=1}^{t} \sum_{m=1}^{M} 2\delta_{s,m} \nabla_w \langle \theta_m, \tilde{x}_s \rangle$$

$$= \sum_{s=1}^{t} \sum_{m=1}^{M} 2\delta_{s,m} \theta_m^\top \nabla_w \phi(A_s; w),$$

where $\delta_{s,m} = f_{\theta_m}(A_s) - Y_s - Z_{s,m}$ and $\tilde{x}_s = \phi(A_s; w)$.

### B.4.2 Impact on Ensemble Diversity

Updating the shared parameters $w$ using gradients aggregated from all ensemble members introduces several issues:

- **Gradient Interference**: The random perturbations $Z_{s,m}$ lead to ensemble members having different loss landscapes. As a result, their gradients with respect to $w$ may point in conflicting directions, causing destructive interference during updates. This interference hampers the effective learning of shared features that could support diverse hypotheses.

- **Homogenization of Features**: Since all ensemble members share $w$, the feature representations $\tilde{x}_s$ become similar across members. The shared feature extractor tends to capture the common patterns favored by the majority of ensemble members, reducing the uniqueness of individual members' internal representations.

- **Reduced Diversity in Predictions**: With similar features and coupled updates, ensemble members produce correlated predictions. The variance among ensemble outputs decreases, which undermines the ensemble's ability to represent uncertainty about $f^*$.

### B.4.3 Reintroduction of Sequential Dependency

Ensemble coupling through shared parameters $w$ reintroduces **sequential dependency** for the following reasons:

- **Dependence on Entire History**: The shared parameters $w$ are updated using all past data $\{(A_s, Y_s)\}_{s=1}^t$ and perturbations $\{Z_{s,m}\}$. Consequently, $w$ becomes a function of the entire training history and the randomness injected into all ensemble members.

- **Feedback Loop in Action Selection**: At time $t$, action $A_t$ is selected based on the current ensemble predictions, which depend on $w$. Since $w$ encapsulates information from past actions and perturbations, the action selection process becomes entangled with historical data, creating a feedback loop.

- **Violation of Independence Assumptions**: The independence between the agent's intrinsic randomness (e.g., from ensemble perturbations) and the history $\mathcal{H}_t$ is crucial for accurate posterior approximation and uncertainty estimation. Ensemble coupling violates this independence, leading to biased estimates and compromised exploration strategies.

### B.4.4 CONSEQUENCES ON EXPLORATION AND REGRET

The reintroduction of sequential dependency and reduction in ensemble diversity have significant impacts:

- **Ineffective Exploration**: With reduced diversity, ensemble members tend to suggest similar actions. The agent explores a narrower region of the action space, diminishing the likelihood of discovering optimal or near-optimal actions.
- **Suboptimal Uncertainty Estimation**: The lack of independent hypotheses leads to overconfident predictions. The agent underestimates the uncertainty in $f^*$, affecting the balance between exploration and exploitation.
- **Linear Regret Growth**: Ineffective exploration results in the cumulative regret $R(T)$ growing linearly with time $T$. The agent consistently fails to improve its policy by not adequately exploring less-known actions, leading to poor long-term performance.

Empirical studies confirm that shared-layer ensembles exhibit linear regret in problems requiring exploration, such as contextual bandits and reinforcement learning tasks.

### B.5 EMPIRICAL EVIDENCE

To further illustrate the ineffectiveness of shared-layer ensembles, we conducted experiments by varying hyperparameters such as prior scale and the number of updates per step. Despite these adjustments, shared-layer ensembles failed to achieve sublinear regret.

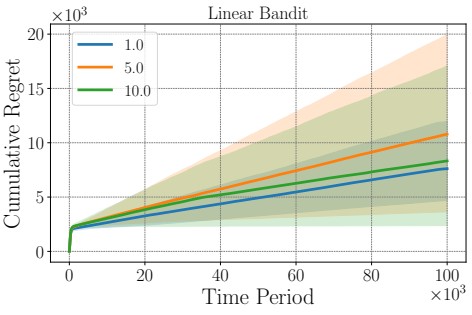
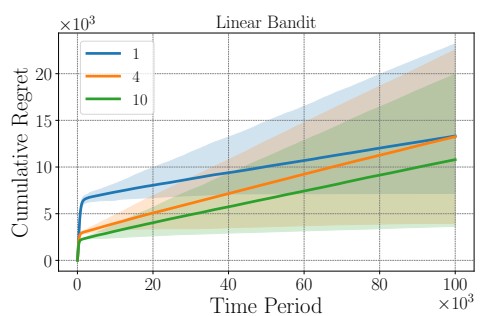

(a) Results with different prior scales (10 updates per step).

(b) Results with different numbers of updates per step (prior scale = 5).

Figure 7: Hyperparameter sweeps for shared-layer ensembles on the Linear Gaussian Model (Russo & Van Roy, 2018).

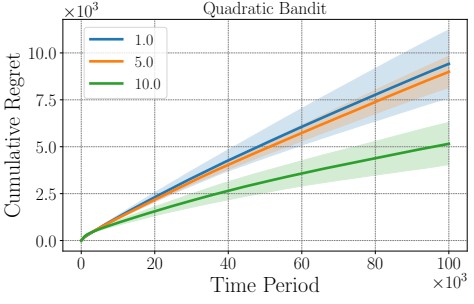
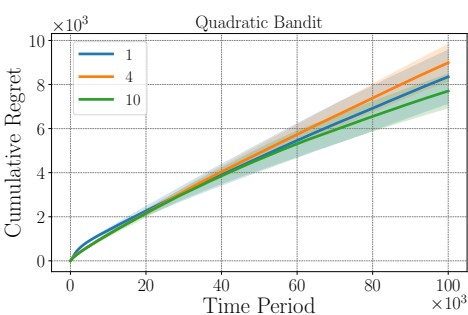

(a) Results with different prior scales (4 updates per step).

(b) Results with different numbers of updates per step (prior scale = 5).

Figure 8: Hyperparameter sweeps for shared-layer ensembles on the Quadratic Bandit problem.

The figures demonstrate that even with different settings, shared-layer ensembles consistently exhibit linear regret, highlighting the fundamental issue caused by ensemble coupling.

### B.5.1 SUMMARY AND IMPLICATIONS

Ensemble coupling in shared-layer neural networks compromises the fundamental advantages of ensemble methods in sequential decision-making:

- It reintroduces sequential dependency by making the shared parameters $w$ a nexus of historical data and randomness.

- It diminishes ensemble diversity, which is critical for representing uncertainty and driving effective exploration.

- These factors collectively lead to ineffective exploration and suboptimal long-term performance, as evidenced by linear regret growth.

## C ENSEMBLE++ ALGORITHM DETAILS

Here we provide detailed derivations and design choices for the Ensemble++ algorithm. Let $x \in \mathcal{X}$ denote the input, and $h(x; w)$ be the shared feature extractor parameterized by $w$. The extracted features are denoted by $\tilde{x} = h(x; w)$. The base network $\psi(\tilde{x}; b)$, parameterized by $b$, estimates the mean prediction based on the shared features. The ensemble components $\{\psi(\text{sg}(\tilde{x}); \theta_m)\}_{m=1}^{M}$, parameterized by $\theta_m$, capture the uncertainty in the prediction. The stop-gradient operator $\text{sg}(\cdot)$ prevents gradients from flowing through $\tilde{x}$ when computing gradients with respect to $\theta_m$, effectively decoupling the ensemble components from the shared layers. At time $t$, the Ensemble++ model predicts:

$$f_\theta^{++}(x, \zeta_t) = \psi(\tilde{x}; b) + \sum_{m=1}^{M} \zeta_{t,m} \psi(\text{sg}(\tilde{x}); \theta_m) + \sum_{m=1}^{M} \zeta_{t,m} \underbrace{\text{sg}(\psi(\tilde{x}; \theta_{0,m}))}_{\textbf{Fixed Prior Ensemble}}, \quad (17)$$

where $\zeta_t = (\zeta_{t,1}, \ldots, \zeta_{t,M})^\top$ is a random vector sampled from an index distribution $P_\zeta$ at time $t$, and $\theta = \{w, b, \theta_1, \ldots, \theta_M, \theta_{0,1}, \ldots, \theta_{0,M}\}$ are the model parameters

The prior ensembles networks are fixed throughout the learning process, incentivizing diverse exploration with prior variations in the initial stage where the data region is under-explored.

### C.1 LOSS FUNCTION DERIVATION

Starting from the loss function with symmetric auxiliary variables:

$$L(\theta; D) = \frac{1}{2M} \sum_{m=1}^{M} \sum_{s=1}^{N} \sum_{\beta \in \{1,-1\}} (Y_s + \beta Z_{s,m} - \psi(\tilde{x}_s; b) - \beta \psi(\text{sg}(\tilde{x}_s); \theta_m))^2 + \Phi(\theta). \quad (18)$$

Expanding the square and summing over $\beta$:

$$\sum_{\beta \in \{1,-1\}} (Y_s + \beta Z_{s,m} - \psi(\tilde{x}_s; b) - \beta \psi(\text{sg}(\tilde{x}_s); \theta_m))^2$$

$$= \sum_{\beta \in \{1,-1\}} ((Y_s - \psi(\tilde{x}_s; b)) + \beta(Z_{s,m} - \psi(\text{sg}(\tilde{x}_s); \theta_m)))^2$$

$$= 2 \left( (Y_s - \psi(\tilde{x}_s; b))^2 + (Z_{s,m} - \psi(\text{sg}(\tilde{x}_s); \theta_m))^2 \right),$$

since the cross terms cancel out due to summing over $\beta \in \{1, -1\}$. This leads to the simplified loss function:

$$L(\theta; D) = \frac{1}{M} \sum_{m=1}^{M} \sum_{s=1}^{N} \left[ \frac{1}{2} \left( Y_s - \psi(\tilde{x}_s; b) \right)^2 + \frac{1}{2} \left( Z_{s,m} - \psi(\mathrm{sg}(\tilde{x}_s); \theta_m) \right)^2 \right] + \Phi(\theta). \tag{19}$$

### C.2 GRADIENT COMPUTATIONS

The gradients with respect to the shared parameters $(w, b)$ are derived solely from the base network loss:

$$\nabla_w L(\theta; D) = \sum_{s=1}^{N} \left( \psi(\tilde{x}_s; b) - Y_s \right) \nabla_{\tilde{x}_s} \psi(\tilde{x}_s; b) \nabla_w h(A_s; w), \tag{20}$$

$$\nabla_b L(\theta; D) = \sum_{s=1}^{N} \left( \psi(\tilde{x}_s; b) - Y_s \right) \nabla_b \psi(\tilde{x}_s; b). \tag{21}$$

The gradients with respect to the ensemble parameters $\theta_m$ are independent of the base network:

$$\nabla_{\theta_m} L(\theta; D) = \sum_{s=1}^{N} \left( \psi(\mathrm{sg}(\tilde{x}_s); \theta_m) - Z_{s,m} \right) \nabla_{\theta_m} \psi(\mathrm{sg}(\tilde{x}_s); \theta_m). \tag{22}$$

Note that due to the stop-gradient operator $\mathrm{sg}(\cdot)$, the ensemble components do not contribute to the gradients of shared parameters.

### C.3 INDEX DISTRIBUTION DESIGN

The choice of index distribution $P_\zeta$ significantly impacts the exploration behavior of Ensemble++. We consider five distribution designs, each offering unique properties for different aspects of the algorithm:

1. **Gaussian Distribution** ($\zeta_t \sim \mathcal{N}(0, I_M)$):
   - Promotes diversity through natural covariance sampling
   - Provides strong theoretical guarantees
2. **Sphere Distribution** ($\zeta_t \sim \sqrt{M} \cdot \mathcal{U}(\mathbb{S}^{M-1})$):
   - Maintains perfect isotropy through rotational invariance
   - Ensures uniform exploration in all directions
   - Controls exploration magnitude with fixed norm
3. **Cube Distribution** ($\zeta_t \sim \mathcal{U}(\{1, -1\}^M)$):
   - Offers discrete exploration with binary choices
   - Provides strong anti-concentration properties
   - Computationally efficient for implementation
4. **Coordinate Distribution** ($\zeta_t \sim \mathcal{U}(\sqrt{M}\{\pm e_1, \ldots, \pm e_M\})$):
   - Enables axis-aligned exploration
   - Minimizes interference between dimensions
   - Particularly useful for feature selection
5. **Sparse Distribution** ($s$-sparse random vectors):
   - Balances exploration and computational efficiency
   - Suitable for high-dimensional problems
   - Adjustable sparsity level for different settings

For detailed theoretical analysis of these distributions' properties (isotropy, concentration, and anti-concentration), see Appendix G.

# D DERIVATION OF THE CLOSED-FORM INCREMENTAL UPDATE

Before proving Proposition 1, we provide useful technical lemmas for isotropic distributions in Definition 9.

**Lemma 2.** *For any isotropic distribution $P$ over $M$-dim vector space, we have for any fixed vector $a \in \mathbb{R}^M$ and $x \in \mathbb{R}^d$, $\mathbb{E}_{X \sim P}[a^\top X x X^\top] = \mathbb{E}_{X \sim P}[X^\top a x X^\top] = x a^\top$.*

*Proof.* The $(i,j)$-th entry of the matrix is

$$[\mathbb{E}_X[X^\top a x X^\top]]_{ij} = \mathbb{E}_X[[X^\top a x X^\top]_{ij}] = \mathbb{E}_X[(\sum_{k=1}^M a_k X_k) x_i X_j] = x_i \sum_{k=1}^M a_k \mathbb{E}_X[X_k X_j] = x_i a_j$$

$\square$

**Lemma 3.** *For any isotropic distribution $P$ over $M$-dim vector space, we have for any fixed matrix $A \in \mathbb{R}^{d \times M}$ and any fixed vector $x \in \mathbb{R}^d$, $\mathbb{E}_{X \sim P}[X^\top A^\top x X^\top] = x^\top A$ and symmetrically, $\mathbb{E}_{X \sim P}[X x^\top A X] = A^\top x$.*

*Proof.* Let $A = (a_1, \ldots, a_M)$ where $a_i \in \mathbb{R}^d$ for $i = 1, 2, \ldots, M$.

$$AX = \sum_{k=1}^M X_i a_i,$$

and

$$(AX)^\top x = \sum_{k=1}^M X_i a_i^\top x.$$

Note $\mathbb{E}_X[X^\top A^\top x X^\top] \in \mathbb{R}^{1 \times M}$. Then, the $j$-th entry of the row vector is

$$[\mathbb{E}_X[X^\top A^\top x X^\top]]_j = \mathbb{E}_X[X_j \sum_{k=1}^M X_i a_i^\top x] = a_j^\top x = x^\top a_j$$

$\square$

**Lemma 4.** *For any isotropic distribution $P$ over $M$-dim vector space, we have for any fixed matrix $B \in \mathbb{R}^{M \times M}$ and any fixed vector $x \in \mathbb{R}^d$, $\mathbb{E}_{X \sim P}[X^\top B X] = \mathrm{tr} B$.*

*Proof.*
$$X^\top A X = \sum_{i,j} X_i X_j B_{ij}$$

By taking the expectation,
$$\mathbb{E}[X^\top A X] = \sum_i B_{ii} = \mathrm{tr} B$$

$\square$

**Remark 4.** *For any distribution $P$ over $M$-dim vector space such that $\mathbb{E}_{X \sim P}[X_i X_j] = \delta_{ij}$, we have for any fixed vector $A \in \mathbb{R}^{d \times M}$,*

$$\mathbb{E}_X[A X X^\top] = A$$

**Definition 1** (Zero-mean). *We say a distribution $P$ over $M$-dim vector space is a zero-mean distribution if $\mathbb{E}_{X \sim P}[X] = 0$.*

### D.1 PROOF OF PROPOSITION 1

In this section, we will derive the closed-form incremental update if the update distribution $P_\xi$ is zero-mean and isotropic. We use short notation $x$ for the feature $\phi(A)$ of some action $A \in \mathcal{A}$. Also, we use short notation $x_t$ for $\phi(A_t)$ for each $t$ respectively.

Let the trainable parameters be $\theta = (\boldsymbol{A}, b)$ and the linear Ensemble++ be

$$f_\theta(x, \xi) = \underbrace{\langle \mathbf{A}\xi + b, x \rangle}_{\textbf{Learnable } f_\theta^L(x,\xi)} + \underbrace{\langle \boldsymbol{\Sigma}_0^{1/2} \mathbf{Z}_0 \xi + \mu_0, x \rangle}_{\textbf{Fixed prior } f^P(x,\xi)}$$

$$= \langle (b + \mu_0), x \rangle + \langle (\boldsymbol{A} + \boldsymbol{\Sigma}_0^{1/2} \mathbf{Z}_0)\xi, x \rangle \tag{23}$$

Let $\Phi(\theta)^2 = \|\boldsymbol{A}\|_F^2 + \|b\|_2^2$ in Equation (2) and $\boldsymbol{\Sigma}_0 = \frac{1}{\lambda}\boldsymbol{I}$.

By writing done the gradient w.r.t. $b$ and taking a look at the stationary point,

$$\frac{\partial L(\theta; D_t)}{\partial b} = 2\mathbb{E}_{\xi \sim P_\xi}[\sum_{s=1}^{t}(\langle (\boldsymbol{A} + \boldsymbol{\Sigma}_0^{1/2}\mathbf{Z}_0)\xi + (b + \mu_0), x_s \rangle - y_s - \sigma \mathbf{z}^\top \xi)x_s] + \boldsymbol{\Sigma}_0^{-1}b$$

$$= \sum_{s=1}^{t} x_s x_s^\top (b + \mu_0) - x_s y_s + \boldsymbol{\Sigma}_0^{-1}b \qquad \text{(if } P_\xi \text{ is zero-mean.)}$$

Let $\tilde{b}_t$ be the stationary point. Definition 1 implies posterior mean matching.

$$\mu_t := (\tilde{b}_t + \mu_0) = (\sum_{s=1}^{t} x_s x_s^\top + \boldsymbol{\Sigma}_0^{-1})^{-1}(\sum_{s=1}^{t} x_t y_t + \boldsymbol{\Sigma}_0^{-1}\mu_0) = \boldsymbol{\Sigma}_t^{-1}(\sum_{s=1}^{t} x_t y_t + \boldsymbol{\Sigma}_0^{-1}\mu_0) \tag{24}$$

By writing done the gradient w.r.t. $\boldsymbol{A}$ and taking a look at the stationary point,

$$\frac{\partial L(\theta; D_t)}{\partial \boldsymbol{A}} = \mathbb{E}_{\xi \sim P_\xi}[\sum_{s=1}^{t}(\langle (\boldsymbol{A} + \boldsymbol{\Sigma}_0^{1/2}\mathbf{Z}_0)\xi + (b + \mu_0), x_s \rangle - y_s - \sigma \mathbf{z}_s^\top \xi)x_s \xi^\top] + 2\boldsymbol{\Sigma}_0^{-1}\boldsymbol{A}$$

$$= \sum_{s=1}^{t}(x_s x_s^\top(\boldsymbol{A} + \boldsymbol{\Sigma}_0^{1/2}\mathbf{Z}_0) - \sigma x_s \mathbf{z}_s^\top) + \boldsymbol{\Sigma}_0^{-1}\boldsymbol{A},$$

where the last equality holds if $P_\xi$ is isotropic and zero-mean and we use Lemmas 2 to 4. Let $\tilde{\boldsymbol{A}}_t$ be the stationary point. We have

$$\boldsymbol{A}_t := \tilde{\boldsymbol{A}}_t + \boldsymbol{\Sigma}_0^{1/2}\mathbf{Z}_0 = \boldsymbol{\Sigma}_t \left( \boldsymbol{\Sigma}_0^{-1/2}\mathbf{Z}_0 + \frac{1}{\sigma}\sum_{s=1}^{t} x_s \mathbf{z}_s^\top \right) \tag{25}$$

From the observation of Equations (24) and (25), the solution $\mu_t$ and $\boldsymbol{A}_t$ can be recursively updated from $\mu_{t-1}$ and $\boldsymbol{A}_{t-1}$ respectively as more data gathering in.

## E TECHNICAL DETAILS FOR INCREMENTAL UNCERTAINTY ESTIMATION IN LEMMA 1

The proof of Lemma 1 addresses the sequential dependencies introduced by incremental updates and adaptive action selection through the following steps:

1. **Variance-aware Discretization:**
   - *Objective:* Reduce the infinite-dimensional problem of bounding $|a^\top \boldsymbol{\Sigma}_t^{-1} \mathbf{A}_t \mathbf{A}_t^\top \boldsymbol{\Sigma}_t^{-1} a - a^\top \boldsymbol{\Sigma}_t^{-1} a| \leq \varepsilon a^\top \boldsymbol{\Sigma}_t^{-1} a$ to a finite set of directions.
   - *Approach:* Construct a discretization $\mathcal{C}_\iota$ of the unit sphere $\mathbb{S}^{d-1}$ such that for any $a \in \mathbb{S}^{d-1}$, there exists a $y \in \mathcal{C}_\iota$ with $\|a - y\|_2 \leq \iota$, where $\iota$ is chosen based on the square root of inverse covariance, i.e., random matrix $\boldsymbol{\Sigma}_t^{-1/2}$, to ensure tight approximation.

2. **Sequential Random Projection:**

- *Objective:* Handle the dependencies arising from adaptive action selection and incremental updates. Establish probabilistic bounds on $|a^\top \boldsymbol{\Sigma}_t^{-1} \mathbf{A}_t \mathbf{A}_t^\top \boldsymbol{\Sigma}_t^{-1} a - a^\top \boldsymbol{\Sigma}_t^{-1} a| \leq \varepsilon a^\top \boldsymbol{\Sigma}_t^{-1} a$ for fixed $a \in \mathbb{S}^{M-1}$ jointly over time steps $t$.
  - *Approach:* Utilize sequential random projection techniques (Li, 2024a), ensuring that the estimated variances remain within the desired bounds with high probability over time, despite the dependencies introduced by the incremental model update.

3. **Union Bound Over Discretized Directions:**
   - *Objective:* Extend the concentration results from the finite set $\mathcal{C}_\iota$ to the entire continuous space $\mathbb{S}^{d-1}$.
   - *Approach:* Apply a union bound over all directions in $\mathcal{C}_\iota$, ensuring that the desired variance bounds hold uniformly across all actions in $\mathcal{A}_t$.

4. **Conversion:** Immediately, we can convert the guarantee about $a^\top \boldsymbol{\Sigma}_t^{-1} \mathbf{A}_t \mathbf{A}_t^\top \boldsymbol{\Sigma}_t^{-1} a$ for all $a \in \mathbb{S}^{M-1}$ to $a^\top \mathbf{A}_t \mathbf{A}_t^\top a$ for all $a \in \mathbb{S}^{M-1}$.

Through these steps, the lemma guarantees that with an appropriately chosen ensemble size $M$, the incremental uncertainty estimates remain accurate, thereby facilitating effective exploration and exploitation.

In this section, we rigorously formalize each of these steps.

First, we state the preliminary tools of sequential random projection for completeness, which is adapted form (Li, 2024a). This tool was used to prove incremental posterior approximation argument of HyperAgent in tabular RL setup (Li et al., 2024). As the tool in (Li, 2024a) works only for the scalar process, we need additional technical innovations to deal with high-dimensional vector process. We make a novel utilization of this tool in the linear function approximation setting for the first time, by a non-trivial discretization argument in Appendix E.2.2 and a reduction to the tool of sequential random projection in Appendix E.2.

### E.1 PROBABILITY TOOLS FOR SEQUENTIAL RANDOM PROJECTION

We define some important concept that would be useful in the analysis. Let $(\Omega, \mathcal{F}, \mathbb{F} = (\mathcal{F}_t)_{t \in \mathbb{N}}, \mathbb{P})$ be a complete filtered probability space. We first consider the measurable properties within the filtered probability space.

**Definition 2** (Adapted process). *For an index set $I$ of the form $\{t \in \mathbb{N} : t \geq t_0\}$ for some $t_0 \in \mathbb{N}$, we say a stochastic process $(X_t)_{t \in I}$ is adapted to the filtration $(\mathcal{F}_t)_{t \in I}$ if each $X_t$ is $\mathcal{F}_t$-measurable.*

**Definition 3** ((Conditionally) $\sigma$-sub-Gaussian). *A random variable $X \in \mathbb{R}$ is $\sigma$-sub-Gaussian if*

$$\mathbb{E}[\exp(\lambda X)] \leq \exp\left(\frac{\lambda^2 \sigma^2}{2}\right), \qquad \forall \lambda \in \mathbb{R}.$$

*Let $(X_t)_{t \geq 1} \subset \mathbb{R}$ be a stochastic process adapted to filtration $(\mathcal{F}_t)_{t \geq 1}$. Let $\sigma = (\sigma_t)_{t \geq 0}$ be a stochastic process adapted to filtration $(\mathcal{F}_t)_{t \geq 0}$. We say the process is $(X_t)_{t \geq 1}$ is conditionally $\sigma$-sub-Gaussian if*

$$\mathbb{E}[\exp(\lambda X_t) \mid \mathcal{F}_{t-1}] \leq \exp\left(\frac{\lambda^2 \sigma_{t-1}^2}{2}\right), \quad a.s. \quad \forall \lambda \in \mathbb{R}.$$

*Specifically for the index $t + 1$, we can say $X_{t+1}$ is ($\mathcal{F}_t$-conditionally) $\sigma_t$-sub-Gaussian. If $\sigma_t$ is a constant $\sigma$ for all $t \geq 0$, then we just say (conditionally) $\sigma$-sub-Gaussian.*

*For a random vector $X \in \mathbb{R}^M$ or vector process $(X_t)_{t \geq 1} \subset \mathbb{R}^M$ in high-dimension, we say it is $\sigma$-sub-Gaussian is for every fixed $v \in \mathbb{S}^{M-1}$ if the random variable $\langle v, X \rangle$, or the scalarized process $(\langle v, X_t \rangle)_{t \geq 1}$ is $\sigma$-sub-Gaussian.*

**Definition 4** (Almost sure unit-norm). *We say a random variable $X$ is almost sure unit-norm if $\|X\|_2 = 1$ almost surely.*

**Remark 5.** *When talking about the perturbation distribution $P_{\mathbf{z}}$, we scale all specific distribution discussed in Appendix G by $\sqrt{\frac{1}{M}}$. Then the spherical distribution $\mathcal{U}(\mathbb{S}^{M-1})$ and uniform over scaled cube $\mathcal{U}(\frac{1}{\sqrt{M}}\{1, -1\}^M)$ satisfy the sub-Gaussian condition in Definition 3 with parameter $\sigma = \frac{1}{\sqrt{M}}$ and also satisfy the unit-norm condition in Definition 4 according to the discussion in Appendix G.*

Additionally, we characterize the boundedness on the stochastic processes.

**Definition 5** (Square-bounded process). *For an index set $I$ of the form $\{t \in \mathbb{N} : t \geq t_0\}$ for some $t_0 \in \mathbb{N}$, the stochastic process $(X_t)_{t \in I}$ is $c$-square-bounded if $X_t^2 \leq c$ almost surely for all $t \in I$.*

Now, we are ready to state the important tool that is fundamental to our analysis.

**Theorem 2** (Sequential random projection in adaptive process (Li, 2024a)). *Let $\varepsilon \in (0, 1)$ be fixed and $(\mathcal{F}_t)_{t \geq 0}$ be a filtration. Let $\mathbf{z}_0 \in \mathbb{R}^M$ be an $\mathcal{F}_0$-measurable random vector satisfies $\mathbb{E}[\|\mathbf{z}_0\|^2] = 1$ and $|\|\mathbf{z}_0\|^2 - 1| \leq (\varepsilon/2)$. Let $(\mathbf{z}_t)_{t \geq 1} \subset \mathbb{R}^M$ be a stochastic process adapted to filtration $(\mathcal{F}_t)_{t \geq 1}$ such that it is $\sqrt{c_0/M}$-sub-Gaussian and each $\mathbf{z}_t$ is unit-norm. Let $(x_t)_{t \geq 1} \subset \mathbb{R}$ be a stochastic process adapted to filtration $(\mathcal{F}_{t-1})_{t \geq 1}$ such that it is $c_x$-square-bounded. Here, $c_0$ and $c_x$ are absolute constants. For any fixed $x_0 \in \mathbb{R}$, if the following condition is satisfied*

$$M \geq \frac{16 c_0 (1 + \varepsilon)}{\varepsilon^2} \left( \log \left( \frac{1}{\delta} \right) + \log \left( 1 + \frac{c_x T}{x_0^2} \right) \right), \tag{26}$$

*we have, with probability at least $1 - \delta$*

$$\forall t \in \{0, 1, \ldots, T\}, \quad (1 - \varepsilon) \left( \sum_{i=0}^{t} x_i^2 \right) \leq \| \sum_{i=0}^{t} x_i \mathbf{z}_i \|^2 \leq (1 + \varepsilon) \left( \sum_{i=0}^{t} x_i^2 \right). \tag{27}$$

### E.2 REDUCE LEMMA 1 TO SEQUENTIAL RANDOM PROJECTION

Without loss of generality, let us consider the set $\mathbb{S}^{d-1}$. First, we define a fine-grained good event for desired approximation error $\varepsilon \in (0, 1)$: the approximate posterior variance $a^\top \mathbf{A}_t \mathbf{A}_t^\top a$ is $\varepsilon$-close to the true posterior variance $a^\top \boldsymbol{\Sigma}_t a$ for direction $a$ at time $t \in \mathcal{T} := \{0, 1, \ldots, T\}$, i.e.,

$$\mathcal{G}_t(a, \varepsilon) = \left\{ |a^\top \mathbf{A}_t \mathbf{A}_t^\top a - a^\top \boldsymbol{\Sigma}_t a| \leq \varepsilon a^\top \boldsymbol{\Sigma}_t a \right\}, \tag{28}$$

and corresponding joint event over the set $\mathbb{S}^{d-1}$,

$$\mathcal{G}_t(\varepsilon) = \bigcap_{a \in \mathbb{S}^{d-1}} \mathcal{G}_t(a, \varepsilon). \tag{29}$$

The good event at time priod $t$ defined in Lemma 1 is indeed $\mathcal{G}_t(1/2)$.

**A reduction.** To fully utilize the probability tool for sequential random projection in Theorem 2, we make use of the following reduction from vector process to scalar process. For a fixed $a \in \mathbb{S}^{d-1}$, we let $\mathbf{s}(a) = a^\top \boldsymbol{\Sigma}_0^{-1/2} \mathbf{Z}_0, s(a)^2 = a^\top \boldsymbol{\Sigma}_0^{-1} a$. Further define short notation $\mathbf{z}_0 := \mathbf{s}(a)^\top / s(a)$ and $x_0 := s(a)$. and $x_t = a^\top \phi(A_t)$ for all $t \in [T]$, then we can relate the incremental update in Proposition 1

$$a^\top \boldsymbol{\Sigma}_t^{-1} \mathbf{A}_t = \underbrace{a^\top \boldsymbol{\Sigma}_0^{-1/2} \mathbf{Z}_0}_{\mathbf{s}(a) = \mathbf{z}_0^\top x_0} + \sum_{i=1}^{t} \underbrace{a^\top \phi(A_i)}_{x_i} \mathbf{z}_i^\top, \quad a^\top \boldsymbol{\Sigma}_t^{-1} a = \underbrace{a^\top \boldsymbol{\Sigma}_0^{-1} a}_{x_0^2} + \sum_{i=1}^{t} \underbrace{a^\top \phi(A_i) \phi(A_i) a}_{x_i^2}$$

to the scalar sequence $(x_t)_{t \geq 0}$ and the vector sequence $(\mathbf{z}_t)_{t \geq 0}$ that would be applied in Theorem 2.

Recall that $\mathcal{H}_t$ the $\sigma$-algebra generated from history $(\mathcal{A}_1, A_1, Y_1, \ldots, \mathcal{A}_{t-1}, A_{t-1}, Y_{t-1}, \mathcal{A}_t)$. Denote $\mathcal{Z}_1 = \sigma(\mathbf{Z}_0)$ and $\mathcal{Z}_t = \sigma(\mathbf{Z}_0, \mathbf{z}_1, \ldots, \mathbf{z}_{t-1})$ for $t \geq 2$. We observe the following statistical relationship, which is further demonstrated in Figure 9

- $\mathbf{z}_t \perp\!\!\!\perp (\mathcal{H}_t, A_t, \mathcal{Z}_t)$, $\mathbf{x}_t$ is dependent on $\mathcal{H}_t, \mathcal{Z}_t$,
- $\mathbf{A}_{t-1} \in \sigma(\mathcal{H}_t, \mathcal{Z}_t)$,
- $\mu_{t-1}, \boldsymbol{\Sigma}_{t-1} \in \mathcal{H}_t$.

For all $t \geq \mathbb{N}$, let us define the sigma-algebra $\mathcal{F}_t = \sigma(\mathcal{H}_{t+1}, \mathcal{Z}_{t+1}, A_{t+1})$. We can verify $\mathcal{F}_k \subseteq \mathcal{F}_l$ for all $k \leq l$. Thus $\mathbb{F} = (\mathcal{F}_t)_{t \in \mathbb{N}}$ is a filtration. Now, we could verify $(\mathbf{z}_t)_{t \geq 0}$ is adapted to $(\mathcal{F}_t)_{t \geq 0}$ and $(x_t)_{t \geq 1}$ is adapted to $(\mathcal{F}_t)_{t \geq 0}$, satisfying the conditions in Theorem 2.

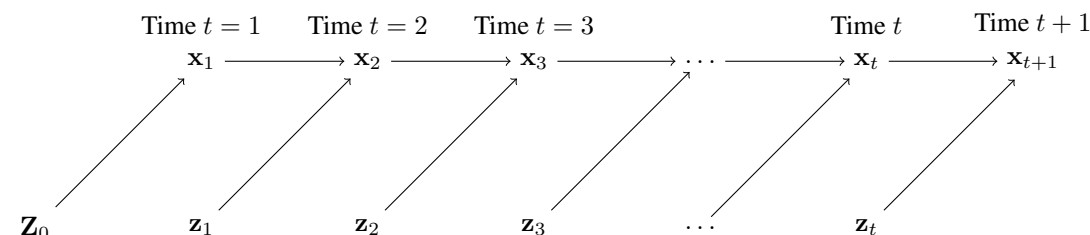

Figure 9: Sequential Dependence Structure in Index Sampling.

### E.2.1 PRIOR APPROXIMATION

First, we state a standard covering argument on sphere.

**Lemma 5** (Covering number of a sphere). *There exists a set $\mathcal{C}_\iota \subset \mathbb{S}^{d-1}$ with $|\mathcal{C}_\iota| \leq (1 + 2/\iota)^d$ such that for all $x \in \mathbb{S}^{d-1}$ there exists a $y \in \mathcal{C}_\iota$ with $\|x - y\|_2 \leq \iota$.*

**Lemma 6** (Computing spectral norm on a covering set). *Let $\boldsymbol{A}$ be a symmetric $d \times d$ matrix, and let $\mathcal{C}_\iota$ be the an $\iota$-covering of $\mathbb{S}^{d-1}$ for some $\iota \in (0, 1)$. Then,*

$$\|\boldsymbol{A}\| = \sup_{x \in \mathbb{S}^{d-1}} |x^\top \boldsymbol{A} x| \leq (1 - 2\iota)^{-1} \sup_{x \in \mathcal{C}_\iota} |x^\top \boldsymbol{A} x|.$$

For compact set $\mathbb{S}^{d-1} = \{x \in \mathbb{R}^d : \|x\| = 1\}$, by standard covering argument in Lemma 6 and the distributional Johnson-Lindenstrauss lemma (Li, 2024b), when

$$M \geq M_1(\varepsilon, \delta) := 256\varepsilon^{-2}(d \log 9 + \log(2/\delta)), \tag{30}$$

the initial good event for prior approximation $G_0(\varepsilon/2)$ holds with probability at least $1 - \delta$.

Next, we are going to show that, under the event $G_0(\varepsilon/2)$, the initial condition on $|\|\mathbf{z}_0\|^2 - 1| \leq (\varepsilon/2)$ in Theorem 2 is satisfied. That is, under the event $G_0(\varepsilon/2)$

$$(1 - \varepsilon/2)a^\top \boldsymbol{\Sigma}_0 a \leq \|a^\top \boldsymbol{\Sigma}_0^{1/2} \mathbf{Z}_0\|^2 \leq (1 + \varepsilon/2)a^\top \boldsymbol{\Sigma}_0 a, \quad \forall a \in \mathbb{S}^{d-1}$$

$$\Leftrightarrow \quad \|\mathbf{Z}_0 \mathbf{Z}_0^\top - \boldsymbol{I}\| \leq \varepsilon/2$$

$$\Leftrightarrow \quad (1 - \varepsilon/2)a^\top \boldsymbol{\Sigma}_0^{-1} a \leq \|a^\top \boldsymbol{\Sigma}_0^{-1/2} \mathbf{Z}_0\|^2 \leq (1 + \varepsilon/2)a^\top \boldsymbol{\Sigma}_0^{-1} a, \quad \forall a \in \mathbb{S}^{d-1}. \tag{31}$$

Recall the short notation $\mathbf{s}(a) = a^\top \boldsymbol{\Sigma}_0^{-1/2} \mathbf{Z}_0$ and $s(a)^2 = a^\top \boldsymbol{\Sigma}_0^{-1} a$, we have $\mathbf{z}_0 = \mathbf{s}(a)^\top / s(a)$ satisfying $|\|\mathbf{z}_0\|^2 - 1| \leq (\varepsilon/2)$ according to Equation (31).

### E.2.2 POSTERIOR APPROXIMATION

Notice that $x_0^2 = a^\top \boldsymbol{\Sigma}_0 a \geq \inf_{a \in \mathbb{S}^{d-1}} a^\top \boldsymbol{\Sigma}_0^{-1} a = s_{\min}^2$. As by the definition of the feature map $\phi(\cdot)$ in assumption 1, we can examine that $x_t^2 = (a^\top \phi(A_t))^2 \leq 1$ for $t \geq 1$. That is, the sequence $(a^\top \phi(A_t))_{t \geq 1}$ is 1-square-bounded for any $a \in \mathbb{S}^{d-1}$.

We could also check that $(\mathbf{z}_t)_{t \geq 1}$ is $1/\sqrt{M}$-sub-Gaussian and with unit-norm when the perturbation distribution $P_z$ is Cube $\mathcal{U}(\{1, -1\}^M)$ or Sphere $\mathcal{U}(\mathbb{S}^{M-1})$.

Under the prior approximation event $\mathcal{G}_0(\varepsilon/2)$, we apply Theorem 2 to show that for any fixed $a \in \mathbb{S}^{d-1}$,

$$\forall t \in \mathcal{T}, E_t(a, \varepsilon) := \left\{ |a^\top \boldsymbol{\Sigma}_t^{-1} \mathbf{A}_t \mathbf{A}_t^\top \boldsymbol{\Sigma}_t^{-1} a - a^\top \boldsymbol{\Sigma}_t^{-1} a| \leq \varepsilon a^\top \boldsymbol{\Sigma}_t^{-1} a \right\} \tag{32}$$

holds with probability at least $1 - \delta$ when

$$M \geq \frac{16(1 + \varepsilon)}{\varepsilon^2} \left( \log\left(\frac{1}{\delta}\right) + \log\left(1 + \frac{T}{s_{\min}^2}\right) \right). \tag{33}$$

We need discretization (covering) argument to relate the result in Equation (32) to the desired good event defined in Equation (29)

$$\mathcal{G}_t(\varepsilon) = \left\{ \|\boldsymbol{\Sigma}_t^{-1/2} \mathbf{A}_t \mathbf{A}_t^\top \boldsymbol{\Sigma}_t^{-1/2} - \boldsymbol{I}\| \leq \varepsilon \right\}.$$

**Standard discretization produces unacceptable results.** Utilizing standard discretization for computing spectral norm in Lemma 6, let $\iota = 1/4$, we can show that

$$\bigcap_{a \in \mathcal{C}_{1/4}} E_t(a, \varepsilon/2T) \subseteq \mathcal{G}_t(\varepsilon).$$

This is due to,

$$
\|\boldsymbol{\Sigma}_t^{-1/2} \mathbf{A}_t \mathbf{A}_t^\top \boldsymbol{\Sigma}_t^{-1/2} - \boldsymbol{I}\| = \sup_{x \in \mathbb{S}^{d-1}} \frac{|x^\top (\boldsymbol{\Sigma}_t^{-1} \mathbf{A}_t \mathbf{A}_t^\top \boldsymbol{\Sigma}_t^{-1} - \boldsymbol{\Sigma}_t^{-1}) x|}{x^\top \boldsymbol{\Sigma}_t^{-1} x}
$$

$$
\leq \frac{2}{\lambda_{\min}(\boldsymbol{\Sigma}_t^{-1})} \sup_{a \in \mathcal{C}_{1/4}} |a^\top (\boldsymbol{\Sigma}_t^{-1} \mathbf{A}_t \mathbf{A}_t^\top \boldsymbol{\Sigma}_t^{-1} - \boldsymbol{\Sigma}_t^{-1}) a|
$$

$$
\leq 2\varepsilon' \frac{\sup_{a \in \mathcal{C}_{1/4}} a^\top \boldsymbol{\Sigma}_t^{-1} a}{\lambda_{\min}(\boldsymbol{\Sigma}_t^{-1})} \leq 2\varepsilon' \cdot \kappa(\boldsymbol{\Sigma}_t^{-1}) \leq 2T\varepsilon'.
$$

Then by union bound over $\mathcal{C}_{1/4}$, plugging in $\varepsilon/2T$ to Equation (33), we require $M \geq \tilde{O}(dT^2 \log T)$ to let $\bigcap_{a \in \mathcal{C}_{1/4}} E_t(a, \varepsilon/2T)$ hold with probability at least $1 - \delta$. This result is not acceptable as the per-step computation complexity is growing unbounded polynimally with the interaction steps $T$. In the next section, we provide a non-trivial discretization to resolve this analytical problem.

**Variance-aware discretization.** The key contribution here is that we choose a variance weighted norm to measure discretization error. This variance-awareness, together with specific choice on a $O(1/\sqrt{T})$-discretization error and a constant approximation error $\varepsilon$, eventually arrives at $O(d \log T)$ log covering number and $M = \tilde{O}(d \log T)$ in Lemma 1.

Let $\mathbf{S}_t = \boldsymbol{\Sigma}_t^{-1} \mathbf{A}_t = \mathbf{X}_t^\top \mathbf{Z}_t$ and $\boldsymbol{\Gamma}_t = \boldsymbol{\Sigma}_t^{1/2} \mathbf{S}_t = \boldsymbol{\Sigma}_t^{-1/2} \mathbf{A}_t$. Notice that, from Equation (32), the event holds with probability at least $1 - \delta'$

$$\forall t \in \mathcal{T}, E_t(a, \varepsilon') = \left\{ \frac{|a^\top \mathbf{S}_t \mathbf{S}_t^\top a - a^\top \boldsymbol{\Sigma}_t^{-1} a|}{a^\top \boldsymbol{\Sigma}_t^{-1} a} \leq \varepsilon' \right\}$$

when

$$M \geq \frac{16(1 + \varepsilon')}{(\varepsilon')^2} \left( \log\left(\frac{1}{\delta'}\right) + \log\left(1 + \frac{T}{s_{\min}^2}\right) \right).$$

Let $\mathcal{C}_\iota \subset \mathbb{S}^{d-1}$ be the $\iota$-covering set in Lemma 5 and the event $\bigcap_{a \in \mathcal{C}_\iota} E_t(a, \varepsilon')$ holds. Let $x \in \mathbb{S}^{d-1}$ and $y \in \mathcal{C}_\iota$ such that $\|x - y\| \leq \iota$. Define short notation $u = \boldsymbol{\Sigma}_t^{-1/2} x$, $v = \boldsymbol{\Sigma}_t^{-1/2} y$.

$$
\frac{|x^\top \mathbf{S}_t \mathbf{S}_t^\top x - x^\top \boldsymbol{\Sigma}_t^{-1} x|}{x^\top \boldsymbol{\Sigma}_t^{-1} x} - \frac{|y^\top \mathbf{S}_t \mathbf{S}_t^\top y - y^\top \boldsymbol{\Sigma}_t^{-1} y|}{y^\top \boldsymbol{\Sigma}_t^{-1} y}
$$

$$
= \frac{|u^\top \boldsymbol{\Gamma}_t \boldsymbol{\Gamma}_t^\top u - u^\top u|}{u^\top u} - \frac{|v^\top \boldsymbol{\Gamma}_t \boldsymbol{\Gamma}_t^\top v - v^\top v|}{v^\top v} = \frac{|\|\boldsymbol{\Gamma}_t u\|^2 - \|u\|^2|}{\|u\|^2} - \frac{|\|\boldsymbol{\Gamma}_t v\|^2 - \|v\|^2|}{\|v\|^2}
$$

$$
\leq \left| \frac{\|\boldsymbol{\Gamma}_t u\|^2}{\|u\|^2} - \frac{\|\boldsymbol{\Gamma}_t v\|^2}{\|v\|^2} \right| = \underbrace{\left| \frac{\|\boldsymbol{\Gamma}_t u\|^2 - \|\boldsymbol{\Gamma}_t v\|^2}{\|u\|^2} \right|}_{(I)} + \underbrace{\|\boldsymbol{\Gamma}_t v\|^2 \left| \frac{1}{\|u\|^2} - \frac{1}{\|v\|^2} \right|}_{(II)}.
$$

We bound $(I)$ and $(II)$ separately. W.L.O.G, assume $\|u\| \geq \|v\|$. Recall $s_{\max}^2 \geq a^\top \boldsymbol{\Sigma}_0^{-1} a \geq s_{\min}^2$ for all $a \in \mathbb{S}^{d-1}$. Since $\|u\| = x^\top \boldsymbol{\Sigma}_t^{-1} x = x^\top (\boldsymbol{\Sigma}_0^{-1} + \sum_{s=1}^t \mathbf{x}_s \mathbf{x}_s^\top) x$, we have $s_{\min}^2 \leq \|u\| \leq s_{\max}^2 + t$. For $(I)$, we have

$$
(I) \leq \frac{(\|\boldsymbol{\Gamma}_t u\| - \|\boldsymbol{\Gamma}_t v\|)(\|\boldsymbol{\Gamma}_t u\| + \|\boldsymbol{\Gamma}_t v\|)}{\|u\|^2} \leq \frac{\|\boldsymbol{\Gamma}_t (u - v)\|}{s_{\min}} \left( \frac{\|\boldsymbol{\Gamma}_t u\|}{\|u\|} + \frac{\|\boldsymbol{\Gamma}_t v\|}{\|v\|} \right)
$$

$$
\leq \frac{\|\boldsymbol{\Gamma}_t\| \|u - v\|}{s_{\min}} (2\|\boldsymbol{\Gamma}_t\|) \leq \frac{2\|\boldsymbol{\Gamma}_t\|^2 \|\boldsymbol{\Sigma}_t^{-1/2}\| \iota}{s_{\min}} \leq \frac{2\|\boldsymbol{\Gamma}_t\|^2 \iota \sqrt{s_{\max}^2 + t}}{s_{\min}}.
$$

For $(II)$, we have

$$(II) \leq \frac{\|\boldsymbol{\Gamma}_t v\|^2}{\|v\|^2} \frac{\|u\|^2 - \|v\|^2}{\|u\|^2} \leq \|\boldsymbol{\Gamma}_t\|^2 \frac{\|u\|^2 - \|v\|^2}{\|u\|^2} \leq \|\boldsymbol{\Gamma}_t\|^2 \frac{(\|u\| - \|v\|)(\|u\| + \|v\|)}{\|u\|^2}$$

$$\leq \frac{2\|\boldsymbol{\Gamma}_t\|^2 \|u - v\|}{s_{\min}} \leq \frac{2\|\boldsymbol{\Gamma}_t\|^2 \|\boldsymbol{\Sigma}_t^{-1/2}\|\iota}{s_{\min}} \leq \frac{2\|\boldsymbol{\Gamma}_t\|^2 \iota \sqrt{s_{\max}^2 + t}}{s_{\min}}.$$

Then, putting $(I)$ and $(II)$ together, by the variance-aware discretization argument, we have the spectral norm

$$\|\boldsymbol{\Sigma}_t^{-1/2} \mathbf{A}_t \mathbf{A}_t^\top \boldsymbol{\Sigma}_t^{-1/2} - \boldsymbol{I}\| = \sup_{x \in \mathbb{S}^{d-1}} \frac{|x^\top (\boldsymbol{\Sigma}_t^{-1} \mathbf{A}_t \mathbf{A}_t^\top \boldsymbol{\Sigma}_t^{-1} - \boldsymbol{\Sigma}_t^{-1}) x|}{x^\top \boldsymbol{\Sigma}_t^{-1} x}$$

$$\leq \frac{4\|\boldsymbol{\Gamma}_t\|^2 \iota \sqrt{s_{\max}^2 + t}}{s_{\min}} + \sup_{y \in \mathcal{C}_\iota} \frac{|y^\top (\boldsymbol{\Sigma}_t^{-1} \mathbf{A}_t \mathbf{A}_t^\top \boldsymbol{\Sigma}_t^{-1} - \boldsymbol{\Sigma}_t^{-1}) y|}{y^\top \boldsymbol{\Sigma}_t^{-1} y}$$

$$\leq \frac{4\|\boldsymbol{\Gamma}_t\|^2 \iota \sqrt{s_{\max}^2 + t}}{s_{\min}} + \varepsilon'. \tag{34}$$

Let

$$\iota = \frac{\alpha s_{\min}}{4\sqrt{s_{\max}^2 + T}},$$

where $\alpha$ to be determined. Equivalent formulation of the norm is $\|\boldsymbol{\Gamma}_t\|^2 = \lambda_{\max}(\boldsymbol{\Gamma}_t \boldsymbol{\Gamma}_t^\top)$ and

$$\|\boldsymbol{\Sigma}_t^{-1/2} \mathbf{A}_t \mathbf{A}_t^\top \boldsymbol{\Sigma}_t^{-1/2} - \boldsymbol{I}\| = \max\{\lambda_{\max}(\boldsymbol{\Gamma}_t \boldsymbol{\Gamma}_t^\top) - 1, 1 - \lambda_{\min}(\boldsymbol{\Gamma}_t \boldsymbol{\Gamma}_t^\top)\}.$$

Thus, we derive from Equation (34),

$$\lambda_{\max}(\boldsymbol{\Gamma}_t \boldsymbol{\Gamma}_t^\top) \leq \frac{1 + \varepsilon'}{1 - \alpha}, \quad \lambda_{\min}(\boldsymbol{\Gamma}_t \boldsymbol{\Gamma}_t^\top) \geq 1 - \varepsilon' - \alpha \lambda_{\max}(\boldsymbol{\Gamma}_t \boldsymbol{\Gamma}_t^\top) \geq 1 - \varepsilon' - \frac{\alpha(1 + \varepsilon')}{1 - \alpha}.$$

**Claim 1.** *If $\frac{1+\varepsilon'}{1-\alpha} = 1 + \varepsilon$ and $\varepsilon' + \frac{\alpha(1+\varepsilon')}{1-\alpha} = \varepsilon$, then*

$$1 - \varepsilon \leq \lambda_{\min}(\boldsymbol{\Gamma}_t \boldsymbol{\Gamma}_t^\top) \leq \lambda_{\max}(\boldsymbol{\Gamma}_t \boldsymbol{\Gamma}_t^\top) \leq 1 + \varepsilon.$$

Let $\varepsilon = 1/2$, then $(\varepsilon', \alpha) = (1/4, 1/6)$ suffices for the Claim 1. That is to say the following configuration for discretization error $\iota$ suffices,

$$\iota = \frac{s_{\min}}{24\sqrt{s_{\max}^2 + T}}.$$

The covering number is $|\mathcal{C}_\iota| \leq (1 + 2/\iota)^d \leq (1 + (48/s_{\min})\sqrt{s_{\max}^2 + T})^d$. By union bound and define $\delta' = \delta/(1 + (48/s_{\min})\sqrt{s_{\max}^2 + T})^d$, we have

$$\mathbb{P}\left(\bigcap_{t \in \mathcal{T}} \mathcal{G}_t(1/2) \mid \mathcal{G}_0(1/4)\right) \geq 1 - \delta,$$

when

$$M \geq M_2(\delta) := \frac{16(5/4)}{(1/4)^2}\left(d\log\left(\frac{1 + (48/s_{\min})\sqrt{s_{\max}^2 + T}}{\delta}\right) + \log\left(1 + \frac{T}{s_{\min}^2}\right)\right).$$

Here the constant is 320.

**Put things together.** When $M \geq M_3 := \max\{M_1(1/2, \delta/2), M_2(\delta/2)\}$, we have

$$\mathbb{P}\left(\bigcap_{t \in \mathcal{T}} \mathcal{G}_t(1/2)\right) = \mathbb{P}\left(\bigcap_{t \in \mathcal{T}} \mathcal{G}_t(1/2) \mid \mathcal{G}_0(1/4)\right) \mathbb{P}\left(\mathcal{G}_0(1/4)\right) \geq (1 - \delta/2)^2 \geq 1 - \delta.$$

With some calculations, we derive

$$M_1(1/2, \delta/2) = 1024(d\log 9 + \log(4/\delta)),$$

and

$$M_2(\delta/2) = 320\left(d\log\left(\frac{2 + (96/s_{\min})\sqrt{s_{\max}^2 + T}}{\delta}\right) + \log\left(1 + \frac{T}{s_{\min}^2}\right)\right).$$

Since the total time periods $T$ is the dominant growing term, there exist a constant $T_0$ such that $M_3 = M_2(\delta/2)$ when $T > T_0$.

## F   TECHNICAL DETAILS IN REGRET ANALYSIS

### F.1   GENERAL REGRET BOUND

We start by providing a general analytical framework for agent, potentially randomized, operating in the generic bandit environments. Let us introduce a few necessary definitions to facilitate the understanding and analysis. The confidence bound is used for uncertainty estimation over the ture function $f^*$ given the history $\mathcal{H}_t$.

**Definition 6** (Confidence bounds). *Confidence bounds are a sequence of real-valued $\mathcal{H}_t$-measurable functions $L_t(\cdot)$ and $U_t(\cdot)$ for $t \in [T]$ such that, w.p. at least $1 - \delta$, the joint event $\mathcal{E} = \cap_{t \in [T]} \mathcal{E}_t$ holds, where $\mathcal{E}_t := \{f^*(a) \in [L_t(a), U_t(a)], \forall a \in \mathcal{A}_t\}$.*

The agent may not perform well unless it is well-behaved, defined by *reasonableness* and *optimism*. Intuitively, an agent that explores too much or too little will incur a high regret. Reasonableness and optimism are the mechanisms for controlling these potential flaws respectively.

**Definition 7** (Reasonableness). *Given confidence bounds $L_t(\cdot)$ and $U_t(\cdot)$ for $t \in [T]$, an (randomized) agent is called reasonable if it produces a sequence of functions $(\tilde{f}_t(\cdot), t \in [T])$ such that w.p. at least $1 - \delta$, the joint event $\tilde{\mathcal{E}} = \cap_{t \in [T]} \tilde{\mathcal{E}}_t$ holds, where $\tilde{\mathcal{E}}_t := \{\tilde{f}_t(a) \in [L_t(a), U_t(a)], \forall a \in \mathcal{A}_t\}$.*

In short, *reasonableness* ensures that the chosen action according to $\tilde{f}_t$ is close to the best action which ensures agent does not explore actions unnecessarily. The following *optimism* guarantees the agent sufficient explores.

**Definition 8** (p-optimism). *Let p be a sequence of positive real number $(p_t, t \in [T])$. We say an (randomized) agent is p-optimistic when it produces a sequence of functions $(\tilde{f}_t(\cdot), t \in [T])$ such that for all $t \in [T]$, $\tilde{f}_t(\cdot)$ is $p_t$-optimistic, i.e., $\mathbb{P}(\max_{a \in \mathcal{A}_t} \tilde{f}_t(a) \geq \max_{a \in \mathcal{A}_t} f^*(a) \mid \mathcal{H}_t) \geq p_t$.*

The generic agent satisfying the conditions on *reasonableness* and *optimism* has desired behavior.

Building upon the definitions of Reasonableness and Optimism, we establish a general regret bound applicable to any agent satisfying these conditions.

**Theorem 3** (General Regret Bound). *Given confidence bounds as defined in Definition 6, and assuming the agent is both reasonable and p-optimistic, the cumulative regret over $T$ time steps satisfies*

$$R(T) \leq \sum_{t=1}^{T} \frac{1}{p_t} \mathbb{E}\left[U_t(A_t) - L_t(A_t) \mid \mathcal{H}_t\right] + \sum_{t=1}^{T} \left(U_t(A_t) - L_t(A_t)\right), \quad (35)$$

*with probability at least $1 - \delta$.*

**Interpretation**   The regret bound in equation 35 decomposes into two main components:

1. **Exploration-Exploitation Trade-off:** The first term scales with $\frac{1}{p_t}$ and the expected width of the confidence bounds. A higher $p_t$ (i.e., greater optimism) reduces this component, promoting exploration.

2. **Confidence Bound Widths:** The second term aggregates the widths of the confidence intervals across all time steps, reflecting the uncertainty inherent in the agent's estimates.

For the regret to be sublinear in $T$, it is essential that the confidence bounds $U_t(a) - L_t(a)$ shrink appropriately as $t$ increases, ensuring that both terms grow slower than linearly with $T$.

*Proof.* Let $A_t = \max_{a \in \mathcal{A}_t} \tilde{f}_t(a)$ and $A_t^* = \max_{a \in \mathcal{A}_t} f^*(a)$. Let $B_t = \max_{a \in \mathcal{A}_t} L_t(a)$, which is $\mathcal{H}_t$-measurable. Conditioned on the event $\mathcal{E} \cap \tilde{\mathcal{E}}$, both $f^*(A_t^*) \geq B_t$ and $\tilde{f}_t(A_t) \geq B_t$ hold. By p-optimism and the fact $(f^*(A_t^*) - B_t)$ is $\mathcal{H}_t$-measurable and positive,

$$p_t \leq \mathbb{P}(f_t(A_t) - B_t \geq f^*(A_t^*) - B_t \mid \mathcal{H}_t) \overset{(*)}{\leq} \mathbb{E}[f_t(A_t) - B_t \mid \mathcal{H}_t] / (f^*(A_t^*) - B_t),$$

where $(*)$ is due to Markov inequality. Rearranging and using the additional fact $B_t \geq L_t(A_t)$ yield

$$f^*(A_t^*) - \tilde{f}_t(A_t) \leq f^*(A_t^*) - B_t \leq \frac{1}{p_t}\mathbb{E}[f_t(A_t) - B_t \mid \mathcal{H}_t] \leq \frac{1}{p_t}\mathbb{E}[U_t(A_t) - L_t(A_t) \mid \mathcal{H}_t]. \tag{36}$$

By the reasonableness, $\tilde{f}_t(A_t) \leq U_t(A_t)$. Then, from the definition of confidence bounds

$$\tilde{f}_t(A_t) - f^*(A_t) \leq U_t(A_t) - L_t(A_t) \tag{37}$$

Putting Equations (36) and (37) together and then summing over the time index $t$ yields the general regret upper bound. $\qquad\square$

### F.2 PROOF OF THEOREM 1 FOR LINEAR CONTEXTUAL BANDITS

To make the proof easy to access, we restate the core results and a few notations that is needed for the proof of the propositions.

| $P_\zeta$ | Gaussian $N(0, I_M)$ | Sphere $\sqrt{M}\mathcal{U}(\mathbb{S}^{M-1})$ | Cube $\mathcal{U}(\{1, -1\}^M)$ | Coord $\mathcal{U}(\{\pm e_i\}_{i\in[M]})$ | Sparse |
|---|---|---|---|---|---|
| $\rho(P_\zeta)$ | $\rho_1 \wedge \rho_3$ | $\rho_2 \wedge \rho_3$ | $\rho_2 \wedge \rho_3$ | $\rho_2$ | $\rho_2$ |
| $p(P_\zeta)$ | $\frac{1}{4\sqrt{e\pi}}$ | $\frac{1}{2} - \frac{e^{1/12}}{\sqrt{2\pi}}$ | $7/32$ | $\frac{1}{2M}$ | N/A |

Table 4: (Restate of Table Table 1) The coefficient $\rho(P_\zeta)$ and $p(P_\zeta)$ related to reasonableness and optimism condition.

Adapting the results from (Abbasi-Yadkori et al., 2011b; Abeille & Lazaric, 2017), let $\beta_t = \sqrt{\lambda} + \sqrt{2\log(1/\delta) + \log\det(\boldsymbol{\Sigma}_{t-1}^{-1}/\lambda^d)}$. Under assumption 1, we define the confidence bound as

$$L_t(\cdot) = (-1) \vee (\langle\mu_{t-1}, \phi(\cdot)\rangle - \beta_t\|\phi(\cdot)\|_{\boldsymbol{\Sigma}_{t-1}}), U_t(\cdot) = 1 \wedge (\langle\mu_{t-1}, \phi(\cdot)\rangle + \beta_t\|\phi(\cdot)\|_{\boldsymbol{\Sigma}_{t-1}})$$

For the purpose of analysis within various reference distribution, we define a slightly inflated confidence bounds as

$$L_t(\cdot; P_\zeta) = (\langle\mu_{t-1}, \phi(\cdot)\rangle - \beta_t\rho(P_\zeta)\|\phi(\cdot)\|_{\boldsymbol{\Sigma}_{t-1}}) \vee (-1),$$
$$U_t(\cdot; P_\zeta) = (\langle\mu_{t-1}, \phi(\cdot)\rangle + \beta_t\rho(P_\zeta)\|\phi(\cdot)\|_{\boldsymbol{\Sigma}_{t-1}}) \wedge 1.$$

$\rho(P_\zeta)$ is defined via $\rho_1 = O(\sqrt{M\log(M/\delta)})$, $\rho_2 = O(\sqrt{M})$, and $\rho_3 = O(\sqrt{\log(|\mathcal{A}|/\delta)})$ and Table 1. An immediate observation is that $[L_t(\cdot), U_t(\cdot)] \subset [L_t(\cdot; P_\zeta), U_t(\cdot; P_\zeta)]$. Thus, $L_t(\cdot; P_\zeta)$ and $U_t(\cdot; P_\zeta)]$ are also confidence bounds. We consider the the following functional form for Ensemble++ under linear setup: for time $t$,

$$\tilde{f}_t(a) := f_{\theta_t}(a, \zeta_t) = \langle\phi(a), \beta_t\mathbf{A}_{t-1}\zeta_t + \mu_{t-1}\rangle, \quad \forall a \in \mathcal{A}, \tag{38}$$

where the parameters include $\theta_t = (\mathbf{A}_t, \mu_t)$.

The condition on the propositions and theorem for regret analysis is when Equation (26) is satisfied, that is when $M = \Theta(d\log T)$, the Lemma 1 implies that with high probability, the good events $\mathcal{G} = \bigcap_{t=0}^T \mathcal{G}_t$ hold jointly, where

$$\mathcal{G}_t := \left\{\frac{1}{2}x^\top\boldsymbol{\Sigma}_t x \leq x^\top\mathbf{A}_t\mathbf{A}_t^\top x \leq \frac{3}{2}x^\top\boldsymbol{\Sigma}_t x, \quad \forall x \in \mathbb{R}^d\right\}.$$

In the following section, we discuss the proof conditioned on the joint event $\mathcal{G}$ and also the confidence event that $f^*(a) \in [L_t(a), U_t(a)]$ for all $t \in [T]$ and $a \in \mathcal{A}$.

### F.2.1 PROOF OF PROPOSITION 2

Notice that from Equation (38), we derive

$$|\tilde{f}_t(a) - \langle \mu_{t-1}, \phi(a) \rangle| = |\langle \phi(a), \beta_t \mathbf{A}_{t-1}\zeta_t \rangle|$$

$$= \beta_t \sqrt{\phi(a)^\top \mathbf{A}_{t-1}\mathbf{A}_{t-1}^\top \phi(a)} \left| \left\langle \frac{\phi(a)^\top \mathbf{A}_{t-1}}{\|\phi(a)^\top \mathbf{A}_{t-1}\|}, \zeta_t \right\rangle \right|$$

$$\leq (3/2)\beta_t \sqrt{\phi(a)^\top \boldsymbol{\Sigma}_{t-1}\phi(a)} \left| \left\langle \frac{\phi(a)^\top \mathbf{A}_{t-1}}{\|\phi(a)^\top \mathbf{A}_{t-1}\|}, \zeta_t \right\rangle \right|,$$

where the last inequality is due to the good event $\mathcal{G}$. For compact action set, we use Cauchy–Schwarz inequality,

$$\left| \left\langle \frac{\phi(a)^\top \mathbf{A}_{t-1}}{\|\phi(a)^\top \mathbf{A}_{t-1}\|}, \zeta_t \right\rangle \right| \leq \|\zeta_t\|.$$

Using the concentration properties of $P_\zeta$ in Appendix G to upper bound $\|\zeta_t\|$ yields part of the results. For finite action set $\mathcal{A}$, also taking the advantages of the concentration properties of several reference distributions $P_\zeta$ in Appendix G to bound the conditionally probability

$$\mathbb{P}\left( \left| \left\langle \frac{\phi(a)^\top \mathbf{A}_{t-1}}{\|\phi(a)^\top \mathbf{A}_{t-1}\|}, \zeta_t \right\rangle \right| \leq \sqrt{\log \frac{2|\mathcal{A}|}{\delta}} \mid \mathcal{H}_t, \mathcal{Z}_t \right) \geq 1 - \delta,$$

as $\xi_t$ is independent of the history $\mathcal{H}_t, \mathcal{Z}_t$. Finally, the inflated coefficient $\rho(P_\zeta)$ defined in Table 1 suffices to make $\tilde{f}_t(\cdot) \in [L_t(\cdot; P_\zeta), U_t(\cdot; P_\zeta)]$ reasonable.

**Proposition 2.** *Under linear setups in Equation (38) and Proposition 1, if Equation (5) is satisfied, linear Ensemble++ is reasonable, i.e., $\forall t \in [T], \tilde{f}_t(\cdot) = f_{\theta_t}(\cdot, \zeta_t) \in [L_t(\cdot; P_\zeta), U_t(\cdot; P_\zeta)]$ w.p. $1-\delta$.*

**Proposition 3.** *Under linear setups in Equation (38) and Proposition 1, if Equation (5) is satisfied, linear Ensemble++ using reference distribution $P_\zeta$ is $p(P_\zeta)$-optimistic.*

### F.2.2 PROOF OF PROPOSITION 3

Let $A_t = \max_{a \in \mathcal{A}_t} \tilde{f}_t(a)$ and $A_t^* = \max_{a \in \mathcal{A}_t} f^*(a)$. Conditioned on $\mathcal{G}$ and confidence event,

$$\tilde{f}_t(A_t) - f^*(A_t^*) \geq \tilde{f}_t(A_t^*) - f^*(A_t^*) \geq \tilde{f}_t(A_t^*) - U^*(A_t^*)$$

$$= \langle \phi(A_t^*), 2\beta_t \mathbf{A}_{t-1}\zeta_t \rangle - \beta_t \|A_t^*\|_{\boldsymbol{\Sigma}_{t-1}}$$

$$= 2\beta_t \sqrt{\phi(A_t^*)^\top \mathbf{A}_{t-1}\mathbf{A}_{t-1}^\top \phi(A_t^*)} \left\langle \frac{\phi(A_t^*)^\top \mathbf{A}_{t-1}}{\|\phi(A_t^*)^\top \mathbf{A}_{t-1}\|}, \zeta_t \right\rangle - \beta_t \|\phi(A_t^*)\|_{\boldsymbol{\Sigma}_{t-1}}$$

$$\geq \beta_t \|\phi(A_t^*)\|_{\boldsymbol{\Sigma}_{t-1}} \left( \left\langle \frac{\phi(A_t^*)^\top \mathbf{A}_{t-1}}{\|\phi(A_t^*)^\top \mathbf{A}_{t-1}\|}, \zeta_t \right\rangle - 1 \right).$$

We consider the conditional probability,

$$\mathbb{P}(\tilde{f}_t(A_t) \geq f^*(A_t^*) \mid \mathcal{H}_t, \mathbf{Z}_t) \geq \mathbb{P}\left( \beta_t \|\phi(A_t^*)\|_{\boldsymbol{\Sigma}_{t-1}} \left( \left\langle \frac{\phi(A_t^*)^\top \mathbf{A}_{t-1}}{\|\phi(A_t^*)^\top \mathbf{A}_{t-1}\|}, \zeta_t \right\rangle - 1 \right) \mid \mathcal{H}_t, \mathcal{Z}_t \right)$$

$$= \mathbb{P}(\langle v, \zeta_t \rangle \geq 1), \tag{39}$$

where $v$ is a fixed unit vector in $\mathbb{R}^M$. The final probability bound in $Equation$ (39) for each reference distribution $P_\zeta$ is essentially the anti-concentration bounds. Please find the anti-concentration results for each distribution in Appendix G, resulting in the Table 1.

*Proof.* The Theorem 1 follows directly from Propositions 2 and 3 and Theorem 3. Additionally, it requires the Azuma's inequality for the sum of bounded martingale difference: as $U_t(\cdot) - L_t(\cdot) \leq 2$ is bounded, we have

$$\sum_{t \in [T]} \mathbb{E}[(U_t(A_t) - L_t(A_t)) \mid \mathcal{H}_t] - (U_t(A_t) - L_t(A_t)) \leq O(\sqrt{T \log(1/\delta)}),$$

with probability at least $1 - \delta$.

Then, it suffices to bound the summation of width between upper and lower confidence bounds

$$\sum_{t\in[T]} (U_t(A_t) - L_t(A_t)) \le \rho(P_\zeta) \sum_{t\in[T]} 2\beta_t \|\phi(A_t)\|_{\Sigma_{t-1}},$$

which depends on a distribution-dependent coefficient $\rho(P_\zeta)$. Under linear bandit setups in assumption 1, we use the elliptical potential lemma (e.g. Lemma 19.4 in (Lattimore & Szepesvári, 2020) and (Abbasi-Yadkori et al., 2011a)) to bound this summation. $\square$

## G   INDEX DISTRIBUTION: SAMPLING, ISOTROPY, CONCENTRATION AND ANTI-CONCENTRATION

**Definition 9** (Isotropic). *A distribution $P$ over $\mathbb{R}^M$ is called isotropic if $\mathbb{E}_{X\sim P}[X_i X_j] = \delta_{ij}$, i.e., $\mathbb{E}_{X\sim P}[XX^\top] = I$. Equivalently, $P$ is isotropic if $\mathbb{E}_{X\sim P}[\langle X, x\rangle^2] = \|x\|^2$, for all $x \in \mathbb{R}^M$.*

Isotropy property (Definition 9) is used for update distribution and proving the Proposition 1. The sub-Gaussianness (Definition 3) in concentration property is used for perturbation distributions and proving Lemma 1. The concentration and anti-concentration properties are used for reference distributions and discussion on the reasonableness condition (Proposition 2) and optimism condition (Proposition 3).

Let us discuss each distribution case by case.

### G.1   SPHERE $P_\zeta = \mathcal{U}(\sqrt{M}\mathbb{S}^{M-1})$

---
**Algorithm 2** Symmetric Index Sampling for $\mathcal{U}(\sqrt{M}\mathbb{S}^{M-1})$

---
**Input:** Number of ensemble members $M$
 1: Sample vector $v$: $v_i \sim N(0,1)$ for $i = 1, \ldots, M$
 2: Construct index vector: $\xi = \sqrt{M}v/\|v\|$
 3: **Return** $\xi$

---

**Isotropy**. By the rotational invariance of sphere distribution, we know for any fixed orthogonal matrix $Q$,

$$\langle \zeta, x\rangle \sim \langle Q\zeta, x\rangle = \langle \zeta, Q^\top x\rangle, \quad \forall x \in \mathbb{R}^d.$$

Then, for any fixed $x$, we select $M$ orthogonal matrix $Q_1, \ldots, Q_M$ to rotate $x$ such that $Q_i^\top x = \|x\|e_i$ where $e_i$ is the $i$-th coordinate vector. With this construction, for any fixed $x$,

$$M\mathbb{E}[\langle \zeta, x\rangle^2] = \mathbb{E}[\sum_{i=1}^{M} \langle \zeta, x_i\rangle^2] = \mathbb{E}[\|x\|^2 \sum_{i=1}^{M} \zeta_i^2] = M\|x\|^2$$

and hence $\mathbb{E}[\langle \zeta, x\rangle^2] = \|x\|^2$, which is the definition of isotropic random vector.

**Concentration**. By definition, $\|\zeta\| = \sqrt{M}$. For a random variable $\zeta \sim \mathcal{U}(\mathbb{S}^{M-1})$ and any fixed $v \in \mathbb{S}^{M-1}$, the inner product follows the transformed Beta distribution

$$\langle \zeta, v\rangle \sim 2\,\mathrm{Beta}(\frac{M-1}{2}, \frac{M-1}{2}) - 1.$$

Evidenced by (Skorski, 2023; Li, 2024a), $P_\zeta = \mathcal{U}(\sqrt{M}\mathbb{S}^{M-1})$ is 1-sub-Gaussian. For finite action set $\mathcal{A}$, using the concentration of Beta random variables with union bound, we have

$$\mathbb{P}\left(\forall a \in \mathcal{A}, \langle \zeta, \phi(a)\rangle \le \|\phi(a)\|\sqrt{\log \frac{2|\mathcal{A}|}{\delta}}\right) \ge 1 - \delta,$$

**Anti-concentration**. Let's start by rewriting the problem in terms of the incomplete Beta function:

Given:

$$X \sim \mathrm{Beta}\left(\frac{M-1}{2}, \frac{M-1}{2}\right)$$

We want to find:

$$P\left(\langle \zeta, v \rangle \geq 1\right) = P\left(2X - 1 > \frac{1}{\sqrt{M}}\right) = P\left(X > \frac{1}{2} + \frac{1}{2\sqrt{M}}\right).$$

**Theorem 4.** *For all $d \geq 2$, the random variable $X \sim \text{Beta}\left(\frac{d-1}{2}, \frac{d-1}{2}\right)$ has the following anti-concentration behavior*

$$P\left(X > \frac{1}{2} + \frac{1}{2\sqrt{d}}\right) \geq \frac{1}{2} - \frac{e^{1/12}}{\sqrt{2\pi}}.$$

**Remark 6.** *We did not find any literature that can help derive such anti-concentration results for Beta distribution.*

*Proof.* Using the incomplete Beta function $I_x(a, b)$, this probability can be expressed as:

$$P\left(X > \frac{1}{2} + \frac{1}{2\sqrt{d}}\right) = 1 - I_{\left(\frac{1}{2} + \frac{1}{2\sqrt{d}}\right)}\left(\frac{d-1}{2}, \frac{d-1}{2}\right)$$

To compute $I_{\left(\frac{1}{2} + \frac{1}{2\sqrt{d}}\right)}\left(\frac{d-1}{2}, \frac{d-1}{2}\right)$, we will use the following relationship for the regularized incomplete Beta function $I_x(a, b)$:

$$I_x(a, b) = \frac{B(x; a, b)}{B(a, b)}$$

where $B(x; a, b)$ is the incomplete Beta function and $B(a, b) := B(1; a, b)$ is the complete Beta function.

For $a = b = \frac{d-1}{2}$, the complete Beta function is:

$$B\left(\frac{d-1}{2}, \frac{d-1}{2}\right) = \frac{\Gamma\left(\frac{d-1}{2}\right)\Gamma\left(\frac{d-1}{2}\right)}{\Gamma(d-1)}$$

Using the property of the Gamma function:

$$\Gamma(n + 1) = n\Gamma(n).$$

Let's compute the incomplete Beta function for $x = \frac{1}{2} + \frac{1}{2\sqrt{d}}$ and $a = b = \frac{d-1}{2}$:

1. Calculate the incomplete Beta function $B\left(x; \frac{d-1}{2}, \frac{d-1}{2}\right)$:

$$B\left(\frac{1}{2} + \frac{1}{2\sqrt{d}}; \frac{d-1}{2}, \frac{d-1}{2}\right) = \int_0^{\frac{1}{2} + \frac{1}{2\sqrt{d}}} t^{\frac{d-3}{2}} (1 - t)^{\frac{d-3}{2}} \, dt$$

As $f(t) = t^{\frac{d-3}{2}} (1 - t)^{\frac{d-3}{2}}$ is symmetric at $t = 1/2$ in the interval $[0, 1]$,

$$B\left(\frac{1}{2} + \frac{1}{2\sqrt{d}}; \frac{d-1}{2}, \frac{d-1}{2}\right) = \frac{1}{2} B\left(\frac{d-1}{2}, \frac{d-1}{2}\right) + \int_{\frac{1}{2}}^{\frac{1}{2} + \frac{1}{2\sqrt{d}}} t^{\frac{d-3}{2}} (1 - t)^{\frac{d-3}{2}} \, dt.$$

2. Calculate the regularized incomplete Beta function $I_x(a, b)$:

$$I_{\left(\frac{1}{2} + \frac{1}{2\sqrt{d}}\right)}\left(\frac{d-1}{2}, \frac{d-1}{2}\right) = \frac{B\left(\frac{1}{2} + \frac{1}{2\sqrt{d}}; \frac{d-1}{2}, \frac{d-1}{2}\right)}{B\left(\frac{d-1}{2}, \frac{d-1}{2}\right)}$$

As the function $f(t) = t^{\frac{d-3}{2}} (1 - t)^{\frac{d-3}{2}}$ achieves the maximum at $t = 1/2$, we could upper bound the incomplete Beta function by

$$\int_{\frac{1}{2}}^{\frac{1}{2} + \frac{1}{2\sqrt{d}}} t^{\frac{d-3}{2}} (1 - t)^{\frac{d-3}{2}} \, dt \leq \left(\frac{1}{4}\right)^{\frac{d-3}{2}} \left(\frac{1}{2\sqrt{d}}\right) = \left(\frac{1}{2}\right)^{d-3} \left(\frac{1}{2\sqrt{d}}\right). \tag{40}$$

The complete Beta function can be expressed as

$$B\left(\frac{d-1}{2},\frac{d-1}{2}\right) = \frac{\Gamma\left(\frac{d-1}{2}\right)\Gamma\left(\frac{d-1}{2}\right)}{\Gamma(d-1)},$$

where $\Gamma(\cdot)$ is the Gamma function. We use the Stirling's Approximation on Gamma function which could provide strict lower bound(Nemes, 2015)

$$\Gamma(z) \geq \sqrt{2\pi}z^{z-\frac{1}{2}}e^{-z},$$

and upper bound (Gronwall, 1918)

$$\Gamma(z) \leq \sqrt{2\pi}z^{z-\frac{1}{2}}e^{-z+\frac{1}{12z}}$$

for all $z > 0$. Immediately, the lower bound of the complete Beta function is

$$B\left(\frac{d-1}{2},\frac{d-1}{2}\right) \geq \frac{\sqrt{2\pi}((d-1)/2)^{d-2}e^{-(d-1)}}{(d-1)^{d-\frac{3}{2}}e^{-d+1+\frac{1}{12(d-1)}}} = \sqrt{2\pi}\left(\frac{1}{2}\right)^{d-2}(d-1)^{-1/2}e^{-\frac{1}{12(d-1)}}.$$

As $e^{-\frac{1}{12(d-1)}} \geq e^{-1/12}$ whenever $d \geq 2$, we further lower bound

$$B\left(\frac{d-1}{2},\frac{d-1}{2}\right) \geq \sqrt{2\pi}e^{-1/12}\left(\frac{1}{2}\right)^{d-2}\frac{1}{\sqrt{d}}. \tag{41}$$

Finally, combining Equations (40) and (41) yields

$$I_{\left(\frac{1}{2}+\frac{1}{2\sqrt{d}}\right)}\left(\frac{d-1}{2},\frac{d-1}{2}\right) \leq \frac{1}{2} + \frac{2e^{1/12}\left(\frac{1}{2\sqrt{d}}\right)}{\sqrt{2\pi}\frac{1}{\sqrt{d}}}$$

$$\leq \frac{1}{2} + \frac{e^{1/12}}{\sqrt{2\pi}},$$

and

$$P(X > \frac{1}{2} + \frac{1}{2\sqrt{d}}) \geq \frac{1}{2} - \frac{e^{1/12}}{\sqrt{2\pi}} \approx 0.0668.$$

$\square$

### G.2  CUBE $P_\zeta = \mathcal{U}(\{1,-1\}^M)$

---
**Algorithm 3** Symmetric Index Sampling for $\mathcal{U}(\{-1,1\}^M)$

---
**Input:** Number of ensemble members $M$
1: Sample vector $\xi$: $\xi_i \sim \mathcal{U}(\{-1,1\})$ for $i = 1,\ldots,M$
2: **Return** $\xi$

---

**Isotropy**. Easy to verify by definition.

**Concentration**. By definition, $\|\xi\| = \sqrt{M}$. Also notice that we could sample the random vector $\zeta$ by sample each entry independently from $\zeta_i \sim \mathcal{U}(\{1,-1\})$ for $i \in [M]$. Then, for any $v \in \mathbb{S}^{M-1}$, by independence,

$$\mathbb{E}[\exp(\lambda\langle v,\zeta\rangle)] = \prod_{i=1}^m \mathbb{E}[\exp(\lambda v_i z_i)] \leq \prod_{i=1}^m \exp(\lambda^2 v_i^2) = \exp(\lambda^2 \sum_i v_i^2).$$

The inequality is due to MGF of rademacher distribution (e.g. Example 2.3 in (Wainwright, 2019)). Then we confirm that $P_\zeta = \mathcal{U}(\{1,-1\}^M)$ is 1-sub-Gaussian. For finite action set $\mathcal{A}$, we have from sub-Gaussian property

$$\mathbb{P}\left(\forall a \in \mathcal{A}, \langle\zeta,\phi(a)\rangle \leq \|\phi(a)\|\sqrt{\log\frac{2|\mathcal{A}|}{\delta}}\right) \geq 1-\delta.$$

**Anti-concentration**. Using the anti-concentration result from (Hollom & Portier, 2023), we have for any fixed unit vector $v$ in $\mathbb{R}^M$

$$P(\langle\zeta,v\rangle) \geq 7/32 \approx 0.21875.$$

---

**Algorithm 4** Symmetric Index Sampling for $N(0, \boldsymbol{I}_M)$

---

**Input:** Number of ensemble members $M$
1: Sample vector $\xi$: $\xi_i \sim N(0, 1)$ for $i = 1, \ldots, M$
2: **Return** $\xi$

---

### G.3 GAUSSIAN $P_\zeta = N(0, I_M)$

**Isotropy**. Easy to verify by definition.

**Concentration**. The concentration property comes directly from the Chernoff bound for standard Gaussian random variable together with union bound argument. For any $\alpha > 0$, we have

$$\mathbb{P}(\|\zeta\| \leq \alpha\sqrt{M}) \geq \mathbb{P}(\forall 1 \leq i \leq M, |\zeta_i| \leq \alpha) \geq 1 - M\mathbb{P}(|\zeta_i| \geq \alpha).$$

Standard concentration inequality for Gaussian random variable gives, $\forall \alpha > 0$,

$$\mathbb{P}(|\zeta_i| \geq \alpha) \leq 2e^{-\alpha^2/2}.$$

Plugging everything together with $\alpha = \sqrt{2\log\frac{2M}{\delta}}$ gives the desired result, which is

$$\|\zeta\| \leq \sqrt{2M\log\frac{2M}{\delta}}, \quad w.p. \ 1 - \delta.$$

For the case of finite action set $\mathcal{A}$,

$$\mathbb{P}\left(\forall a \in \mathcal{A}, \langle\zeta, \phi(a)\rangle \leq \|\phi(a)\|\sqrt{\log\frac{2|\mathcal{A}|}{\delta}}\right) \geq 1 - \delta.$$

**Anti-concentration**. Here $\langle\zeta, v\rangle \sim N(0, 1)$ for for any fixed unit vector $v$ in $\mathbb{R}^M$.

$$P(N(0, 1) \geq 1) = \frac{1}{2}\operatorname{erfc}\left(\frac{1}{\sqrt{2}}\right) \geq \frac{1}{4\sqrt{e\pi}} \approx 0.0856$$

### G.4 COORD $P_\zeta = \mathcal{U}(\sqrt{M}\{\pm e_1, \ldots, \pm e_M\})$

---

**Algorithm 5** Symmetric Index Sampling for $\mathcal{U}(\sqrt{M}\{\pm e_1, \ldots, \pm e_M\})$

---

**Input:** Number of ensemble members $M$
1: Sample index: $i \sim \mathcal{U}(\{1, \ldots, M\})$
2: Sample sign: $s \sim \mathcal{U}(\{-1, 1\})$
3: Construct index vector: $\xi = s\sqrt{M}e_i$
4: **Return** $\xi$

---

**Isotropy**. Easy to verify by definition,

$$\mathbb{E}[\zeta\zeta^\top] = \frac{1}{2M}\sum_{i=1}^M 2Me_ie_i^\top = \boldsymbol{I}. \tag{42}$$

**Concentration**. By definition, $\|\zeta\| = \sqrt{M}$.

**Anti-concentration.**

$$P(\langle\zeta, v\rangle \geq 1) = \frac{1}{2M}\sum_{j\in[M]}(\mathbb{1}_{v_j \geq \frac{1}{\sqrt{M}}} + \mathbb{1}_{-v_j \geq \frac{1}{\sqrt{M}}}) = \frac{1}{2M}\sum_{j\in[M]}(\mathbb{1}_{|v_j| \geq \frac{1}{\sqrt{M}}}) \geq \frac{1}{2M},$$

where the last inequality is due to a simple fact that for any fixed $v \in \mathbb{R}^M$ with unit norm $\|v\| = 1$, there always exists an entry $j \in [M]$ with $|v_j| \geq \frac{1}{\sqrt{M}}$.

---

**Algorithm 6** Symmetric Index Sampling for $s$-sparse random vector

---

**Input:** Number of ensemble members $M$, sparsity $s$
1: Sample sign: $\omega_i \sim \mathcal{U}(\{-1, 1\})$ for $i = 1, \ldots, M$
2: Construct a set $\mathcal{S}$ by randomly pick $s$ elements from $\{1, \ldots, M\}$ without replacement
3: Let $\eta_i = 1$ for $i \in \mathcal{S}$ and $\eta_{i'} = 0$ for $i' \in \{1, \ldots, M\} \setminus \mathcal{S}$
4: Construct index vector $\xi$: $\xi_i = \omega_i \cdot \eta_i$
5: **Return** $\xi$

---

G.5   SPARSE DISTRIBUTION $P_\zeta$

**Definition 10** ($s$-sparse distribution). *The sparse vector is in the form $\zeta = \sqrt{\frac{M}{s}} \eta \odot \omega$ where $P_\omega := \mathcal{U}(\{1, -1\}^M)$, and $\eta$ is independently and uniformly sampled from all possible $s$-hot vectors, where $s$-hot vectors is with exactly $s$ non-zero entries with number $1$. This construction is introduced by (Kane & Nelson, 2014).*

**Isotropy**. By definition,

$$\mathbb{E}[\zeta_j \zeta_k] = \frac{M}{s} \mathbb{E}[\eta_j \eta_k] \mathbb{E}[\omega_j \omega_k] = \frac{M}{s} \delta_{jk} \mathbb{E}[\omega_j] = \delta_{hk}. \tag{43}$$

Therefore, the sparse distribution in Definition 10 is indeed isotropic distribution.

**Concentration**. $\|\zeta\| = \sqrt{M}$.

**Anti-concentration**. Not clear.

## H  APPLICATION OF ENSEMBLE++ IN REAL-WORLD DECISION-MAKING: A CASE STUDY ON CONTENT MODERATION

Real-world decision-making often faces uncertainty due to incomplete information about the environment. Intelligent agents must not only quantify this uncertainty but also actively gather information to resolve it. This challenge is particularly pronounced in real-time online decision-making involving foundation models—large-scale AI models pretrained on vast datasets that process unstructured inputs like text and images.

**Case Study: Content Moderation**  Content moderation on digital platforms provides a compelling example of these challenges (Gorwa et al., 2020). Initially, human reviewers detected violations of community standards (Roberts, 2019), but as platforms like Facebook (Meta, 2024), Twitter (Corp., 2024), and Reddit (Reddit, 2024) scaled, the sheer volume of posts necessitated automated content moderation. AI systems leveraging foundation models (Weng et al., 2023) offer real-time moderation capabilities, reducing human workload. However, these models, pretrained on historical data, often struggle with uncertainty in novel and rare situations encountered in real-time online traffic, resulting in errors (Markov et al., 2023).

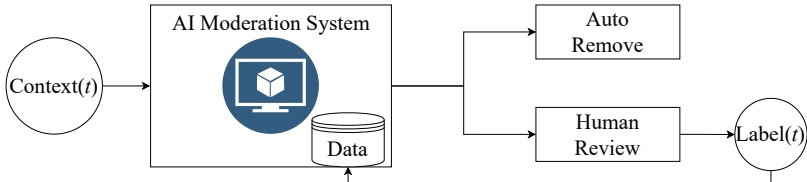

Figure 10: The Human-AI agile collaboration pipeline for content moderation: At time $t \in \mathbb{N}$, the AI moderation system receives a post context $(x_t)$ and decides whether to auto-remove or send it for human review. If reviewed, the AI system integrates human feedback via label $(y_t)$ to overturn the AI decision and improve its future performance. This pipeline balances minimizing human workload with ensuring long-term reliability and safety.

To ensure reliable content moderation, real-time human feedback is crucial for correcting AI errors, reducing uncertainty, and refining detection policies. The collaboration between humans and AI seeks to minimize human involvement (by exploiting current AI capabilities) while ensuring long-term reliability (by exploring uncertain content for human review to improve future decision-making). This pipeline is illustrated in Figure 10, where human reviewers intervene only when necessary, allowing the system to adapt and improve over time.

**Contextual Bandit Framework for Content Moderation**  The challenges of content moderation can be framed within the contextual bandit problem (Wang et al., 2005; Langford & Zhang, 2007), a fundamental online decision-making task where contextual information, such as unstructured language and visual inputs, influences decision-making. In this setting, the agent (the AI moderation system) selects an action (e.g., whether to auto-remove or flag content for human review) based on the context (the post content) and continuously updates its policy based on the feedback it receives (human corrections). This interaction is depicted in Figure 10, where the AI system must decide on each post's fate and incorporate human feedback to refine its moderation capabilities.

**Challenges of Foundation Models in Decision-Making**  Krishnamurthy et al. (2024) demonstrated that even state-of-the-art large language models, like GPT-4 (OpenAI, 2023), struggle to make effective online decisions in environments such as the multi-armed bandit (MAB) problem without an external method to estimate uncertainty, such as history summarization. The inherent limitation of foundation models in real-time decision-making highlights the need for algorithmic interventions like fine-tuning or dataset curation to enhance their performance in uncertain environments (Krishnamurthy et al., 2024).

To address these challenges, our work integrates Ensemble++ into foundation models to enable effective online decision-making in complex and uncertain environments. By leveraging Ensem-

ble++'s uncertainty-aware mechanisms, foundation models can dynamically adjust their exploration strategies and provide more reliable decisions in real-time applications like content moderation.

**GPT-Ensemble++ for Context-Aware Decision-Making**  Following the Ensemble++ construction from Equation (2), we integrate the pretrained GPT-2 backbone as a feature extractor $\phi(\cdot; w)$ to create **GPT-Ensemble++**. This model takes a context $x$ and a random index $\zeta$ as input and outputs a value for each action $a \in \mathcal{A}(x)$, the valid decision set associated with the context:

$$f_\theta(x, \zeta)[a] = \langle \phi(x; w), b^a \rangle + \langle \text{sg}[\phi(x; w)], \mathbf{A}^a \zeta \rangle := f_\theta((x, a), \zeta),$$

where $\mathbf{A}^a$ and $b^a$ are action-specific parameters for each action $a$. The contextual bandit formulation allows GPT-Ensemble++ to efficiently balance exploration and exploitation in real-time, addressing both uncertainty estimation and policy refinement.

**Additional Insights from Prior Work**  Krishnamurthy et al. (2024) demonstrated that even the most advanced large language models, such as GPT-4 (OpenAI, 2023) with various advanced prompt designs, are ineffective in making online decisions within simple multi-armed bandit (MAB) problems unless supplemented with MAB-specific uncertainty estimation methods like external history summarization. More sophisticated algorithmic interventions, such as fine-tuning or dataset curation, might be necessary to enhance LLM-based decision-making agents in complex settings (Krishnamurthy et al., 2024). Our work addresses this gap by integrating our uncertainty-aware Ensemble++ techniques into foundation models, thereby enabling effective online decision-making in complex and uncertain environments.

**GPT-Ensemble++ Implementation Details**  Following the theoretical framework outlined in Section 4, we integrate Ensemble++ with GPT-2 to create GPT-Ensemble++. The implementation involves the following steps:

1. **Feature Extraction**: Utilize the pretrained GPT-2 model to extract contextual features $\phi_w(x)$ from unstructured inputs such as text.

2. **Ensemble Integration**: Apply the Ensemble++ architecture by combining the extracted features with action-specific parameters $\mathbf{A}^a$ and $b^a$ to generate action values.

3. **Decision Making**: For each context $x$, sample an index $\zeta$ and compute the value for each action $a$ using the GPT-Ensemble++ model. Select the action with the highest estimated value.

4. **Incremental Updates**: Upon receiving feedback $y_t$ from human reviewers, update the Ensemble++ parameters incrementally to refine future decision-making.

The integration of Ensemble++ with foundation models, particularly GPT-based architectures, addresses the challenges of real-time decision-making in uncertain environments. By enabling fast incremental uncertainty estimation and scalable exploration, GPT-Ensemble++ can adaptively refine its policies in dynamic settings like content moderation, ensuring long-term reliability and minimizing human workload. This approach demonstrates the practical applicability of Ensemble++ in high-dimensional, real-time decision-making scenarios.

# I IN-DEPTH EMPIRICAL AND ABLATION STUDIES

In this section, we dive into the intricacies of each evaluation testbed. Through a comprehensive set of empirical results, we'll further illuminate the benefits afforded by Ensemble++. All experiments are conducted on P40 GPUs to maintain processing standardization.

## I.1 LINEAR BANDIT ENVIRONMENTS

We begin by examining linear Ensemble++ in linear bandits. In this experiment, we focus on studying the impact of perturbation and reference distributions, as well as the effect of #ensembles $M$.

**environment Settings:** We use the action feature set $\mathcal{X}$ to denote the set of features $\phi(a) : a \in \mathcal{A}$ induced by action set $\mathcal{A}$ and feature mapping $\phi(\cdot)$. We build two linear bandit environments with different action distribution as follow:

- **Finite-action Linear Bandit**: We construct the finite set $\mathcal{X}$ by uniformly sampling a set of action features from the range $[-1/\sqrt{5}, 1/\sqrt{5}]^d$ where $d$ is the ambient dimension of the linear reward function. This environment builds upon prior research Russo & Van Roy (2018). We vary the action size $|\mathcal{X}|$ over a set of $\{100, 1000, 10000\}$, and the ambient dimension across $\{10, 50\}$.

- **Compact-action Linear Bandit**: Let the action feature set $\mathcal{X} = \mathbb{S}^{d-1}$ be the unit sphere. In this environment, we vary the ambient dimension $d$ over a set of $\{10, 50, 100\}$.

In both environments, the reward of each feature $X_t \in \mathbb{R}^d$ is computed as $r_t = X_t^\top \theta + \epsilon$, where $\theta \sim \mathcal{N}(0, 10\mathrm{I})$ is drawn from the multivariate Gaussian prior distribution, and $\epsilon \sim \mathcal{N}(0, 1)$ is an independent additive Gaussian noise term. At every step $t$, only the reward from the chosen feature $X_t$ is discernible. To ensure robust results, each experiment is executed a total of 1000 time steps and repeated 200 times.

**Analysis of Results:** We investigated all 25 combinations of perturbation and reference distribution under different scales of the linear bandit environments and numerous #ensembles $M$. As depicted in Figures 11 to 13, the outcomes across diverse problem scales corroborate each other. **The use of a Gaussian reference distribution significantly enhances performance when the $M$ is relatively small, such as when $M$ is 2 or 4.** As the #ensembles $M$ grows, all combinations show an analogous performance under varying problem scales. However, it is worth noting that for extremely large $M$, such as 512 or 1024, combinations involving the Coord perturbation and Coord reference distribution significantly underperform compared to other combinations. Given that Coord distributions are used in the Ensemble+, the results prompt a compelling argument. Ensemble++ equipped with a continuous reference distribution presents a superior performance, suggesting its potential for surpassing traditional Ensemble+ methods. These findings strongly support the superior advantage of our index sampling method, validating our theoretical analysis.

|          | $|\mathcal{A}| = 100$ | $|\mathcal{A}| = 1000$ | $|\mathcal{A}| = 10000$ |
|----------|------------------------|------------------------|-------------------------|
| $d = 10$ | 8                      | 8                      | 8                       |
| $d = 50$ | 16                     | 16                     | 16                      |

Table 5: Minimum #ensembles $M$ required to match the performance of TS.

**Analysis of Computational Efficiency:** We delve deeper into the effects of varying #ensembles $M$ within Ensemble++. We assess its performance across different combinations of perturbation and reference distributions using an assortment of $M \in \{4, 8, 16, 32, 64, 128, 256, 512, 1024\}$. The outcomes, visualized in Figures 14 and 15, are consistent with the findings illustrated in Figures 11 to 13. We observe that for large $M$, the Coord perturbation and Coord reference distributions degrade performance, indicating that the index sampling method employed by Ensemble+ lacks efficiency. However, when Ensemble++ utilizes Gaussian or Sphere reference distributions, it achieves satisfactory performance, comparable to Thompson Sampling with small $M$. We conduct a deeper

investigation on the minimal number of ensembles $M$ required to match the performance of Thompson Sampling using a **Finite-action Linear Bandit**, as shown in Table 5. Our results suggest that linear Ensemble++ is scalable with the size of action spaces, as the **same $M$ can be used to match Thompson Sampling regardless of the action space size** up to $10,000$. These findings are consistent with the theoretical prediction of $M = O(d \log T)$ as noted in Lemma 1.

**Remark 7** (Limitation of Theorem 1.). *Notice that Theorem 1 suggest that when $M \geq O(d \log T)$, the regret bound of Ensemble sampling would increase with factor $M^{3/2}$, which contradicts with our empirical evidence in Figures 11 to 15.*

**Remark 8** (Good prediction of Theorem 1.). *Our empirical evidence in Figures 11 to 15 confirms the Theorem 1 in finite decision set setting for continuous-support reference distributions: when $M$ is larger then a threshold $O(d \log T)$, the regret has no dependence on $M$.*

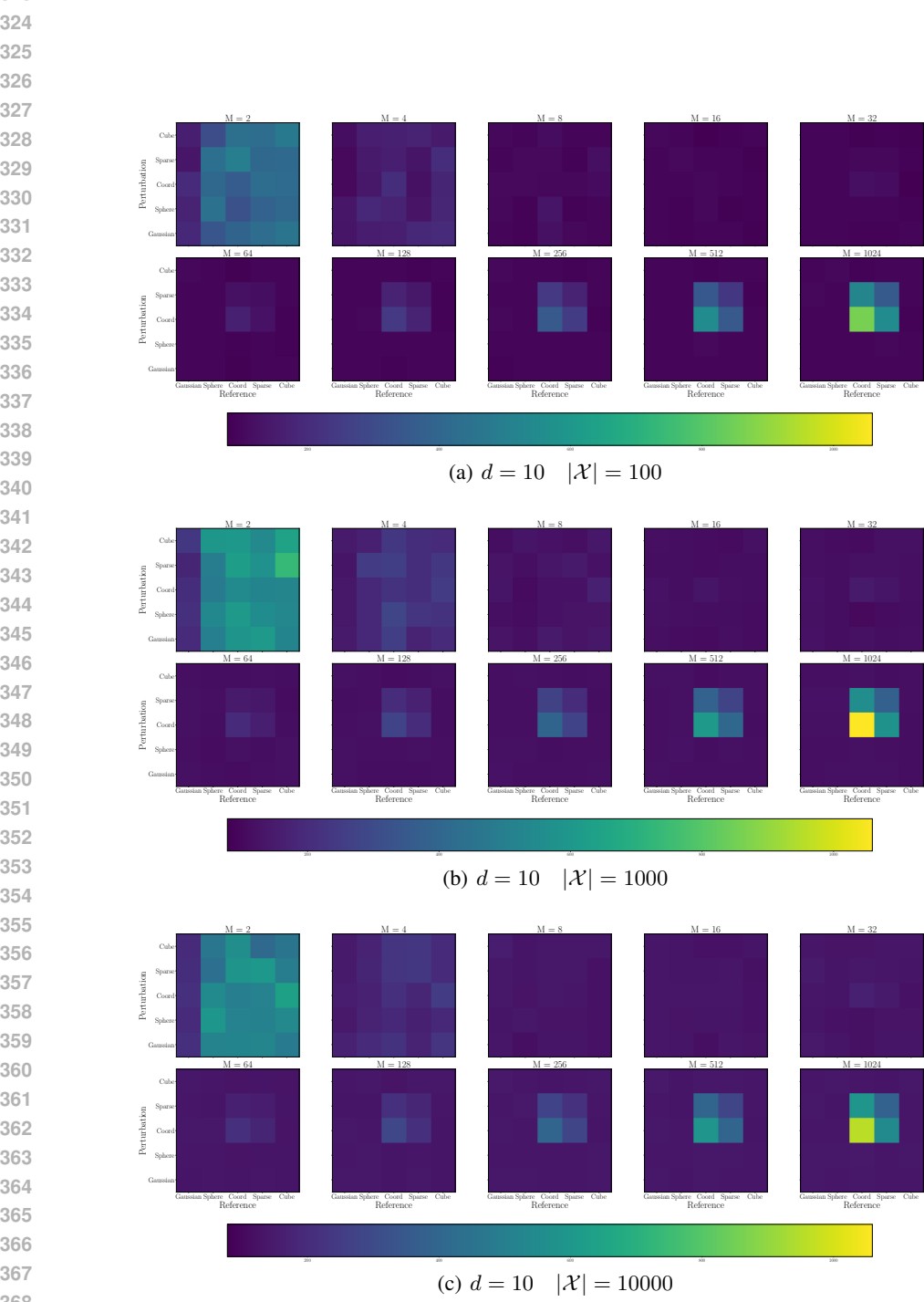

Figure 11: Results on the combinations of perturbation and reference distribution in Finite-action Linear Bandit under action dimension $d = 10$. A deeper color signifies lower accumulated regret and hence superior performance. Gaussian reference distribution significantly enhances performance.

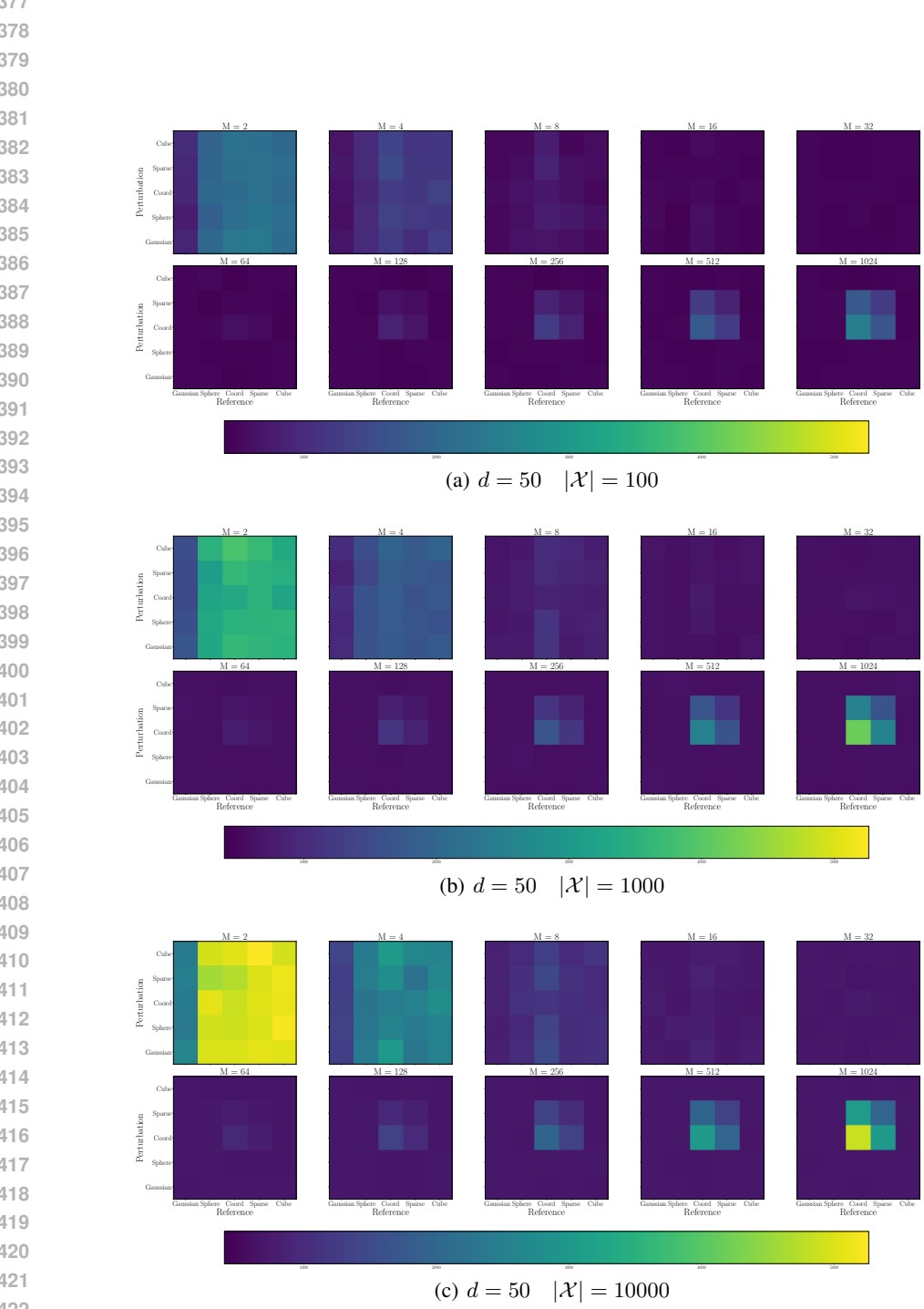

Figure 12: Results on the combinations of perturbation and reference distribution in Finite-action Linear Bandit under action dimension $d = 50$. A deeper color signifies lower accumulated regret and hence superior performance. Gaussian reference distribution significantly enhances performance.

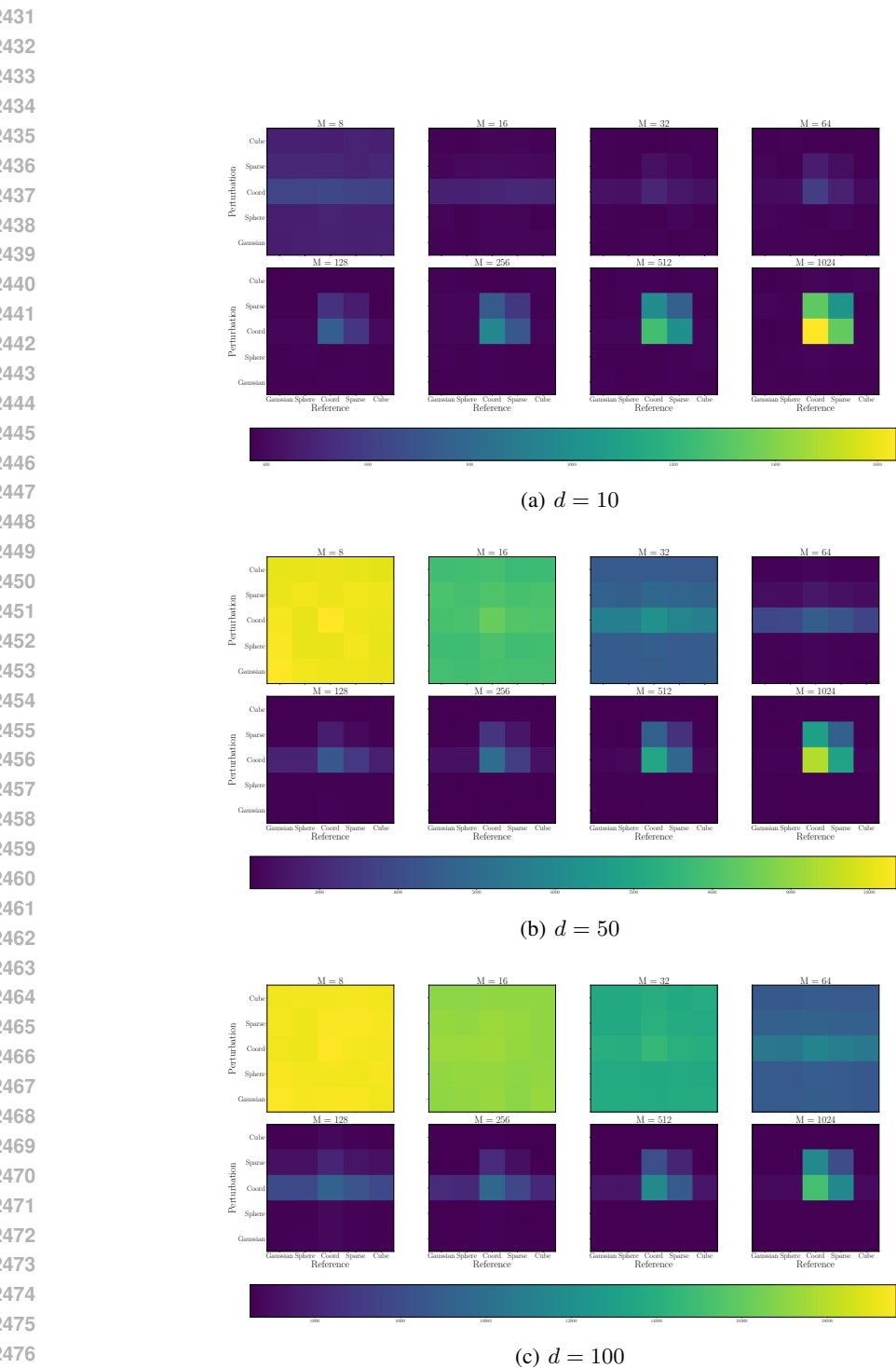

(a) $d = 10$

(b) $d = 50$

(c) $d = 100$

Figure 13: Results on the combinations of perturbation and reference distribution in Compact-action Linear Bandit. A deeper color signifies lower accumulated regret and hence superior performance. Gaussian reference distribution significantly enhances performance.

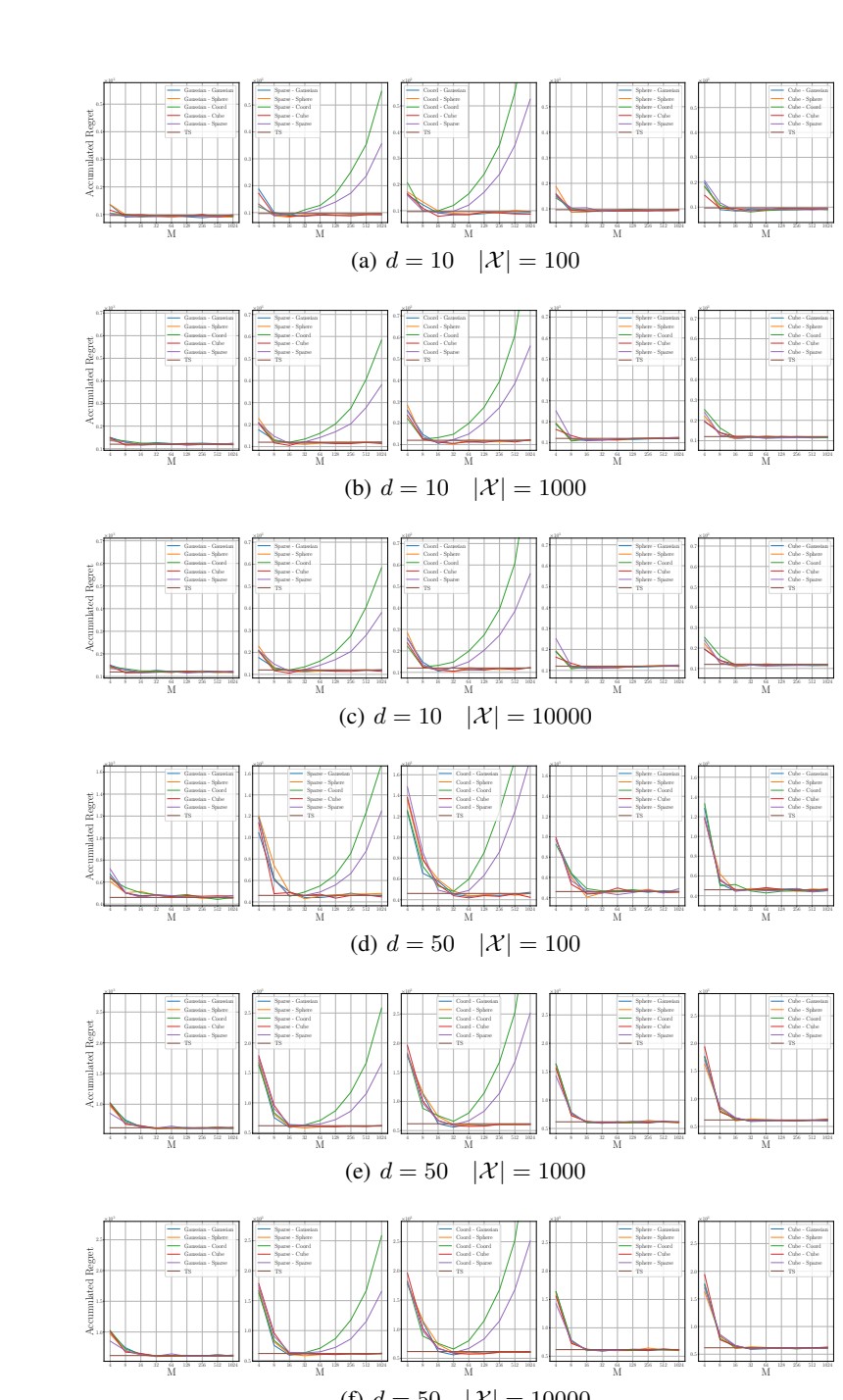

(a) $d = 10 \quad |\mathcal{X}| = 100$

(b) $d = 10 \quad |\mathcal{X}| = 1000$

(c) $d = 10 \quad |\mathcal{X}| = 10000$

(d) $d = 50 \quad |\mathcal{X}| = 100$

(e) $d = 50 \quad |\mathcal{X}| = 1000$

(f) $d = 50 \quad |\mathcal{X}| = 10000$

Figure 14: Results on regret under various #ensembles $M$ in Finite-action Linear Bandit. The label $A - B$ indicates that Ensemble++ uses A as the reference distribution and B as the perturbation distribution. Ensemble++ with Gaussian or Sphere reference distribution could achieve comparable performance with that of Thompson sampling under same $M$ for different action spaces $|\mathcal{X}|$.

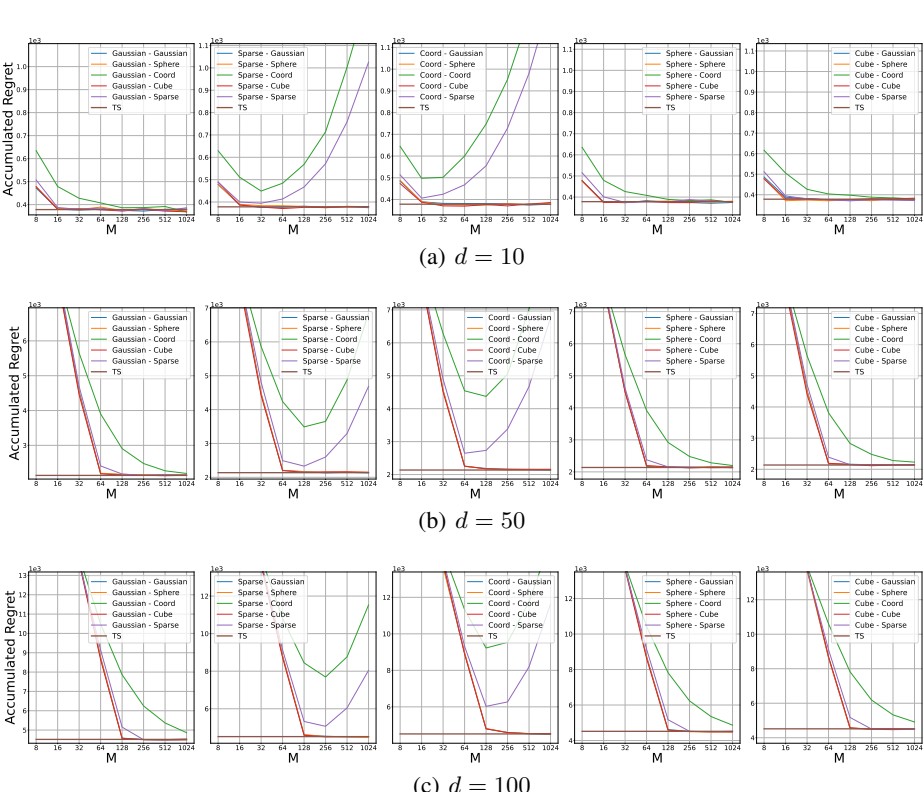

(a) $d = 10$

(b) $d = 50$

(c) $d = 100$

Figure 15: Results on regret under various #ensembles $M$ in Compact-action Linear Bandit. The label $A - B$ indicates that Ensemble++ uses A as the reference distribution and B as the perturbation distribution. Ensemble++ with Gaussian or Sphere reference distribution could achieve comparable performance with that of Thompson sampling under small $M$.

## I.2 NONLINEAR BANDIT ENVIRONMENTS

We conduct a comprehensive comparison of Ensemble++ with several baselines that utilize approximate posterior sampling across a wide range of nonlinear bandits. Furthermore, we provide practical guidelines for effectively employing Ensemble++.

**Environments Settings:** We formulate several nonlinear contextual bandit environments, with rewards generated by nonlinear functions in each.

- **Neural Bandit**: It employs a nonlinear neural model denoted as $f_1(a)$ in reward generation. This model features three fully connected layers, each consisting of 50 units, connected by ReLU activation functions.

- **Quadratic Bandit**: Its reward generation mechanism is built on a quadratic function, expressed as $f_2(a) = 10^{-2}(a^\top \Theta \Theta^\top a)$. Here, $a \in \mathbb{R}^d$ stands for the action, while $\Theta \in \mathbb{R}^{d \times d}$ is a matrix filled with random variables originating from $\mathcal{N}(0, 1)$. This environment is used as the testbed in Zhou et al. (2020).

- **Vector Quadratic Bandit**: Its reward generation mechanism is built on a different quadratic function, expressed as $f_3(a) = 10(a^\top \theta)^2$. Here, $a \in \mathbb{R}^d$ stands for the action, while $\theta \in \mathbb{R}^d$ is a vector filled with random variables generated from a uniform distribution over the unit ball. This environment is utilized as the testbed in Zhou et al. (2020); Xu et al. (2022).

- **UCI Dataset**: Following prior works (Riquelme et al., 2018; Kveton et al., 2020b), we conduct contextual bandits with $N$-class classification using the UCI datasets (Asuncion et al., 2007) Mushroom and Shuttle. Specifically, given a data feature $x \in \mathbb{R}^d$ in the dataset, we construct context vectors for $N$ arms, such as $a^{(1)} = (x, 0, \cdots, 0), \cdots, a^{(N)} = (0, 0, \cdots, x) \in \mathbb{R}^{Nd}$. Only the arm $a^{(j)}$ where $j$ matches the correct class of this data $x$ has a reward of 1, while all other arms have a reward of 0.

In all environments, the original reward $r$ is disrupted by Gaussian noise $\epsilon$ drawn from $\mathcal{N}(0, 0.1)$. For the first three environments, we set the action dimension $d$ to 100 and generate a total of 1000 actions, randomly sampling 50 actions in each round. Each experiment is repeated with 5 distinct random seeds to ensure robust results.

**Comparison Results with Baselines:** We set the Sphere reference distribution, Coord update distribution, and Sphere perturbation distribution for Ensemble++ to compare with baselines. When comparing with Ensemble+ (Osband et al., 2018) and EpiNet (Osband et al., 2023), we use the same hyperparameters, such as prior scale, learning rate, and batch size. Additionally, we employ the same network backbone for feature extraction to ensure fairness. As shown in Figure 16(a) and (b), **Ensemble++ achieves sublinear regret and consistently outperforms these baselines across all environments, demonstrating superior data efficiency.**

For comparison with LMCTS (Xu et al., 2022), we use its official implementation[4] to ensure credible results. As illustrated in Figure 16(c), Ensemble++ consistently outperforms LMCTS. Notably, LMCTS uses the entire buffer data to update the network per step, which incurs significant computational costs. In contrast, **Ensemble++ achieves better performance with bounded computational steps, requiring only a minibatch to update the network.** These findings highlight the effective exploration and computational efficiency of Ensemble++.

**Ablation Study on Quadratic Bandit:** To further evaluate the impact of different design of distributions, we perform an ablation study on the Quadratic Bandit. When fixing the Sphere reference distribution, we find that discrete update distributions such as Coord, Cube, and Sparse achieve similar performance, as shown in Figure 17(a). Conversely, when fixing the Coord update distribution, continuous reference distributions like Sphere and Gaussian also yield comparable performance, as depicted in Figure 17(b). Regarding the perturbation distribution, our findings indicate that it does not significantly influence performance when the neural network is involved in Ensemble++. This is evidenced in Figure 17(c), where all different perturbation distributions achieve similar performance.

---

[4] https://github.com/devzhk/LMCTS

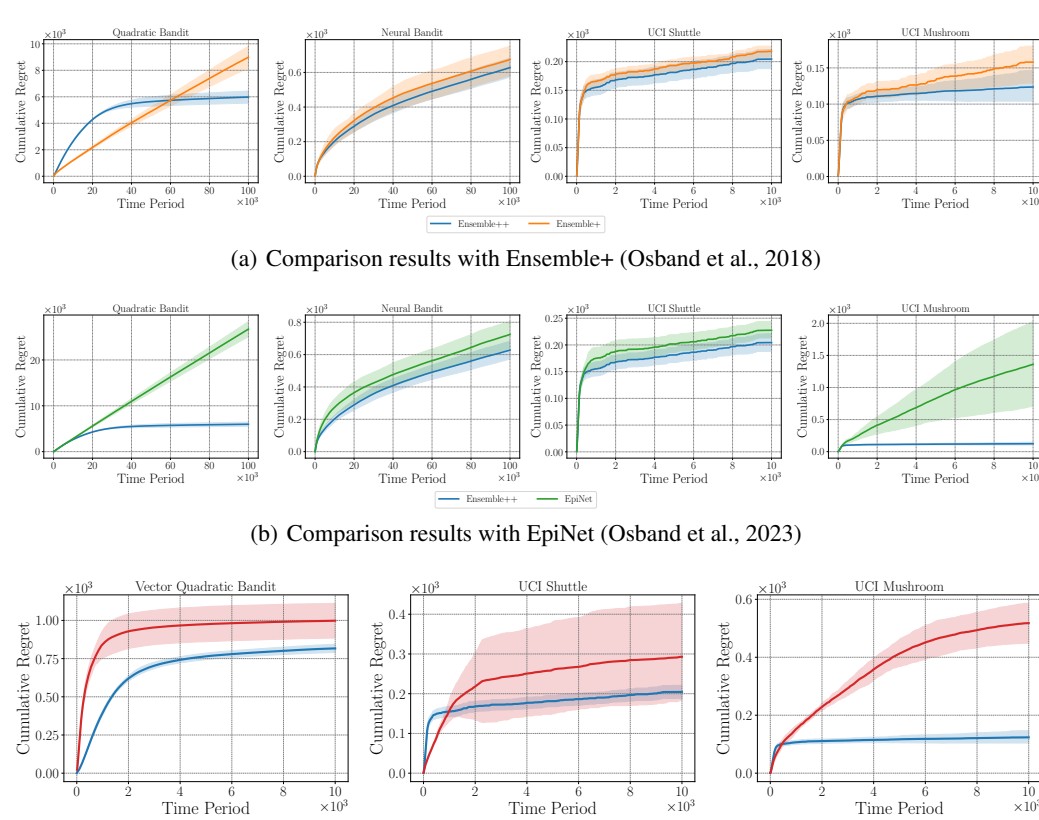

(a) Comparison results with Ensemble+ (Osband et al., 2018)

(b) Comparison results with EpiNet (Osband et al., 2023)

(c) Comparison results with LMCTS (Xu et al., 2022)

Figure 16: Experimental results for different bandits with various baselines. Ensemble++ consistently achieves better performance compared to other methods.

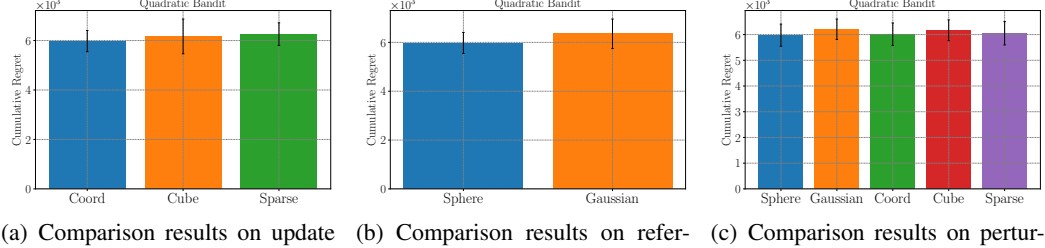

(a) Comparison results on update distribution.  (b) Comparison results on reference distribution.  (c) Comparison results on perturbation distribution.

Figure 17: Ablation studies about different distributions on the Quadratic Bandit.

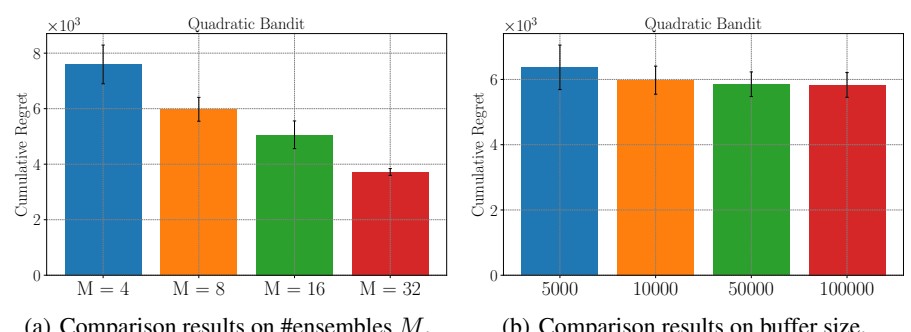

(a) Comparison results on #ensembles $M$.  (b) Comparison results on buffer size.

Figure 18: Ablation studies about key settings on the Quadratic Bandit.

We further conduct additional ablation studies on key settings in Ensemble++. The #ensembles $M$ is a critical hyperparameter in Ensemble++. We find that a larger $M$ can boost performance, as shown in Figure 18(a), which is consistent with findings in linear bandits. However, a larger $M$ also incurs higher computational costs due to the need for more indices during updates. Therefore, we select a moderate $M = 8$ for all other experiments with Ensemble++. The theoretical analysis in Appendix F suggests that it is unnecessary to store the entire history of data. We compare different buffer sizes over a fixed period of 100,000 time steps. As shown in Figure 18(b), too small buffer size does lead to a slight performance drop. Nonetheless, Ensemble++ still achieves comparable performance with a smaller buffer size than the total time period. We set the buffer size to 50,000 for all other experiments with Ensemble++. These ablation studies demonstrate the computational and memory efficiency of Ensemble++.

## I.3 CONFIGURATIONS AND PER-STEP COMPUTATIONAL COSTS

We compare the cumulative regret along with the computation cost of different methods: Ensemble+, EpiNet, and LMCTS, against Ensemble++ on the **Quadratic Bandit** environment as described in Appendix I.2, as shown in Figure 1. For a fair comparison, we configure Ensemble+, EpiNet, and Ensemble++ with the same #ensembles ($M = 8$), identical hidden networks for layer-sharing architecture. All agents use the same update ratio, and the same optimization configurations for learning rate, batch size, and weight decay, etc. For LMCTS and LMCTS(original), we use the official implementation[5] to conduct the experiments. The LMCTS(original) follows the original official implementation, which employs a schedule with increased update number per environment step. For LMCTS, we set the same update ratio as Ensemble++ to ensure bounded per-step computation. Additionally, we use identical hidden networks for both LMCTS and LMCTS(original) as those used in Ensemble++. The detailed comparison the neural architecture and parameter counts with EpiNet can be found in Appendix J. In summary, Ensemble++ can achieve the best sublinear regret with the lowest bounded computational cost.

---

[5]Footnote 4

## J   DETAILED COMPARISON WITH EPINET

In this section, we provide a detailed comparison between Ensemble++ and EpiNet (Osband et al., 2023), focusing on architectural differences, parameter counts, and computational efficiency.

### J.1   OVERVIEW OF EPINET ARCHITECTURE

EpiNet is designed to estimate epistemic uncertainty in neural networks by incorporating an *epistemic index* $z \in \mathbb{R}^{D_z}$ into the model. The EpiNet architecture consists of a base network $\mu_\zeta$ and an epinet $\sigma_\eta^L$, which together produce predictions that account for uncertainty.

The epinet $\sigma_\eta^L$ is typically implemented as a multi-layer perceptron (MLP) with Glorot initialization:

$$\sigma_\eta^L(\tilde{x}, z) := \mathrm{mlp}_\eta([\tilde{x}, z])^\top z \in \mathbb{R}^C,$$

where:

- $\mathrm{mlp}_\eta$ is an MLP parameterized by $\eta$ with output dimensions $\mathbb{R}^{D_z \times C}$.
- $\tilde{x}$ is the shared feature representation of the input $x$.
- $[\tilde{x}, z]$ denotes the concatenation of $\tilde{x}$ and the epistemic index $z$.
- $C$ is the number of output classes.

The final prediction of EpiNet is given by combining the base network and the epinet:

$$f_{\mathrm{EpiNet}}(x, z) = \mu_\zeta(x) + \sigma_\eta^L(\tilde{x}, z) + \sigma^P(\tilde{x}, z),$$

where $\sigma^P$ is a fixed prior function that introduces additional uncertainty modeling.

### J.2   PARAMETER COUNT COMPARISON

We analyze the parameter counts of both EpiNet and Ensemble++ to highlight the differences in computational efficiency.

Consider the following components in EpiNet:

- **Base Network** $\mu_\zeta$: A shared feature extractor with input dimension $D_{\mathrm{in}}$, output dimension $D$, and parameters $\theta_\mu$.
- **Epinet** $\sigma_\eta^L$: An MLP that takes $[\tilde{x}, z]$ as input and outputs $\mathbb{R}^{D_z \times C}$.
- **Fixed Prior** $\sigma^P$: An ensemble of $D_z$ small MLPs, each producing logits for the $C$ classes.

The parameter count for the epinet $\sigma_\eta^L$ is:

$$\mathrm{Params}_{\sigma_\eta^L} = (D + D_z) \times H + H \times (D_z \times C),$$

where $H$ is the hidden layer width of the epinet MLP.

The fixed prior $\sigma^P$ consists of $D_z$ small MLPs. If each MLP has parameters $\mathrm{Params}_{\sigma_{\mathrm{MLP}}^P}$, the total parameter count for $\sigma^P$ is:

$$\mathrm{Params}_{\sigma^P} = D_z \times \mathrm{Params}_{\sigma_{\mathrm{MLP}}^P}.$$

Thus, the total parameter count for EpiNet (excluding the base network) is:

$$\mathrm{Total\ Params}_{\mathrm{EpiNet}} = \mathrm{Params}_{\sigma_\eta^L} + \mathrm{Params}_{\sigma^P}.$$

In Ensemble++, the architecture includes:

- **Base Network** $\psi(\tilde{x}; b)$: Similar to the base network in EpiNet, with parameters $b$.

- **Ensemble Components** $\{\psi(\mathrm{sg}(\tilde{x}); \theta_m)\}_{m=1}^M$: Each ensemble component is a linear head that operates on the shared feature representation $\tilde{x}$.

- **Fixed Prior Ensemble Components** $\{\mathrm{sg}(\psi(\tilde{x}; \theta_{0,m}))\}_{m=1}^M$: Each ensemble component is a linear head that operates on the shared feature representation $\tilde{x}$.

The parameter count for the ensemble components is:

$$\mathrm{Params}_{\mathrm{Ensemble}} = M \times D \times C,$$

since each ensemble component is a linear layer mapping $\mathbb{R}^D$ to $\mathbb{R}^C$.

**Comparison**   For a fair comparison, we set the epistemic index dimension $D_z = M$ in EpiNet. Let's analyze the parameter counts under the assumption that both methods use the same shared feature dimension $D$, output dimension $C$, and ensemble size $M$.

EpiNet Parameters:

$$\mathrm{Params}_{\sigma_\eta^L} = (D + M) \times H + H \times (M \times C).$$

Ensemble++ Parameters:

$$\mathrm{Params}_{\mathrm{Ensemble}} = M \times D \times C.$$

**Key Differences:**

- **EpiNet** introduces additional parameters due to:

  - Concatenation of the epistemic index $z$ with $\tilde{x}$, increasing the input dimension of the epinet MLP.
  - The output dimension of the epinet MLP being $M \times C$, which is larger than $C$ when $M > 1$.
  - The use of an MLP with hidden layers, adding more parameters compared to linear layers.
  - The fixed prior $\sigma^P$, which adds extra parameters due to the ensemble of small MLPs.

- **Ensemble++** uses simple linear heads for ensemble components, resulting in fewer parameters and reduced computational overhead.

J.3   COMPUTATIONAL EFFICIENCY

Ensemble++ offers computational advantages over EpiNet:

- **Simpler Ensemble Components**: Ensemble++ uses linear layers for ensemble components, leading to faster computations during both forward and backward passes.

- **Stop-Gradient Operator**: By applying the stop-gradient operator to the shared features in Ensemble++, we reduce the computational cost of backpropagation, as gradients do not flow through the shared layers from the ensemble components.

- **No Additional Inputs**: Ensemble++ does not concatenate the epistemic index with the shared features, avoiding the increase in input dimensionality that occurs in EpiNet.

In contrast, EpiNet's architecture introduces additional computational costs due to:

- **Higher Input Dimensionality**: Concatenating $z$ with $\tilde{x}$ increases the input dimension of the epinet MLP from $D$ to $D + M$.

- **Complex MLP Structure**: The epinet MLP with hidden layers and larger output dimensions requires more computations, both in the forward pass and during backpropagation.

- **Fixed Prior Computations**: The fixed prior $\sigma^P$ involves additional MLPs, adding to the computational burden.

### J.4 PRACTICAL IMPLICATIONS

The architectural simplicity of Ensemble++ translates to practical benefits:

- **Faster Training and Inference**: Fewer parameters and simpler computations lead to faster model updates and predictions.
- **Scalability**: Ensemble++ scales better to larger ensemble sizes $M$ and higher feature dimensions $D$, making it suitable for high-dimensional tasks.
- **Effective Exploration**: Despite its simplicity, Ensemble++ effectively captures epistemic uncertainty and facilitates efficient exploration, as demonstrated in our empirical evaluations.

In contrast, EpiNet's more complex architecture may hinder its performance in practice:

- **Increased Computational Load**: The larger parameter count and complex computations can slow down training and inference, especially in resource-constrained environments.
- **Diminished Performance**: Empirical studies have shown that EpiNet may demonstrate suboptimal exploration performance in practice (Li et al., 2024), potentially due to the challenges in training the epinet effectively.

### J.5 CONCLUSION

Our comparison highlights that Ensemble++ offers a more parameter-efficient and computationally efficient approach to capturing epistemic uncertainty compared to EpiNet. By utilizing simple linear ensemble components and avoiding unnecessary increases in input and output dimensionality, Ensemble++ achieves better practical performance without sacrificing theoretical guarantees.

