# OpenReview forum: "Scalable Exploration via Ensemble++"
_ICLR.cc/2025/Conference — ICLR 2025 Conference Withdrawn Submission_

### Official Review · Reviewer_rTcE · 2024-11-02

**Soundness:** 3
**Presentation:** 2
**Contribution:** 2
**Rating:** 5
**Confidence:** 3

**Summary:**

This paper presents Ensemble++, an ensemble sampling method that enhances the computational efficiency of Thompson sampling. Recently, Ensemble+ was developed to tackle this challenge; however, it faces an issue with ensemble coupling that adversely affects its performance. To address this problem, Xu et al. later introduced Langevin Monte Carlo Thompson Sampling, which, unfortunately, incurs high computational costs. The aim of this paper is to improve the approach to ensemble coupling while minimizing computational expenses. The proposed approach, Ensemble++, addresses this by implementing decoupled optimization and lifted index sampling, enhancing exploration and uncertainty estimation. Theoretically, the paper shows that it achieves the same regret bounds as exact Thompson sampling in linear contextual bandits, with a per-step computation complexity of $\tilde{O}(\log T)$.
Empirical results in the performance of  Ensemble++ are presented.

**Strengths:**

The paper tackles an important issue: scalable exploration poses a significant challenge in sequential decision-making tasks, including reinforcement learning (RL) and contextual bandits, especially relevant in high-dimensional practical environments.

**Weaknesses:**

The paper lacks details in many instances, which hinders understanding the contributions of the paper.

**Questions:**

1. The two key innovations in the proposed model are noted as (1) a variance-aware discretization method that prevents the exponential growth in ensemble size and (2) a reduction to sequential random projection techniques. However, the current presentation of the paper does not clearly explain how these innovations are applied. It is not clear how the two proposed changes improve upon the shortcomings of the existing approaches. Additionally, the solution approach lacks clarity without the algorithm.

    a) Can you provide a high-level description of how the variance-aware discretization method works and how it prevents exponential growth in ensemble size?

    b) Explain the key steps in applying sequential random projection techniques to their problem.

    c) Include a pseudocode description of the Ensemble++ algorithm to clarify the solution approach.

    d) Explain how the proposed modifications address the shortcomings of existing approaches like Ensemble+.

2. The total computational complexity is $O(d^3 \log T)$. However, it is not clear from the main paper how this result was concluded. Can you provide a proof sketch and key insights into how the specific approach resulted in this computational complexity and how it overcame the bottleneck in the existing approaches?

3. The primary contribution is the computational advantage in high-dimensional settings. To validate the effectiveness of the proposed approach, experiments showcasing the variation of regret with respect to computational time will be beneficial.

    a) Can you include a plot showing regret vs. computation time for Ensemble++, Ensemble+, and the Langevin Monte Carlo approach?

    b) Provide a detailed comparison with the Langevin Monte Carlo approach in  Xu et al., highlighting both performance and computational efficiency.

    c) Discuss any trade-offs between regret and computation time for each method.

**Details Of Ethics Concerns:**

None.

---

### Official Review · Reviewer_DeKQ · 2024-11-03

**Soundness:** 3
**Presentation:** 2
**Contribution:** 3
**Rating:** 5
**Confidence:** 4

**Summary:**

This paper proposes an improved ensemble exploration method called Ensemble++, which incorporates decoupled optimization and lifted index sampling for more efficient exploration and uncertainty estimation. The architecture outperforms existing methods and is scalable in terms of computational cost.

**Strengths:**

(1) As demonstrated in the paper, Ensemble++ is computationally efficient, which is highly beneficial in practice.

(2) Ensemble++ addresses the ensemble coupling issue by utilizing the stop-gradient operator.

**Weaknesses:**

In my understanding, the primary weakness of the paper lies in the theoretical analysis in Section 4. Specifically, the impact of the stop-gradient operator on regret performance is unclear, which raises concerns about whether the regret analysis fully supports the main idea.  However, if I am missing something, please let me know, and I will reconsider my evaluation.

**Questions:**

More details should be provided on the stop-gradient operator in the network. Is it solely intended to block gradient flow? If so, how to do the ensemble process? I think the current explanation may not be sufficient for readers to fully understand this aspect.

---

### Official Review · Reviewer_PN7r · 2024-11-03

**Soundness:** 3
**Presentation:** 3
**Contribution:** 3
**Rating:** 6
**Confidence:** 4

**Summary:**

Scalable exploration is challenging in high-dimensional bandit settings. Ensembling is one (computationally expensive) way to try to approximate thompson sampling; ensembling can be made computationally cheaper by sharing one feature extractor between ensemble members, but can underestimate uncertainty due to the ensemble coupling issue, where ensemble members are too similar.

Authors propose Ensemble++, which uses stop gradients and a high-dimensional ensemble index. Empirically, authors find Ensemble++ outperforms existing methods in bandit settings. Authors analyze regret and computation for Ensemble++ in linear contextual bandit settings; Ensemble++ achieves the same regret bounds as exact TS, and has $\tilde O(\log T)$ per-step computation complexity.

============================
I appreciate the author's response to my questions and comments and have accordingly raised my score.

**Strengths:**

The motivation seems strong, the method is largely straightforward and well-motivated (see questions for exceptions), the paper seems to be overall well-written and well-organized, and the experiments seem promising.

**Weaknesses:**

* **Insufficient discussion of related work** While there is a mention and empirical comparison to epinet in the paper, I felt there was insufficient discussion of how the proposed method is methodologically different from epinets, as they also use index functions.

* **Unclear references to posterior approximation** There are references made to things approximating a posterior, but it was not clear what kind of an approximation that is (e.g. the way MCMC sampling is a posterior approximation, vs VI is a posterior approximation, vs a closed-form posterior for a different but similar problem is an approximation).


    * **Proposition 1** Proposition 1 is described as a closed-form update; while an earlier sentence mentions posterior approximation, from the Appendix, Proposition 1 is deriving the closed-form minimizers of the loss (2) with respect to $A$ and $\mu$. Is this supposed to also be a posterior update? If so, what are the assumptions on the Bayesian model?

    * **Lemma 1** This result assumes $\Sigma_t$ is the true posterior variance, but it is not clear why that should be the posterior variance. It seems like there are some assumptions and/or definitions missing.

    * **Experiments** There are no experiments that compare Ensemble++ vs other methods to a ground truth posterior, despite claims about Ensemble++ better approximating the posterior. (The first setting in 5.1 does appear to compare with ground-true TS, if I am understanding correctly, but this is the only example.)

* **Sequential dependency** I still don't understand what exactly is the problem in regular ensembles and how it is mitigated in Ensemble++.

* There are claims that the stop gradient prevents ensemble coupling; it would help if this could be empirically measured (beyond better bandit regret).

* I think it could help for readability to have an algorithm box to summarize the method.

Also see questions for areas that were confusing.

**Questions:**

* In the abstract (and elsewhere in the paper), how is ensemble sampling a computationally efficient approximation of Thompson sampling? It seems that it is in specific settings, e.g. the Gaussian setting in Lemma 3 in [1], and the linear setting in this paper where $P_\xi$ is zero-mean and isotropic. If I am understanding correctly, then this limitation in the connection between ensembles and posteriors needs to be specified.
* In line 195-196, it is mentioned that there are $M$ ensemble components, but $A$ is simply a matrix of size $d\times M$; are these $M$ ensemble components separate in any way?
* How is the variance of the $Z_{s,m}$ perturbations (139-140) chosen? Are these heuristics, or is there a principled reason for their existence?
* The abstract / intro mention computational complexity but I don't see it derived anywhere. Is this supposed to be in the paper?

[1] Ian Osband, John Aslanides, and Albin Cassirer. Randomized prior functions for deep reinforcement learning. Advances in Neural Information Processing Systems, 31, 2018.

---

### Official Review · Reviewer_zviA · 2024-11-05

**Soundness:** 2
**Presentation:** 3
**Contribution:** 3
**Rating:** 5
**Confidence:** 3

**Summary:**

Scalable exploration in sequential decision-making is challenging, especially in high-dimensional environments. Ensemble sampling, an approximation of Thompson sampling, is widely used but can suffer from performance degradation due to ensemble coupling in shared-layer networks. The Ensemble++ architecture is proposed to overcome this limitation by introducing decoupled optimization and lifted index sampling. Empirical results show that Ensemble++ outperforms existing methods in regret minimization across various tasks.

**Strengths:**

(1) The paper proposes the Ensemble++ architecture, which achieves scalability with O(d³log T) per-step computation. This closes a long-standing gap in scalable exploration theory.

(2) The paper empirically validates the scalability and efficiency of Ensemble++ through experiments in bandit tasks, including language-input contextual bandits using a GPT backbone.

**Weaknesses:**

I appreciate the motivation and approach presented. However, I find that the methods, analysis, and experiments are inconsistent. The motivation of this method is primarily based on ensemble neural networks, which suffer from gradient coupling issues. Yet, the analysis and experiments focus on linear contextual bandits, where gradient descent is not commonly used. Additionally, an important related work, "neural contextual bandits [1,2,3]," has been overlooked. I believe that the analysis from neural contextual bandits could be adapted to this method.


[1] Neural Contextual Bandits with UCB-based Exploration
[2] Ee-net: Exploitation-exploration neural networks in contextual bandits.
[3] Federated Neural Bandits

**Questions:**

(1) May include additional analysis or experiments on neural network models to demonstrate how Ensemble++ addresses the gradient coupling issue in that setting.

---

### Public Comment · ~Haque_Ishfaq1 · 2024-11-20
**Missing related works on perturbed history exploration and Langevin Monte Carlo based RL algorithms**

Dear author,

From the writing, it seems that Ensemble++ is proposed for general sequential decision making problems (that includes reinforcement learning as well) and only for the theoretical regret guarantee, the authors consider the linear contextual bandit setting. Despite claiming to be a general method for sequential decision making problems, the authors missed many relevant papers that tackles exploration in RL. I would like to highlight some of them.

In Line 38, you mention perturbed history exploration and cite Kventon et al 2020 which considered linear bandit setting. In Ishfaq et al 2021, we proposed RLSVI-PHE which uses perturbed history exploration in RL and provides provable regret bound in MDP under general function approximation setting. This work should be cited along with Kventon et al 2020.

In Line 46 (and other places) you discuss Langevin Monte Carlo based Thompson sampling (Xu et al 2022) which only considered bandit setting. However, Ishfaq et al 2024a and Ishfaq et al 2024b, both of them proposes Langevin Monte Carlo based RL algorithms which are both provably efficient and practically efficient for deep RL with SOTA performance among randomized exploration based approaches. These 2 works should also be cited with Xu et al 2022.

I would appreciate if you could cite these works as they are highly relevant and SOTA randomized exploration methods with provable regret guarantees.


```
Ishfaq, Haque, et al. "Randomized exploration in reinforcement learning with general value function approximation." International Conference on Machine Learning. PMLR, 2021.

Ishfaq, Haque, et al. "Provable and Practical: Efficient Exploration in Reinforcement Learning via Langevin Monte Carlo." The Twelfth International Conference on Learning Representations. (2024a)

Ishfaq, Haque, et al. "More Efficient Randomized Exploration for Reinforcement Learning via Approximate Sampling." Reinforcement Learning Conference (2024b).
```

---

### Note · Authors · 2025-10-26

**Comment:**

We are writing to formally withdraw our submission, "Scalable Exploration via Ensemble++" (Submission #11456). This paper has recently been accepted for publication at NeurIPS 2025.

We would like to sincerely thank the ICLR 2025 program chairs and the reviewers for their time and effort in reviewing our work.

**Withdrawal Confirmation:**

I have read and agree with the venue's withdrawal policy on behalf of myself and my co-authors.

---

### Meta-Review · Area_Chair_kVQ4 · 2024-12-21

**Metareview:**

This paper studies ensemble sampling, an approximation of Thompson sampling, which is widely used but prone to performance degradation due to ensemble coupling in shared-layer networks. To address this limitation, the authors propose the Ensemble++ architecture, introducing decoupled optimization and lifted index sampling. Empirical results suggest that the proposed method outperforms existing approaches in regret minimization. However, during the discussion phase, reviewers raised concerns about the consistency of the proposed method and its effectiveness in addressing the core issue of ensemble coupling. Additionally, the focus on regret analysis for linear contextual bandits felt disconnected from the primary challenge of gradient coupling in ensemble sampling. While theoretical insights from linear models can sometimes inform understanding of nonlinear methods, it remains unclear how the provided analysis specifically explains or resolves the challenges in neural network-based ensemble sampling methods. Given these concerns, reviewers were mostly unenthusiastic and leaned toward rejecting the paper. I encourage the authors to address these questions thoroughly and provide a more coherent and aligned presentation in their future version.

**Additional Comments On Reviewer Discussion:**

The reviewers appreciated the detailed explanations provided in the response, particularly regarding questions about computational complexity and comparisons with existing work in terms of algorithm design. However, the issue of incoherence between the theoretical analysis and the claimed challenge of ensemble sampling remains unresolved and less clearly addressed.

---

### Decision · Program_Chairs · 2025-01-22

Reject